# Intertwined order of generalized global symmetries

Benjamin Moy and Eduardo Fradkin

*Department of Physics and Anthony J. Leggett Institute for Condensed Matter Theory,*
*The Grainger College of Engineering,*
*University of Illinois at Urbana-Champaign,*
*1110 West Green Street, Urbana, Illinois 61801, USA*

December 5, 2024

### Abstract

We investigate the interplay of generalized global symmetries in 2+1 dimensions by introducing a lattice model that couples a $\mathbb{Z}_N$ clock model to a $\mathbb{Z}_N$ gauge theory via a topological interaction. This coupling binds the charges of one symmetry to the disorder operators of the other, and when these composite objects condense, they give rise to emergent generalized symmetries with mixed 't Hooft anomalies. These anomalies result in phases with ordinary symmetry breaking, topological order, and symmetry-protected topological (SPT) order, where the different types of order are not independent but intimately related. We further explore the gapped boundary states of these exotic phases and develop theories for phase transitions between them. Additionally, we extend our lattice model to incorporate a non-invertible global symmetry, which can be spontaneously broken, leading to domain walls with non-trivial fusion rules.

# 1 Introduction

Symmetry plays an fundamental role in physics and particularly in condensed matter physics. In the conventional Landau theory of phase transitions, phases of matter are classified by the global symmetries they spontaneously break. On the other hand, symmetry-protected topological phases (SPTs) are characterized by their nontrivial response to probes of global symmetries or by the 't Hooft anomalies of their boundary states [1–6].

Recent developments have expanded the concept of global symmetry in several ways, thus widening our perspective on how to characterize phases of matter [7–10]. While an ordinary global symmetry acts on local operators, a $q$-form global symmetry acts on operators supported on $q$-dimensional objects. Using this terminology, an ordinary global symmetry is a zero-form symmetry. We will be especially interested in one-form global symmetries, which act on operators defined on noncontractible loops, since these symmetries can be used to characterize topologically ordered phases [11, 12]. Indeed, the braiding of two Abelian anyons in (2+1)d results in a phase factor, which can be interpreted as resulting from the action of a one-form symmetry by the worldline of one anyon on another. A phase with Abelian topological order may then be thought of as "spontaneously" breaking a discrete one-form global symmetry in the sense that the symmetry is respected by the low energy effective action (or Hamiltonian) but not by a given ground state[1]. We emphasize that this one-form symmetry is an emergent infrared (IR) symmetry in the usual condensed matter setting, but once it emerges, it is extremely robust [14].

A distinct generalization of symmetry that has garnered much attention recently is non-invertible symmetry [15, 16]. Conventional symmetry operators are unitary (or antiunitary) and thus have inverses. Recently, it has been argued that topological operators that do not necessarily have inverses should be viewed as symmetry operators [17, 18]. While ordinary global symmetries are characterized mathematically by groups, symmetry operators for non-invertible symmetries can have multiple fusion channels. A flurry of recent work has introduced many examples of field theories [19–22] and lattice models [23–25] with non-invertible symmetries, demonstrating that these symmetries can lead to novel SPTs [26, 27]

---

[1]For ordinary zero-form global symmetries, the *precise* definition of spontaneous symmetry breaking is that upon coupling to a local external symmetry-breaking field $h(x)$, the order parameter remains nonzero if we take the thermodynamic limit and then $h \to 0$. A analogue of this criterion does not exist for higher-form symmetries because the observables are non-local. However, a spontaneously broken one-form global symmetry can be defined as the condition that the correlators of Wilson operators on noncontractible loops obey clustering. A well-known example is the Polyakov loop used to diagnose the thermal phase transition from a confined to a deconfined phase of a pure gauge theory [13].

and intricate phase diagrams with exotic symmetry-breaking patterns [28–30].

Experience with ordinary global symmetries suggests that the interplay of different symmetries can give rise to rich phase diagrams with a variety of interesting phases. In particular, the concept of intertwined order has been invoked to understand general principles at play in the complex phase diagrams of correlated electronic systems [31]. The idea is that the different orders in a complex phase diagram can arise from an underlying microscopic primary state. A paradigmatic example of a possible parent state is a pair density wave superconductor [32], which spontaneously breaks $U(1)$ particle number conservation and translation symmetry, which hosts several ordered states.

We therefore should expect that a rich set of physics to arise from models with an interplay of generalized global symmetries. In this work, we develop such models with intertwined generalized symmetries. One way in which generalized symmetries can be intertwined is that they can form a higher-group symmetry [33, 34], but we take a different approach here. With some inspiration from a previous work of the authors [35], our starting point is a 3d Euclidean lattice model consisting of a $\mathbb{Z}_N$ clock model coupled to a $\mathbb{Z}_N$ gauge theory (defined on dual lattices) by a topological term rather than by minimal coupling[2]. This topological coupling is analogous to the theta term in (3+1)d gauge theories, which gives magnetic monopoles an electric charge—a phenomenon known as the Witten effect [36]. Similarly, our (2+1)d topological interaction binds vortices of the spin model to an electric charge of the gauge theory, and the magnetic monopoles of the gauge theory become bound to the spins. Different combinations of these composite operators can condense, leading to a rich phase diagram with a variety of interesting phases. Using duality arguments, we can deduce much of this phase diagram and the physics of the phases in the model.

One class of phases that arise in this model are SPTs protected by the $\mathbb{Z}_N$ zero-form and $\mathbb{Z}_N$ one-form global symmetries. We denote a $\mathbb{Z}_N$ $q$-form global symmetry by $\mathbb{Z}_N^{(q)}$ so that the symmetry protecting the SPTs of this type is $G = \mathbb{Z}_N^{(0)} \times \mathbb{Z}_N^{(1)}$. These SPT phases are characterized in the continuum by the response,

$$S_{\mathrm{SPT}}[A_\mu, B_{\mu\nu}] = \frac{iNp}{2\pi} \int A \wedge B = \frac{iNp}{4\pi} \int d^3x \, \varepsilon_{\mu\nu\lambda} \, A_\mu \, B_{\nu\lambda}, \qquad (1.1)$$

where $A_\mu$ is a $\mathbb{Z}_N$ one-form background gauge field, $B_{\mu\nu}$ is a $\mathbb{Z}_N$ two-form background gauge field, and $p \in \mathbb{Z}$ is a parameter characterizing the SPT phase. A lattice model for an SPT of this type was first described for $N = 2$ and $p = 1$ on the triangular lattice in Ref. [37]. For a given $N$, an SPT with each possible value of $p$ is realized in a phase of the lattice model.

---

[2]In this model the $\mathbb{Z}_N$ global symmetry of the clock model is not gauged and the $\mathbb{Z}_N$ local symmetry of the gauge theory is not Higgsed.

Our lattice model thus serves as a microscopic model for every SPT phase with a response of the form in Eq. (1.1), thus providing a versatile platform to study these phases and their phase transitions.

Our lattice model also admits more intricate phases characterized by an effective field theory that is a gauged version of Eq. (1.1), meaning that $A_\mu$ and $B_{\mu\nu}$ are both promoted to fluctuating gauge fields. By the correspondence between SPTs and topologically ordered states [38], these phases are symmetry enriched topological states (SETs) that have topological order, but there is some additional symmetry breaking of the ordinary global symmetry. These phases are generated by simultaneously condensing bound states of local operators with spin charge $N$ and magnetic charge $p$ and loop operators with electric charge $N$ and vorticity $p$, which is reminiscent of the physics of oblique confining phases in (3+1)d gauge theories [39–41], so we refer to this class of (2+1)d phases as oblique phases. We demonstrate that for $L = \gcd(N, p) > 1$, the $G = \mathbb{Z}_N^{(0)} \times \mathbb{Z}_N^{(1)}$ symmetry has mixed 't Hooft anomalies with emergent generalized symmetries, resulting in a breaking of this symmetry to a nontrivial subgroup, $H = \mathbb{Z}_{N/L}^{(0)} \times \mathbb{Z}_{N/L}^{(1)}$, giving rise to both ordinary symmetry breaking and topological order. Furthermore, the unbroken subgroup $H$ has a nontrivial response to background fields, signaling SPT order for this symmetry.

To emphasize the new features of our lattice model, let us contrast with another physical system that has both ordinary symmetry breaking and topological order. For example, fractional quantum Hall systems can certainly host broken symmetry states with nematic order that coexists with topological order [42, 43]. However, the symmetry structure of the systems we are studying here are more intricate. The physics described above for an oblique phase is representative of the interplay of the zero-form and one-form symmetries in our lattice model more generally. The patterns of symmetry breaking for the two symmetries are not independent and do not merely coexist. For *any* phase of our lattice model, if the $\mathbb{Z}_N$ zero-form symmetry is spontaneously broken to a nontrivial subgroup, the $\mathbb{Z}_N$ one-form symmetry is also broken to the *same* subgroup. The remaining unbroken zero-form and one-form symmetries additionally have mixed SPT order. These features demonstrate that the symmetries are not independent but indeed have a special interplay throughout the phase diagram because of the way in which they are coupled in the lattice model.

For systems with an open boundary, for every SPT of the form in Eq. (1.1), preserving the zero-form and one-form symmetries inevitably leads to boundary modes. The one-form symmetry cannot be spontaneously broken along the (1+1)d boundary, so any gapped boundary state must necessarily break the zero-form symmetry spontaneously. We develop the precise criteria the boundary theory must satisfy to cancel the 't Hooft anomaly of

the bulk and provide examples. We also develop two different gapped boundary states for oblique phases. One of these boundary conditions, which we refer to as an electric boundary condition, preserves the $G = \mathbb{Z}_N^{(0)} \times \mathbb{Z}_N^{(1)}$ symmetry of the bulk. This boundary condition is consistent with the SPT order of the unbroken $H = \mathbb{Z}_{N/L}^{(0)} \times \mathbb{Z}_{N/L}^{(1)}$ subgroup and results in the spontaneous breaking of $H$ at the boundary. The other gapped boundary condition, which we call the magnetic boundary state, preserves a magnetic $\mathbb{Z}_p^{(0)} \times \mathbb{Z}_p^{(1)}$ symmetry, resulting in the spontaneous breaking of this symmetry at the boundary.

We also find possible gapless boundary states for $\mathbb{Z}_N^{(0)} \times \mathbb{Z}_N^{(1)}$ SPTs. By analyzing possible perturbations that gap the boundary, we show that the gapless state can be interpreted as a quantum critical point (or line) that has additional emergent symmetries. In the simplest case, we consider the gapless boundary state of the SPT, Eq. (1.1), and add only perturbations that preserve the $\mathbb{Z}_N^{(0)} \times \mathbb{Z}_N^{(1)}$ symmetry coupled to the bulk. For $L = \gcd(N, p) \geq 4$, our gapless state is a quantum critical point (critical line for $L > 4$) where the global symmetry is enhanced to $U(1)^{(0)} \times U(1)^{(0)} \times \mathbb{Z}_N^{(0)} \times \mathbb{Z}_N^{(1)}$. This critical point or line then separates two gapped phases, each of which has a different $\mathbb{Z}_N^{(0)}$ symmetry that is spontaneously broken.

Finally, we consider generalizations of our Euclidean lattice model in which the parameter $\Theta$, the coefficient of the topological interaction, is promoted to a dynamical matter field associated with an additional $\mathbb{Z}_N^{(0)}$ global symmetry. We refer to this new matter field as a $\mathbb{Z}_N$ axion in analogy with the usual axion that couples to the theta term of a gauge theory in (3+1)d. The new $(\mathbb{Z}_N^{(0)})_{\text{axion}}$ symmetry has a mixed 't Hooft anomaly with the original $G = \mathbb{Z}_N^{(0)} \times \mathbb{Z}_N^{(1)}$ symmetry, which ensures that no phase of this lattice model is gapped and preserves all the symmetries. An especially interesting phase occurs when the $(\mathbb{Z}_N^{(0)})_{\text{axion}}$ symmetry is spontaneously broken while the remaining $G$ symmetry is preserved. In this case, the domain walls for the $(\mathbb{Z}_N^{(0)})_{\text{axion}}$ symmetry separate distinct $G$ SPT states.

If we now gauge the $G = \mathbb{Z}_N^{(0)} \times \mathbb{Z}_N^{(1)}$ symmetry of the lattice model, the domain walls will now separate distinct oblique states characterized by different topological orders, spontaneous symmetry breaking, and mixed SPT orders. As with the boundary states of the oblique phases, we must then necessarily decorate the domain walls with dynamical degrees of freedom, which is somewhat reminiscent of the decorated domain wall construction for SPTs [44], but here, the domain walls are decorated with a lower-dimensional state with spontaneous symmetry breaking rather than SPT order. The physical consequence of introducing these degrees of freedom is that the domain walls no longer obey group-like fusion rules. This generalization of our lattice model thus has a phase in which a non-invertible symmetry is spontaneously broken. Indeed, the mixed anomaly implies that gauging $G$ anomalously breaks the $(\mathbb{Z}_N^{(0)})_{\text{axion}}$ symmetry, but we demonstrate that it can alternatively

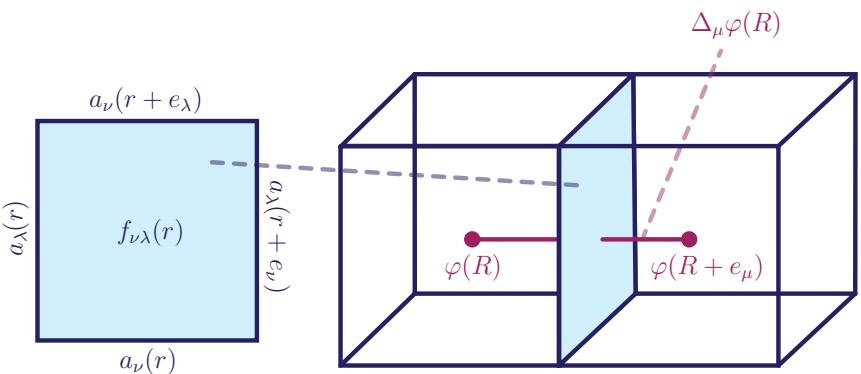

**Figure 1:** Depiction of the topological interaction on the lattice, Eq. (2.4). This (2+1)d analogue of the theta term represents an interaction of the field strengths, $f_{\nu\lambda}$, which are on plaquettes of the direct lattice, and spin phase variations, $\Delta_\mu\varphi$, which live on links of the dual lattice.

be viewed as a non-invertible symmetry. This notion of generating non-invertible symmetries from mixed anomalies as appeared in other contexts previously [19, 21].

This paper is organized as follows. In Section 2, we introduce our lattice model and discuss its global symmetries and phase diagram. Section 3 describes the low energy physics and effective field theory of the oblique phases. In Section 4, we discuss the intertwined SPT response of a generic oblique phase. Section 5 develops the phase transitions of the bulk phases of the lattice model in more detail. We then derive possible gapped boundary states of both the SPT state and the oblique phases in Section 6. Section 7 introduces a Hamiltonian lattice model that realizes oblique phases as its ground states. In Section 8, we develop gapless boundary states of the SPT states and present examples of boundary phase transitions into other gapped boundary states. Finally, in Section 9, we study the two $\mathbb{Z}_N$ axion generalizations of our lattice model, one of which has mixed anomalies of generalized symmetries in the UV, and another which has a non-invertible symmetry. In Appendix A, we review $\mathbb{Z}_N$ gauge fields in the continuum. In Appendix B, we find dualities of our lattice model. In Appendix C, we derive an effective Coulomb gas action for our lattice model and relate it to duality. Appendix D reviews the lattice version of (2+1)d BF theory and its counterpart in the ordered phase of the clock model. Appendix E contains some of the more technical calculations associated with the $\mathbb{Z}_N$ axion models.

## 2 Euclidean lattice model

### 2.1 Model

To study the interplay of zero-form and one-form global symmetries, we should use a model that couples these symmetries together. We thus consider a model of a $\mathbb{Z}_N$ lattice gauge theory coupled to $\mathbb{Z}_N$ matter on a three-dimensional cubic lattice in Euclidean spacetime[3]. However, the matter is *not* minimally coupled to the gauge field because this type of coupling removes the global symmetries we are interested in examining. Instead, we place the gauge fields $a_\mu$ on links of the lattice, matter fields $\varphi$ on sites of the dual lattice (i.e., centers of cubes on the direct lattice), and couple the matter to the gauge fields using a lattice analogue of a topological term. This choice of model was largely inspired by previous work of the authors [35] on the Cardy-Rabinovici model [40, 41, 48–50], a (3+1)d $\mathbb{Z}_N$ lattice gauge theory with an analogue of a theta term. Here, we consider a partition function in the Villain form [51], given by

$$
Z = \int [da_\mu][d\varphi] \sum_{\{n,\, n_\mu,\, s_\mu,\, s_{\mu\nu}\}} \delta(\Delta_\mu n_\mu)\, e^{-S[\varphi,\, a_\mu,\, n,\, n_\mu,\, s_\mu,\, s_{\mu\nu}]},
$$

$$
S = \frac{1}{4e^2} \sum_P (\Delta_\mu a_\nu - \Delta_\nu a_\mu - 2\pi s_{\mu\nu})^2 - iN \sum_\ell n_\mu\, a_\mu + \frac{1}{2g^2} \sum_{\tilde{\ell}} (\Delta_\mu \varphi - 2\pi s_\mu)^2
$$

$$
- iN \sum_R n(R)\varphi(R) + \frac{iN\Theta}{8\pi^2} \sum_{r,\, R} \varepsilon_{\mu\nu\lambda} (\Delta_\mu \varphi - 2\pi s_\mu)(R)(\Delta_\nu a_\lambda - \Delta_\lambda a_\nu - 2\pi s_{\nu\lambda})(r),
$$

$$\tag{2.1}$$

where $\Delta_\mu \varphi(R) \equiv \varphi(R + e_\mu) - \varphi(R)$ denotes the lattice difference along the link between site $R$ and site $R + e_\mu$. The gauge field $a_\mu \in \mathbb{R}$ and electric particle worldlines $n_\mu \in \mathbb{Z}$ are on links $\ell$ between sites $r$, and the Dirac strings $s_{\mu\nu} \in \mathbb{Z}$ of the monopoles live on plaquettes $P$. Meanwhile, $\varphi \in \mathbb{R}$ and $n \in \mathbb{Z}$ live on sites $R$ of the dual lattice, and the $s_\mu \in \mathbb{Z}$ are on links $\tilde{\ell}$ of the dual lattice. The last term, a (2+1)d analogue of the theta term in (3+1)d, is an interaction between each plaquette and the link on the dual lattice that intersects it (see Figure 1).

In the way we have expressed Eq. (2.1), we have embedded the $\mathbb{Z}_N$ gauge theory and the $\mathbb{Z}_N$ matter theory into $U(1)$ theories. Indeed, if $n_\mu = n = 0$, then the action reduces to that of $U(1)$ gauge theory coupled to an $XY$ model by a topological term. But the gauge group and matter fields should still be regarded as $\mathbb{Z}_N$ fields. By the Poisson summation

---

[3]See Refs. [45–47] for a review of lattice gauge theory.

formula, summing over $n_\mu$ and $n$ introduce constraints that $\varphi, a_\mu \in \frac{2\pi}{N}\mathbb{Z}$. Hence, we can either interpret $n_\mu$ and $n$ as electric worldlines and spin operators respectively, or we can view them as Lagrange multipliers that constrain $a_\mu$ and $\varphi$ to be $\mathbb{Z}_N$ fields.

The lattice model is compact, so there are additionally two classes of disorder operators (for a review see Ref. [52]) in this theory. The vortex currents of the spin model are represented by

$$m_\mu(r) = \varepsilon_{\mu\nu\lambda} \Delta_\nu s_\lambda \tag{2.2}$$

and are on links of the direct lattice. The monopoles of the gauge theory,

$$m(R) = -\frac{1}{2}\varepsilon_{\mu\nu\lambda} \Delta_\mu s_{\nu\lambda}, \tag{2.3}$$

are on sites of the dual lattice. To elucidate the physical role of the topological term, it is useful to express this term as

$$S_{\text{topo}} = \frac{iN\Theta}{8\pi^2} \sum_{r,R} \varepsilon_{\mu\nu\lambda} (\Delta_\mu\varphi - 2\pi s_\mu)(\Delta_\nu a_\lambda - \Delta_\lambda a_\nu - 2\pi s_{\nu\lambda}) \tag{2.4}$$

$$= \sum_{r,R} \left( \frac{iN\Theta}{4\pi^2} \varepsilon_{\mu\nu\lambda} \Delta_\mu\varphi \, \Delta_\nu a_\lambda + \frac{iN\Theta}{2} \varepsilon_{\mu\nu\lambda} \, s_\mu \, s_{\nu\lambda} - \frac{iN\Theta}{2\pi}(m_\mu \, a_\mu + m \, \varphi) \right). \tag{2.5}$$

The last two terms couple the vortices and monopoles to $a_\mu$ and $\varphi$ just like $n_\mu$ and $n$ respectively. Thus, the topological term effectively shifts the quantum numbers to

$$(n, m) \mapsto \left( n + \frac{\Theta}{2\pi} m, m \right), \qquad (n_\mu, m_\mu) \mapsto \left( n_\mu + \frac{\Theta}{2\pi} m_\mu, m_\mu \right). \tag{2.6}$$

The vortices of the spin model gain an electric charge proportional to $\Theta$, and the monopoles of the gauge theory obtain a fractional $\mathbb{Z}_N$ spin. This phenomenon is the analogue of the Witten effect in (3+1)d, under which a magnetic monopole gains an electric charge proportional to $\Theta$ when there is a theta term [36]. For this reason, we refer to Eq. (2.6) as the *generalized Witten effect*.

## 2.2 Global symmetries

As mentioned previously, our model is equipped with two global symmetries, which we now discuss more formally. There is an ordinary (i.e., zero-form) $\mathbb{Z}_N$ global symmetry under which $\varphi$ transforms as

$$\varphi \to \varphi + \frac{2\pi}{N}. \tag{2.7}$$

The local operator that carries the $\mathbb{Z}_N$ charge is

$$V(R) = e^{i\varphi(R)} \tag{2.8}$$

on a dual lattice site $R$. Under the symmetry transformation, Eq. (2.7), this operator transforms as

$$V(R) \to \omega V(R), \tag{2.9}$$

where $\omega$ is an $N$th root of unity. There is also a $\mathbb{Z}_N$ one-form symmetry that transforms the gauge field as

$$a_\mu \to a_\mu + \frac{2\pi\eta_\mu}{N}, \tag{2.10}$$

where $\eta_\mu \in \mathbb{Z}$ and $\Delta_\mu \eta_\nu - \Delta_\nu \eta_\mu = 0$ (i.e., $\eta_\mu$ is a flat connection). The gauge invariant operator that transforms under this global symmetry is the Wilson loop,

$$W(\Gamma) = \prod_{\ell \in \Gamma} e^{ia_\mu(\ell)}, \tag{2.11}$$

which is a product over all links $\ell$ which define the noncontractible loop $\Gamma$. The one-form global symmetry, Eq. (2.10), transforms the Wilson loop to

$$W(\Gamma) \to \omega W(\Gamma), \tag{2.12}$$

where $\omega$ is again an $N$th root of unity. As mentioned in Section 1, we will use the notation $\mathbb{Z}_N^{(q)}$ to denote a $\mathbb{Z}_N$ $q$-form symmetry, so our lattice model in Eq. (2.1) then has the global symmetry $G = \mathbb{Z}_N^{(0)} \times \mathbb{Z}_N^{(1)}$.

To probe these global symmetries, it is useful to couple to background fields. The $\mathbb{Z}_N^{(0)}$ global symmetry couples to a background field $2\pi A_\mu / N$ where $A_\mu \in \mathbb{Z}$. Similarly, the $\mathbb{Z}_N^{(1)}$ couples to a two-form field $2\pi B_{\mu\nu}/N$, where $B_{\mu\nu} \in \mathbb{Z}$ is antisymmetric in its indices. In the presence of background fields, the partition function becomes

$$Z[A_\mu, B_{\mu\nu}] = \int [da_\mu][d\varphi] \sum_{\{n, n_\mu, s_\mu, s_{\mu\nu}\}} \delta(\Delta_\mu n_\mu) \, e^{-S[\varphi, a_\mu, n, n_\mu, s_\mu, s_{\mu\nu}; A_\mu, B_{\mu\nu}]},$$

$$S = \frac{1}{4e^2} \sum_P (f_{\mu\nu}[B])^2 - iN \sum_\ell n_\mu a_\mu + \frac{1}{2g^2} \sum_{\tilde{\ell}} (\omega_\mu[A])^2 - iN \sum_R n(R)\varphi(R)$$

$$+ \frac{iN\Theta}{8\pi^2} \sum_{r,R} \varepsilon_{\mu\nu\lambda} (\omega_\mu[A]) (f_{\nu\lambda}[B]),$$

$$\tag{2.13}$$

where we have defined

$$\omega_\mu[A] = \Delta_\mu \varphi - 2\pi s_\mu - \frac{2\pi A_\mu}{N}, \qquad f_{\mu\nu}[B] = \Delta_\mu a_\nu - \Delta_\nu a_\mu - 2\pi s_{\mu\nu} - \frac{2\pi B_{\mu\nu}}{N}. \qquad (2.14)$$

The gauge transformations for the background fields act as

$$A_\mu \to A_\mu + \Delta_\mu \chi + N \mathcal{N}_\mu, \qquad \varphi \to \varphi + \frac{2\pi}{N} \xi, \qquad s_\mu \to s_\mu - \mathcal{N}_\mu, \qquad (2.15)$$

$$B_{\mu\nu} \to B_{\mu\nu} + \Delta_\mu \xi_\nu - \Delta_\nu \xi_\mu + N \mathcal{M}_{\mu\nu}, \qquad a_\mu \to a_\mu + \frac{2\pi \xi_\mu}{N}, \qquad s_{\mu\nu} \to s_{\mu\nu} - \mathcal{M}_{\mu\nu},$$

where $\chi, \xi_\mu, \mathcal{N}_\mu, \mathcal{M}_{\mu\nu} \in \mathbb{Z}$. From the background fields, $A_\mu$ and $B_{\mu\nu}$, we can define corresponding field strengths,

$$F_{\mu\nu} = \Delta_\mu A_\nu - \Delta_\nu A_\mu, \qquad H_{\mu\nu\lambda} = \Delta_\mu B_{\nu\lambda} + \Delta_\nu B_{\lambda\mu} + \Delta_\lambda B_{\mu\nu}, \qquad (2.16)$$

which are antisymmetric in all their indices. On the lattice, these field strengths can take any value (mod $N$). In the continuum limit, as reviewed in Appendix A, any gauge field for a discrete symmetry must be locally flat (i.e., $F_{\mu\nu} = H_{\nu\lambda\sigma} = 0$ mod $N$). Since our interest is ultimately in the continuum limit, we will require $A_\mu$ and $B_{\mu\nu}$ to be locally flat. In the next section, we will see that coupling to probes for the global symmetries is an important tool that will help us determine the phases and phase diagram.

## 2.3 Duality

Before turning to the phase diagram of our lattice model, Eq. (2.1), it is useful to determine any non-perturbative information we can exploit to constrain the phase diagram. The model turns out to have two kinds of duality transformations. First, we note that in the absence of background fields, the partition function is periodic under shifts of $\Theta$ by integer multiples of $2\pi$. If we also introduce background gauge fields and perform the shift $\Theta \to \Theta + 2\pi$, the action in Eq. (2.13) changes by

$$\Delta S = \frac{iN2\pi}{8\pi^2} \sum_{r,R} \varepsilon_{\mu\nu\lambda} \left( \Delta_\mu \varphi - 2\pi s_\mu - \frac{2\pi A_\mu}{N} \right) \left( \Delta_\nu a_\lambda - \Delta_\lambda a_\nu - 2\pi s_{\nu\lambda} - \frac{2\pi B_{\nu\lambda}}{N} \right) \qquad (2.17)$$

$$= \frac{iN}{2\pi} \sum_{r,R} \varepsilon_{\mu\nu\lambda} \Delta_\mu \varphi \, \Delta_\nu a_\lambda + i2\pi N \sum_{r,R} \frac{1}{2} \varepsilon_{\mu\nu\lambda} \left[ s_\mu \left( s_{\nu\lambda} + \frac{B_{\nu\lambda}}{N} \right) + \varepsilon_{\mu\nu\lambda} \frac{A_\mu}{N} s_{\nu\lambda} \right] \qquad (2.18)$$

$$- iN \sum_R \left( m - \frac{1}{2N} \varepsilon_{\mu\nu\lambda} \Delta_\mu B_{\nu\lambda} \right) \varphi - iN \sum_\ell \left( m_\mu + \frac{1}{N} \varepsilon_{\mu\nu\lambda} \Delta_\nu A_\lambda \right) a_\mu$$

$$+ \frac{iN}{2\pi} \sum_{r,R} \varepsilon_{\mu\nu\lambda} \frac{2\pi A_\mu}{N} \frac{1}{2} \frac{2\pi B_{\nu\lambda}}{N}.$$

The first term in the first line of Eq. (2.18) is a total lattice derivative and thus will not contribute if we take periodic boundary conditions. The remaining terms in the first line of Eq. (2.18) are integer multiples of $2\pi i$, so they also can be ignored. The terms in the second line of Eq. (2.18) may be removed by correspondingly shifting

$$n \to n - m + \frac{1}{2N} \, \varepsilon_{\mu\nu\lambda} \, \Delta_\mu B_{\nu\lambda}, \qquad n_\mu \to n_\mu - m_\mu - \frac{1}{N} \, \varepsilon_{\mu\nu\lambda} \, \Delta_\nu A_\lambda. \tag{2.19}$$

Because the background fields are locally flat, Eq. (2.19) is a shift of $n$ and $n_\mu$ by integers, so such a transformation is well-defined. After the shift $\Theta \mapsto \Theta + 2\pi$, the only term we cannot remove is

$$\Delta S = \frac{iN}{2\pi} \sum_{r,\,R} \varepsilon_{\mu\nu\lambda} \frac{2\pi A_\mu}{N} \frac{1}{2} \frac{2\pi B_{\nu\lambda}}{N}. \tag{2.20}$$

Thus, although the partition function is invariant under shifting $\Theta$ by $2\pi$ in the absence of background fields, it gains an additional phase factor in the presence of background fields. Hence, we may regard shifting $\Theta$ by $2\pi$ as stacking a $G = \mathbb{Z}_N^{(0)} \times \mathbb{Z}_N^{(1)}$ SPT.

Next, we turn to the other duality transformation, which is analogous to Kramers-Wannier duality. As demonstrated in Appendix C, by manipulating the partition function, Eq. (2.13), in ways similar to Refs. [7, 53], we can find a dual theory with action,

$$\begin{aligned}
S =\ & \frac{1}{4\tilde{e}^2} \sum_P (\tilde{f}_{\mu\nu}[b])^2 - iN \sum_\ell \tilde{n}_\mu \, \tilde{a}_\mu + \frac{1}{2\tilde{g}^2} \sum_{\tilde\ell} (\tilde\omega_\mu[c])^2 - iN \sum_R \tilde{n}(R)\tilde\varphi(R) \\
& - \frac{iN\widetilde\Theta}{8\pi^2} \sum_{r,\,R} \varepsilon_{\mu\nu\lambda} \, \tilde\omega_\mu[c] \, \tilde{f}_{\nu\lambda}[b] + \frac{\pi i}{N} \sum_{r,\,R} \varepsilon_{\mu\nu\lambda} \left( b_{\nu\lambda} \, \Delta_\mu \tilde{k} + k_\mu \, \Delta_\nu c_\lambda \right) \\
& + \frac{\pi i}{N} \sum_{r,\,R} \varepsilon_{\mu\nu\lambda} \left( c_\mu \, B_{\nu\lambda} + b_{\mu\nu} \, A_\lambda \right),
\end{aligned} \tag{2.21}$$

where we use the abbreviated notation,

$$\tilde\omega_\mu[c] = \Delta_\mu \tilde\varphi - 2\pi \tilde{s}_\mu - \frac{2\pi c_\mu}{N}, \qquad \tilde{f}_{\mu\nu}[b] = \Delta_\mu \tilde{a}_\nu - \Delta_\nu \tilde{a}_\mu - 2\pi \tilde{s}_{\mu\nu} - \frac{2\pi b_{\mu\nu}}{N}, \tag{2.22}$$

and we have defined

$$\frac{\tilde{e}^2}{e^2} = \frac{\tilde{g}^2}{g^2} = \left( \frac{\Theta}{2\pi} \right)^2 + \left( \frac{2\pi}{Nge} \right)^2, \qquad \widetilde\Theta = -\frac{\Theta}{\left( \frac{\Theta}{2\pi} \right)^2 + \left( \frac{2\pi}{Nge} \right)^2}. \tag{2.23}$$

The gauge field $\tilde{a}_\mu \in \mathbb{R}$ is on links of the direct lattice, and $\tilde\varphi \in \mathbb{R}$ is on sites of the dual lattice. We also have $\tilde{n}_\mu \in \mathbb{Z}$ lives on links and $\tilde{n} \in \mathbb{Z}$ on dual sites. The variables $\tilde{s}_{\mu\nu}, b_{\mu\nu} \in \mathbb{Z}$ are on plaquettes while $\tilde{s}_\mu, c_\mu \in \mathbb{Z}$ are on dual links. Finally, $k_\mu \in \mathbb{Z}$ are on links, and $\tilde{k} \in \mathbb{Z}$

are on dual sites. All these lattice variables are dynamical except $A_\mu$ and $B_{\mu\nu}$, which are probes.

The duality of Eq. (2.21) and Eq. (2.13) demonstrates that our lattice model is locally self-dual. Indeed, Eq. (2.21) is similar to Eq. (2.13) but has two important differences. First, the coupling constants have been modified. This mapping of coupling constants is nicely packaged if we define the complex coupling constant $\tau$ and the coupling ratio $\kappa$ to be

$$\tau = \frac{\Theta}{2\pi} + i\frac{2\pi}{Nge}, \qquad \kappa = \frac{e}{g}. \tag{2.24}$$

Duality then maps the coupling constants as

$$\mathcal{S}: \qquad \tau \to -\frac{1}{\tau}, \qquad \kappa \to \kappa. \tag{2.25}$$

Secondly, a more subtle difference is that the flat background gauge fields in Eq. (2.13) are now dynamical in Eq. (2.21), so the lattice model is dual to a gauged version of itself. This duality is a generalization of the duality relating a $\mathbb{Z}_N$ clock model and a $\mathbb{Z}_N$ gauge theory in (2+1)d.

To summarize, we have two kinds of duality transformations. The partition function $Z[\tau; A_\mu, B_{\mu\nu}]$ depends on[4] the complex coupling constant $\tau$ and the background fields $A_\mu$ and $B_{\mu\nu}$. We then define transformations $\mathcal{T}$ and $\mathcal{S}$ of the partition function as

$$\mathcal{T}\left(Z[\tau; A_\mu, B_{\mu\nu}]\right) = Z[\tau + 1; A_\mu, B_{\mu\nu}] = Z[\tau; A_\mu, B_{\mu\nu}]e^{-\frac{\pi i}{N}\sum_{r,R}\varepsilon_{\mu\nu\lambda}A_\mu B_{\nu\lambda}},$$

$$\mathcal{S}\left(Z[\tau; A_\mu, B_{\mu\nu}]\right) = Z\left[-1/\tau; A_\mu, B_{\mu\nu}\right] = \sum_{\{a_\mu, b_{\mu\nu}\}} Z[\tau; a_\mu, b_{\mu\nu}]e^{-\frac{\pi i}{N}\sum_{r,R}\varepsilon_{\mu\nu\lambda}(a_\mu B_{\nu\lambda}+b_{\mu\nu}A_\lambda)}.$$

$$\tag{2.26}$$

where $a_\mu, b_{\mu\nu} \in \mathbb{Z}$ are locally flat dynamical $\mathbb{Z}_N$ gauge fields while $A_\mu, B_{\mu\nu} \in \mathbb{Z}$ are locally flat background $\mathbb{Z}_N$ gauge fields. It is also useful to consider the continuum analogues of the transformations $\mathcal{S}$ and $\mathcal{T}$. The analogous transformations for a Euclidean action $S$ are

$$\mathcal{T}: \qquad S[A, B] \to S[A, B] + \frac{iN}{2\pi}\int A \wedge B,$$

$$\mathcal{S}: \qquad S[A, B] \to S[a, b] + \frac{iN}{2\pi}\int (a \wedge B + b \wedge A), \tag{2.27}$$

where $a_\mu$ and $b_{\mu\nu}$ are dynamical $\mathbb{Z}_N$ gauge fields while $A_\mu$ and $B_{\mu\nu}$ are background $\mathbb{Z}_N$ gauge fields. We will use both Eqs. (2.26) and (2.27) interchangeably. The mathematical structure

---

[4]The partition function also depends on $\kappa = e/g$, but we suppress that dependence since $\kappa$ is invariant under $\mathcal{T}$ and $\mathcal{S}$ in Eq. (2.26).

of the $\mathcal{S}$ and $\mathcal{T}$ transformations is reminiscent of that in the fractional quantum Hall effect and (2+1)d conformal field theories [54, 55]. A similar structure also arises in (3+1)d lattice gauge theory with a theta term [41].

As we demonstrate in Appendix C, we can deduce how the $\mathcal{S}$ and $\mathcal{T}$ transformations map between different operators, allowing us determine what objects condense in a given phase. The local operators are labeled $(q_s, q_m)$ by a spin charge $q_s$ and a monopole charge $q_m$, and the loop operators are labeled $(q_e, q_v)$ by an electric charge $q_e$ and a vorticity $q_v$. Because the dynamical spins and electric charges are multiples of $N$, we note that a local operator that can condense must have $q_s \in N\mathbb{Z}$, and a loop operator that condense must have an electric charge $q_e \in N\mathbb{Z}$. The $\mathcal{S}$ and $\mathcal{T}$ transformations act on these objects as

$$
\begin{aligned}
\mathcal{S}: \quad & \tau \mapsto -\frac{1}{\tau}, \quad & (q_e, q_v) \mapsto (-Nq_v, q_e/N), \quad & (q_s, q_m) \mapsto (-Nq_m, q_s/N), \\
\mathcal{T}: \quad & \tau \mapsto \tau + 1, \quad & (q_e, q_v) \mapsto (q_e - Nq_v, q_v), \quad & (q_s, q_m) \mapsto (q_s - Nq_m, q_m).
\end{aligned}
\tag{2.28}
$$

We can then place many constraints on the phase diagram using these transformations. For example, if a certain phase has local operators with $(q_s, q_m)$ condensed, then the image of this phase under $\mathcal{S}$ will have local operators with charges $(-Nq_m, q_s/N)$ condensed. The phase diagram can then be deduced by acting with a series of $\mathcal{S}$ and $\mathcal{T}$ transformations on the phases near $\Theta = 0$.

## 2.4 Phases and phase diagram

We are now equipped to understand the phases of the model and how the phase diagram is organized. We begin with the phase structure at $\Theta = 0$, where we have a decoupled $\mathbb{Z}_N$ clock model and $\mathbb{Z}_N$ gauge theory. Here, the physics is well understood. The clock model has two phases—the ordered phase and the disordered phase. At small $g$, the $\mathbb{Z}_N$ spins condense into an ordered phase in which the $\mathbb{Z}_N^{(0)}$ symmetry is spontaneously broken completely. (See Appendix D.1 for a review of the effective field theory of this phase at low energies.) At a large $g$, the clock model is in a trivial disordered phase in which the vortices condense. These two phases are separated by a single direct phase transition [56–60] at a finite value of $g$, which we define as $g_c$. This phase transition is expected to be a continuous transition for all $N$ except $N = 3$ where there is solid numerical evidence that it is weakly first order [56, 61–63].

Next, we turn to the physics of $\mathbb{Z}_N$ gauge theory, which may be established using duality and our knowledge of the $\mathbb{Z}_N$ spin model [46, 64]. The $\mathbb{Z}_N$ gauge theory at coupling $e$ is dual to a gauged $\mathbb{Z}_N$ clock model at coupling $g = 2\pi/eN$, where the gauge field coupled to the $\mathbb{Z}_N$

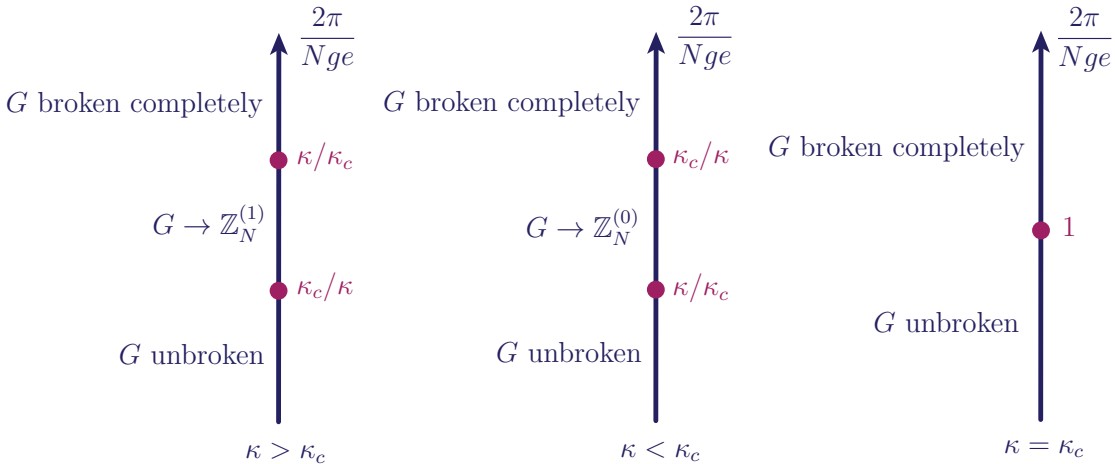

**Figure 2:** The phase diagram at $\Theta = 0$ for fixed $\kappa = e/g$ as $\mathrm{Im}(\tau) = 2\pi/Nge$ is varied. In this limit, the lattice model reduces to a decoupled $\mathbb{Z}_N$ clock model and $\mathbb{Z}_N$ gauge theory. At large $e$ and $g$, there is a trivial phase in which the clock model is disordered and the gauge theory is confining, fully preserving the global symmetry $G = \mathbb{Z}_N^{(0)} \times \mathbb{Z}_N^{(1)}$. At small $e$ and $g$, the clock model orders, and the gauge theory is deconfined (and topologically ordered), breaking the full symmetry group $G$. Depending on the value of $\kappa = e/g$, there may be an intermediate phase in which one of $\mathbb{Z}_N^{(0)}$ or $\mathbb{Z}_N^{(1)}$ is broken but not the other.

matter is locally flat. The phase structure of the gauge theory can then be determined from that of the clock model. The gauge theory also has two phases separated by a single phase transition. Under duality, the image of the disordered phase of the clock model maps to the deconfined phase of the gauge theory at small $e$. Electric wordlines condense, breaking the $\mathbb{Z}_N^{(1)}$ symmetry, which leads to topological order of the same type as the $\mathbb{Z}_N$ toric code. The TQFT describing this phase is $(2+1)$d BF theory at level $N$. (See Appendix D.2 for a review.) The image of the ordered phase of the clock model, appearing at large $e$, is the confining phase. Monopoles condense in this phase, leading to a gapped trivial vacuum that preserves the $\mathbb{Z}_N^{(1)}$ symmetry. By duality, the phase transition from this topological phase to the confining phase occurs at $e = e_c = 2\pi/Ng_c$, and the transition is in the same universality class as the clock model transition (up to global structure because of the $\mathbb{Z}_N^{(0)}$ gauging).

To set up the discussion of the phase diagram for nonzero $\Theta$, it is useful to restate the structure of the phase diagram at $\Theta = 0$ in terms of $\kappa \equiv e/g$ and $\tau \equiv \frac{\Theta}{2\pi} + i\frac{2\pi}{Nge}$ rather than $e$ and $g$. We then fix $\kappa$ and determine the phase diagram as $\tau$ is varied along the imaginary axis. For convenience, we define the critical ratio $\kappa_c$ as

$$\kappa_c \equiv \frac{e_c}{g_c} = \frac{2\pi}{Ng_c^2} = \frac{Ne_c^2}{2\pi}. \tag{2.29}$$

Since we have fixed $\kappa$, we should write $\tau$ at $\Theta = 0$ in terms of $\kappa$,

$$\tau = i\frac{2\pi}{Nge} = i\frac{2\pi}{N\kappa g^2} = i\frac{2\pi\kappa}{Ne^2}. \tag{2.30}$$

Because we also know that the spin model has a transition at $g = g_c$, we know that for fixed $\kappa$, the spin model transition at $\Theta = 0$ must occur at

$$\tau_s = i\frac{2\pi}{N\kappa g_c^2} = i\frac{\kappa_c}{\kappa}. \tag{2.31}$$

The gauge theory transition will occur at the image of $\tau_s$ under $\mathcal{S}$,

$$\tau_g = -\frac{1}{\tau_s} = i\frac{\kappa}{\kappa_c}. \tag{2.32}$$

These two transitions coincide on the imaginary $\tau$ axis only if $\kappa = \kappa_c$, in which case the simultaneous transition occurs at $\tau = i$, the fixed point of $\tau$ under $\mathcal{S}$.

We now summarize the phase structure at $\Theta = 0$ for fixed $\kappa$, as depicted in Figure 2. At large $\text{Im}(\tau) = 2\pi/Nge$, the spin model is ordered, and the gauge theory is also deconfined. For small $\text{Im}(\tau)$, the spin model is disordered, and the gauge theory confines, giving a completely trivial phase. There is a direct transition between these two phases only if $\kappa = \kappa_c$. For $\kappa > \kappa_c$, there will be a single intermediate phase where the spin model orders but the gauge theory confines. For $\kappa < \kappa_c$, the intermediate phase will have topological order while the spin model is disordered. We expect these phases and transitions to extend to a finite region for nonzero $\Theta$.

We are now ready to generalize our discussion to other phases of the lattice model. We can immediately understand the physics near $\Theta = 2\pi p$ for $p \in \mathbb{Z}$ by acting with $\mathcal{T}^p$ on the phases and transitions at $\Theta = 0$. Recall that a $\mathcal{T}$ transformation is the same as stacking a $G = \mathbb{Z}_N^{(0)} \times \mathbb{Z}_N^{(1)}$ SPT. The phases that break either the $\mathbb{Z}_N^{(0)}$ or $\mathbb{Z}_N^{(1)}$ (or both) at low energies are invariant under the $\mathcal{T}^p$ transformation because the topological term, Eq. (2.4), is trivial if $ge$ is small since monopoles and vortices are suppressed in this limit. For more details, an explicit calculation demonstrating that these phases are invariant under $\mathcal{T}$ transformations is presented in Appendix D.3.

On the other hand, the effective action for the trivial confining/disordered phase at large $ge$ vanishes. Since $\mathcal{T}^p$ stacks an SPT, the image of the trivial phase under $\mathcal{T}^p$ will be a nontrivial SPT with the classical action,

$$S_{\text{SPT}} = \frac{i\pi p}{N} \sum_{r,R} \varepsilon_{\mu\nu\lambda} A_\mu B_{\nu\lambda}. \tag{2.33}$$

In the notation for continuum field theory, this SPT action is

$$S_{\text{SPT}} = \frac{iNp}{2\pi} \int A \wedge B, \tag{2.34}$$

where $A_\mu$ and $B_{\mu\nu}$ are background $\mathbb{Z}_N$ gauge fields. The phase diagram at $\Theta = 2\pi p$ is then the same as in Figure 2, except that the trivial $G$ preserving phase at $\Theta = 0$ is replaced by an SPT phase described by Eq. (2.34). Thus, every SPT protected by $G = \mathbb{Z}_N^{(0)} \times \mathbb{Z}_N^{(1)}$ with an effective action of the form in Eq. (2.34) appears as a phase of our lattice model near integer multiples of $\Theta/2\pi$ and large $ge$. In Section 4, we will discuss the physical meaning of this SPT response in greater detail.

A particularly interesting state arises if we act on the SPT phase, Eq. (2.34), with an $\mathcal{S}$ transformation. The SPT phase, which appears are large $ge$ and $\Theta = 2\pi p$, maps to a phase at large $ge$ and $\Theta = -2\pi/p$. Since this phase may be obtained by acting with $\mathcal{S}$ on the SPT at $\Theta = 2\pi p$, the effective theory at $\Theta = -2\pi/p$ can be obtained by gauging the SPT, Eq. (2.33). Making the $\mathbb{Z}_N$ gauge fields in Eq. (2.33) dynamical, we arrive at

$$S = \frac{\pi i p}{N} \sum_{r,R} \varepsilon_{\mu\nu\lambda}\, c_\mu\, b_{\nu\lambda} + \frac{\pi i}{N} \sum_{r,R} \varepsilon_{\mu\nu\lambda}\, b_{\mu\nu}\, A_\lambda + \frac{\pi i}{N} \sum_{r,R} \varepsilon_{\mu\nu\lambda}\, c_\mu\, B_{\nu\lambda}, \tag{2.35}$$

where $c_\mu \in \mathbb{Z}$ and $b_{\mu\nu} \in \mathbb{Z}$ are dynamical while $A_\mu \in \mathbb{Z}$ and $B_{\mu\nu} \in \mathbb{Z}$ are background variables. Recall that all these fields are locally flat. To make the constraint of local flatness explicit, we introduce Lagrange multipliers $\tilde{k}, k_\mu, \tilde{\phi}, \tilde{a}_\mu \in \mathbb{Z}$, giving an action of

$$S = \frac{\pi i p}{N} \sum_{r,R} \varepsilon_{\mu\nu\lambda}\, c_\mu\, b_{\nu\lambda} + \frac{\pi i}{N} \sum_{r,R} \varepsilon_{\mu\nu\lambda}\, b_{\mu\nu}\, (\Delta_\lambda \tilde{k} + A_\lambda) + \frac{\pi i}{N} \sum_{r,R} \varepsilon_{\mu\nu\lambda}\, c_\mu\, (\Delta_\nu a_\lambda - \Delta_\lambda a_\nu + B_{\nu\lambda})$$
$$+ \frac{\pi i}{N} \sum_{r,R} \varepsilon_{\mu\nu\lambda}\, \tilde{\phi}\, \Delta_\mu B_{\nu\lambda} + \frac{\pi i}{N} \sum_{r,R} \varepsilon_{\mu\nu\lambda}\, \tilde{a}_\mu\, \Delta_\nu A_\lambda. \tag{2.36}$$

By an analysis similar to Appendix D, the corresponding effective field theory in the continuum is

$$S = \frac{iNp}{2\pi} \int c \wedge b + \frac{iN}{2\pi} \int c \wedge (da + B) + \frac{iN}{2\pi} \int b \wedge (d\varphi + A) + \frac{iN}{2\pi} \int \tilde{a} \wedge dA - \frac{iN}{2\pi} \int \tilde{\phi} \wedge dB, \tag{2.37}$$

where $\varphi$ is a dynamical $2\pi$ periodic scalar, $a_\mu$ and $c_\mu$ are dynamical $U(1)$ one-form gauge fields, and $b_{\mu\nu}$ is a dynamical $U(1)$ two-form gauge field. Here, $A_\mu$ and $B_{\mu\nu}$ are background $U(1)$ one-form and two-form gauge fields respectively. The $2\pi$ periodic scalar $\tilde{\phi}$ and $U(1)$ one-form gauge field $\tilde{a}_\mu$ are dynamical fields that constrain $A_\mu$ and $B_{\mu\nu}$ to be $\mathbb{Z}_N$ gauge fields.

The phase governed by the effective field theory, Eq. (2.37), may be obtained by acting with $\mathcal{ST}^p$ on the trivial phase at large $ge$ and near $\Theta = 0$, which has condensed vortices and monopoles. The $\mathcal{S}$ and $\mathcal{T}$ transformations in Eq. (2.28) show us that the condensed objects in this phase have charges of

$$\mathcal{ST}^p : \quad \begin{aligned} (q_s, q_m) &= (0,1) \mapsto (N, p), \\ (q_e, q_v) &= (0,1) \mapsto (N, p), \end{aligned} \tag{2.38}$$

so the condensed local operators are spin charge $N$ and magnetic charge $p$ while the condensed loop operators have electric charge $N$ and vorticity $p$. This class of phases is analogous to oblique confining phases of (3+1)d gauge theories in which bound states of electric and magnetic charges condense [39–41], and thus, we will refer to this type of (2+1)d phase as an oblique phase labeled by $(N, p)$. We will explore the bulk physics of these phases using the effective field theory, Eq. (2.37), in Section 3.

Using the same reasoning as above, we can then find the other phases in the phase diagram by mapping the phases near $\Theta = 0$ by a series of $\mathcal{S}$ and $\mathcal{T}$ transformations. The most generic transformation is

$$\mathcal{M} = \mathcal{T}^{n_1} \mathcal{S} \mathcal{T}^{n_2} \mathcal{S} \cdots \mathcal{T}^{n_{k-1}} \mathcal{S} \mathcal{T}^{n_k}, \tag{2.39}$$

where $k$ and each $n_j$ is a nonnegative integer. If we keep $\kappa$ finite and take $eg \to \infty$, the above transformation $\mathcal{M}$ on $\tau = 0$ will map to a rational value of $\tau = \Theta/2\pi$ determined by the finite continued fraction,

$$\frac{\Theta}{2\pi} = n_1 - \cfrac{1}{n_2 - \cfrac{1}{n_3 - \cfrac{1}{\ddots - \frac{1}{n_k}}}}. \tag{2.40}$$

We can then determine the effective field theory describing each gapped phase and the condensed operators that give rise to it by acting with the appropriate $\mathcal{S}$ and $\mathcal{T}$ transformations. For simplicity, we will focus on phases with $eg \to \infty$ and $\Theta/2\pi = -1/p$ for $p \in \mathbb{Z}$, which are described by Eq. (2.37), since the physics of other rational $\Theta/2\pi$ is qualitatively similar.

# 3  Oblique phases: Bulk effective field theory

Now that we have established the effective field theory in the oblique phases, we shall dedicate this section to investigating the physics of this field theory, Eq. (2.37). We begin with

establishing how the global symmetries of the lattice model, Eq. (2.1), as discussed in Section 2.2, act in this effective field theory, Eq. (2.37). The $\mathbb{Z}_N^{(0)}$ of the microscopic theory, Eq. (2.7), acts as

$$\varphi \to \varphi + \frac{2\pi}{N}, \tag{3.1}$$

while the $\mathbb{Z}_N^{(1)}$ symmetry acts as

$$a \to a + \frac{\eta}{N}, \tag{3.2}$$

where $\eta$ is a flat connection, $d\eta = 0$, with quantized cycles $\oint \eta \in 2\pi\mathbb{Z}$. The background field $A_\mu$ probes the $\mathbb{Z}_N$ zero-form symmetry while $B_{\mu\nu}$ is a probe for the $\mathbb{Z}_N$ one-form symmetry. If we set these background fields to zero, $A_\mu = B_{\nu\lambda} = 0$, then we have a $\mathbb{Z}_L^{(1)}$ symmetry, where $L = \gcd(N, p)$, which acts as

$$c \to c + \frac{1}{L}\,\tilde{\eta}^{(1)}, \qquad \varphi \to \varphi - \frac{p}{L}\,\tilde{\eta}^{(0)}, \tag{3.3}$$

where $\tilde{\eta}^{(1)} = d\tilde{\eta}^{(0)}$ is a locally flat connection and $\oint \tilde{\eta}^{(1)} \in 2\pi\mathbb{Z}$. Similarly, there is a $\mathbb{Z}_L^{(2)}$ global symmetry acting as

$$b \to b + \frac{1}{L}\,\bar{\eta}^{(2)}, \qquad a \to a - \frac{p}{L}\,\bar{\eta}^{(1)}, \tag{3.4}$$

where $\bar{\eta}^{(2)} = d\bar{\eta}^{(1)}$ is a locally flat connection and $\oint \tilde{\eta}^{(2)} \in 2\pi\mathbb{Z}$. However, if the background fields $A_\mu$ and $B_{\mu\nu}$ are turned on, then the $\mathbb{Z}_L^{(1)}$ and $\mathbb{Z}_L^{(2)}$ transformations change the action, Eq. (2.37), by

$$\Delta S = \frac{iN}{2\pi L} \int \left( -\frac{p}{L}\,\tilde{\eta}^{(1)} \wedge \bar{\eta}^{(2)} + \tilde{\eta}^{(1)} \wedge da + \bar{\eta}^{(2)} \wedge d\varphi \right) + \frac{iN}{2\pi L} \int \left( \bar{\eta}^{(2)} \wedge A + \tilde{\eta}^{(1)} \wedge B \right). \tag{3.5}$$

On a closed manifold, the terms that do not depend on $A_\mu$ or $B_{\mu\nu}$ evaluate to an integer multiple of $2\pi i$. The terms with background fields can be eliminated if we take

$$\tilde{\phi} \to \tilde{\phi} - \frac{1}{L}\,\tilde{\eta}^{(0)}, \qquad \tilde{a} \to \tilde{a} - \frac{1}{L}\,\bar{\eta}^{(1)}, \tag{3.6}$$

which is allowed only if $L = 1$. Hence, if $L > 1$, the $\mathbb{Z}_L^{(1)}$ and $\mathbb{Z}_N^{(1)}$ symmetries have a mixed 't Hooft anomaly, and similarly, the $\mathbb{Z}_L^{(2)}$ and $\mathbb{Z}_N^{(0)}$ have a mixed anomaly. If $L = 1$, there is no mixed anomaly.

Next, we determine the physics of an oblique phase by analyzing the allowed operators. The analysis is similar to that of other field theories in (1+1)d and (3+1)d [7]. The physical operators must be gauge invariant, so we first establish the gauge symmetries of Eq. (2.37).

For now, we turn off the background fields, setting $A_\mu = B_{\nu\lambda} = 0$. We consider gauge transformations,

$$b \to b + d\lambda, \qquad a \to a + d\chi - p\,\lambda, \qquad c \to c + d\xi, \qquad \varphi \to \varphi - p\,\xi, \tag{3.7}$$

where $\lambda_\mu$ is a $U(1)$ one-form gauge field while $\xi$ and $\chi$ are $2\pi$-periodic scalars. Under Eq. (3.7), the action in Eq. (2.37) (with $A_\mu = B_{\nu\lambda} = 0$) changes by

$$\Delta S = \frac{iN}{2\pi} \int (d\lambda \wedge d\varphi + d\xi \wedge da) - \frac{iNp}{2\pi} \int d\lambda \wedge d\xi, \tag{3.8}$$

which is an integer multiple of $2\pi$ on a closed manifold, so the partition function is gauge invariant.

We can now determine the gauge invariant operators. For a loop operator involving $a_\mu$ to be gauge invariant, we must attach a surface to form the operator,

$$W_{ab}(\Gamma, \Sigma) = \exp\left( i \oint_\Gamma a + ip \int_\Sigma b \right), \tag{3.9}$$

where $\Gamma$ is a loop in spacetime and $\Sigma$ is a surface such that $\Gamma = \partial\Sigma$. Physically, the loop $\Gamma$ is the worldline of an electric charge bound to a vortex of vorticity $p/N$, and the surface $\Sigma$ is the worldsheet swept by the branch cut of the vortex. Because $W_{ab}(\Gamma, \Sigma)$ requires attaching a topological surface. However, we can sometimes generate genuine loop operators from $W_{ab}(\Gamma, \Sigma)$. Since $b_{\mu\nu}$ is a $\mathbb{Z}_N$ gauge field (after integrating out $\varphi$), the operator $W_{ab}(\Gamma, \Sigma)^q$ is a genuine loop operator if $q$ is a multiple of $N$ since the choice of surface $\Sigma$ will not matter in this case—the surface is undetectable. We can thus form genuine loop operators from

$$W_a(\Gamma) = W_{ab}(\Gamma, \Sigma)^{N/L}, \tag{3.10}$$

where $L = \gcd(N, p)$. Similarly, we can construct a genuine local operator,

$$V(\mathcal{P}) = \exp\left( i\frac{N}{L}\varphi(\mathcal{P}) - i\frac{N}{L}\varphi(\mathcal{P}') + i\frac{Np}{L} \int_{\mathcal{P}'}^{\mathcal{P}} c \right), \tag{3.11}$$

where $\mathcal{P}$ is a point in spacetime and $\mathcal{P}'$ is some other fixed reference point (at infinity, for example). Indeed, the string attached to $V(\mathcal{P})$ is invisible to all other operators, so $V(\mathcal{P})$ constitutes a genuine local order parameter. The physical interpretation of $V(\mathcal{P})$ is that there is a bound state of a $\mathbb{Z}_N$ spin and a magnetic monopole at $\mathcal{P}$, and the line operator attached is a Dirac string of magnetic flux, which in this case is undetectable. Furthermore, we have surface and loop operators of the form

$$U(\Sigma) = \exp\left( i \oint_\Sigma b \right), \qquad W_c(\Gamma) = \exp\left( i \oint_\Gamma c \right), \tag{3.12}$$

where $\Sigma$ is a closed surface and $\Gamma$ is a loop. We must have $W_c(\Gamma)^N = U(\Sigma)^N = 1$ since $a_\mu$ and $\varphi$ turn $c_\mu$ and $b_{\mu\nu}$ into $\mathbb{Z}_N$ gauge fields. Moreover, the operator $W_c(\Gamma)^p$ is trivial since it can be opened and end on local operators, and $U(\Sigma)^p$ can similarly be opened and end on a loop. Thus, we conclude that $W_c(\Gamma)^L = U(\Sigma)^L = 1$.

The line operators $W_a(\Gamma)$ and $W_c(\Gamma)$ have correlation functions

$$\langle W_a(\Gamma)^{q_a} W_c(\Gamma')^{q_c} \rangle = \exp\left( i \frac{2\pi}{L} q_a q_c \, \Phi_{\text{link}}(\Gamma, \Gamma') \right), \tag{3.13}$$

where $\Phi_{\text{link}}(\Gamma, \Gamma')$ is the linking number of loops $\Gamma$ and $\Gamma'$. These operators represent the worldlines of particles that have trivial self-statistics but fractional mutual statistics. The topological order realized by these loop operators results from the breaking of $\mathbb{Z}_N^{(1)}$ symmetry to $\mathbb{Z}_{N/L}^{(1)}$. We then effectively obtain the topological order of a $\mathbb{Z}_N/\mathbb{Z}_{N/L} \cong \mathbb{Z}_L$ toric code.

The operators $V(\mathcal{P})$ and $U(\Sigma)$ reflect the spontaneous breaking of the $\mathbb{Z}_N^{(0)}$ symmetry to $\mathbb{Z}_{N/L}^{(0)}$. The local operator $V(\mathcal{P})$ is an order parameter, and the surface operators $U(\Sigma)$ are domain walls. When a local operator $V(\mathcal{P})$ crosses a domain wall $U(\Sigma)$, its expectation value changes by a phase $e^{2\pi i/L}$ since the domain walls interpolate between different vacua. Another point of view is to Wick rotate the action, Eq. (2.37), to Minkowski space and perform canonical quantization. Then, the equal-time canonical commutation relations lead to an equivalent local operator $V$ and an operator $U$ supported on all space. These two unitary operators obey

$$V\,U = e^{2\pi i/L} U\,V, \qquad V^L = U^L = 1, \tag{3.14}$$

which is the $\mathbb{Z}_L$ clock and shift algebra and results in $L$ degenerate ground states.

To summarize, there are $L = \gcd(N, p)$ genuine local operators generated by $V(\mathcal{P})$, $L$ genuine line operators generated by $W_a(\Gamma)$, $L$ genuine line operators generated by $W_c(\Gamma)$, and $L$ surface operators generated by $U(\Sigma)$. The spontaneous breaking of the $\mathbb{Z}_N^{(0)}$ symmetry to $\mathbb{Z}_{N/L}^{(0)}$ contributes a factor of $L$ to the ground state degeneracy. The topological order from the breaking of the one-form symmetry contributes a factor of $L^{2g_h}$ to the ground state degeneracy, where $g_h$ is the genus of the manifold on which the theory is placed. Therefore, the full ground state degeneracy is $L^{2g_h+1}$. In the special case in which $L = 1$, the ground state is nondegenerate and is generically a $G = \mathbb{Z}_N^{(0)} \times \mathbb{Z}_N^{(1)}$ SPT, whose physical properties we will discuss in more detail in Sections 4 and 6. This pattern of symmetry breaking is consistent with our analysis of mixed 't Hooft anomalies above.

Finally, we also comment on how the $\mathbb{Z}_N^{(0)}$ and $\mathbb{Z}_N^{(1)}$ symmetries are coupled. Although the topological order and zero-form symmetry breaking may seem independent, they are not

because they result from the same pattern of symmetry breaking, namely $\mathbb{Z}_N \to \mathbb{Z}_{N/L}$. This observation is true for our lattice model quite generally. Indeed, in every phase of our lattice model, Eq. (2.1), whenever the $\mathbb{Z}_N^{(0)}$ and $\mathbb{Z}_N^{(1)}$ symmetries are both broken to a nontrivial subgroup, they are necessarily broken to the *same* subgroup.

## 4 Intertwined response

In the following section, we will elaborate on another way in which the $\mathbb{Z}_N^{(0)}$ and $\mathbb{Z}_N^{(1)}$ symmetries are intertwined—through the mixed SPT response of their unbroken subgroup. We can examine response by coupling to background fields for the global symmetries. As we have seen, the $G = \mathbb{Z}_N^{(0)} \times \mathbb{Z}_N^{(1)}$ is partially broken, but there is a residual $H = \mathbb{Z}_{N/L}^{(0)} \times \mathbb{Z}_{N/L}^{(1)} \subset G$ subgroup, which we will find has SPT order. We will couple to background fields for the full $G$ symmetry since an observer in the deep UV only knows about the $G$ global symmetry and does not a priori know that this symmetry is broken to $H$. Nonetheless, with probes for the full $G$ symmetry, we can detect that the unbroken $H$ subgroup has SPT order.

To demonstrate the response more formally, it is simplest to use a lattice regularization of the effective field theory, Eq. (2.37). Recall from Eq. (2.35) that the lattice action is

$$S = \frac{\pi i p}{N} \sum_{r,\,R} \varepsilon_{\mu\nu\lambda}\, c_\mu\, b_{\nu\lambda} + \frac{\pi i}{N} \sum_{r,\,R} \varepsilon_{\mu\nu\lambda}\, b_{\mu\nu}\, A_\lambda + \frac{\pi i}{N} \sum_{r,\,R} \varepsilon_{\mu\nu\lambda}\, c_\mu\, B_{\nu\lambda}, \tag{4.1}$$

where $c_\mu$ and $b_{\mu\nu}$ are locally flat dynamical $\mathbb{Z}_N$ one-form and two-form gauge fields respectively while $A_\mu$ and $B_{\mu\nu}$ are locally flat background $\mathbb{Z}_N$ one-form and two-form gauge fields. Summing over $b_{\mu\nu}$ gives the constraint

$$p\, c_\mu = -A_\mu \qquad \mathrm{mod}\ N. \tag{4.2}$$

This constraint implies that $A_\mu$ is a linear combination of $p$ and $N$ with integer coefficients, which implies that $A_\mu$ is an integer multiple of $L = \gcd(N, p)$. We then must have

$$\frac{2\pi}{N} A_\mu \in \frac{2\pi}{N/L}\mathbb{Z}, \tag{4.3}$$

indicating that $A_\mu$ is in fact probing the subgroup $\mathbb{Z}_{N/L}^{(0)} \subset \mathbb{Z}_N^{(0)}$ of the symmetry. By similar reasoning, we can also show that $B_{\mu\nu}$ is constrained to probe $\mathbb{Z}_{N/L}^{(1)}$ subgroup.

This constraint, Eq. (4.3), is how the UV observer, who only knows to probe the full $\mathbb{Z}_N^{(0)}$ symmetry, detects the unbroken subgroup. The partition function vanishes unless the $A_\mu$ is probing the unbroken $\mathbb{Z}_{N/L}^{(0)}$. Physically, $A_\mu$ represents introducing twisted boundary

conditions (see Appendix A for a review), which is compatible with the long-ranged order only if $A_\mu$ probes the unbroken subgroup of the symmetry. This restriction on allowed configurations of $A_\mu$ probing the $\mathbb{Z}_N^{(0)}$ symmetry is analogous to the Meissner effect for a $U(1)$ background gauge field in a superconductor. In the superconducting state, which spontaneously breaks a $U(1)$ global symmetry, a background electromagnetic field probing the $U(1)$ symmetry is suppressed in regions where the superconducting order parameter takes a nonzero expectation value. Here, we see that the background field $A_\mu$ for the $\mathbb{Z}_N^{(0)}$ is suppressed unless it probes the unbroken $\mathbb{Z}_{N/L}^{(0)}$ symmetry.

To solve Eq. (4.3) for $c_\mu$, we write $A_\mu = L \widetilde{C}_\mu$, where $\widetilde{C}_\mu \in \mathbb{Z}$, and use that there always exists $k \in \mathbb{Z}$ such that $p\,k = L \bmod N$. Multiplying Eq. (4.3) by $k$, we have

$$L\, c_\mu = -kL\, \widetilde{C}_\mu \qquad \bmod N, \tag{4.4}$$

so we may write $c_\mu = -k\, \widetilde{C}_\mu = -k\, A_\mu/L$. We then have the effective action,

$$S_{\mathrm{resp}}[A, B] = -\frac{\pi i k}{NL} \sum_{r,\,R} \varepsilon_{\mu\nu\lambda}\, A_\mu\, B_{\nu\lambda}, \tag{4.5}$$

where we recall that $B_{\nu\lambda}$ is on a plaquette of the direct lattice and $A_\mu$ is on an intersecting link of the dual lattice. In continuum notation, this response is

$$S_{\mathrm{resp}}[A, B] = -\frac{iNk}{2\pi L} \int A \wedge B. \tag{4.6}$$

The coefficient of this response is an observable labeling the phase that is analogous to the Hall conductivity. Unlike the Hall conductivity, however, Eq. (4.6) cannot be understood as a response to local probes since the background fields probe discrete symmetries rather than continuous symmetries. Instead, we can interpret this global response as follows. Suppose we add a symmetry defect for the $\mathbb{Z}_N^{(1)}$ global symmetry. Specifically, we consider a configuration of the background field $B_{\mu\nu}$ that introduces magnetic flux through the $xy$-plane such that

$$\int B_{xy}\, dx\, dy = \frac{2\pi}{N/L}. \tag{4.7}$$

In this case, the response in Eq. (4.6) reduces to

$$S_{\mathrm{resp}} = -ik \int A_t\, dt, \tag{4.8}$$

which means that a static charge of $-k$ is induced for $A_\mu$. We find that magnetic flux induces a $\mathbb{Z}_{N/L}^{(0)}$ charge.

As we reviewed in Appendix A, giving a flux to $B_{\mu\nu}$ is the same as introducing a defect for the one-form global symmetry. Thus, the defects (or equivalently, the symmetry operators) for the $\mathbb{Z}_{N/L}^{(1)}$ global symmetry carry charge under the $\mathbb{Z}_{N/L}^{(0)}$ global symmetry. Similarly, domain walls for the $\mathbb{Z}_{N/L}^{(0)}$ symmetry transform nontrivially under the $\mathbb{Z}_{N/L}^{(1)}$ symmetry. This phenomenon is another manifestation of the generalized Witten effect discussed previously in Section 2.1 for the dynamical charges. We thus see that the unbroken subgroups of the zero-form and one-form symmetries have a nontrivial SPT response, demonstrating another way in which these symmetries are intertwined.

## 5 Bulk phase transitions

### 5.1 Decoupled transitions at $\Theta = 0$

Now that we have established the physics of the gapped phases of our lattice model, Eq. (2.1), as a result of exploiting the duality transformations introduced in Section 2.3, we turn to the phase transitions of our model. In this section, we discuss how duality also allows us to relate the different phase transitions in the phase diagram. First, we review the transitions at $\Theta = 0$, where the $\mathbb{Z}_N$ spin model and $\mathbb{Z}_N$ gauge theory are decoupled, and then generalize to other kinds of transitions. The transition for the $\mathbb{Z}_N$ spin model in Euclidean spacetime can be described using a field theory with action,

$$
\begin{aligned}
S_{\text{spin}}[\zeta; A_\mu] = \int d^3x \left( |D_\mu[A]\zeta|^2 + \mu|\zeta|^2 + u|\zeta|^4 + g_N \left[ e^{-i\Phi}\zeta^N + e^{i\Phi}(\zeta^*)^N \right] \right) \\
+ \frac{i}{2\pi} \int \rho \wedge (d\Phi - NA).
\end{aligned}
\tag{5.1}
$$

where $\zeta(x)$ is a dynamical complex scalar field that serves as an order parameter for the $\mathbb{Z}_N^{(0)}$ global symmetry, $A_\mu$ is a background $U(1)$ gauge field, $\Phi$ is a background $2\pi$ periodic scalar, and $D_\mu[A] = \partial_\mu - iA_\mu$ is the covariant derivative. The last term in Eq. (5.1) explicitly breaks the $U(1)$ zero-from symmetry down to $\mathbb{Z}_N$. The dynamical $U(1)$ two-form gauge field $\rho_{\mu\nu}$ is a Lagrange multiplier for the constraint $A_\mu = \partial_\mu\Phi/N$, which turns $A_\mu$ into a $\mathbb{Z}_N$ background gauge field that probes the $\mathbb{Z}_N^{(0)}$ global symmetry. We thus think of $A_\mu$ as a $\mathbb{Z}_N$ background gauge field (see Appendix A for more details). If $\mu$ is large and positive, the field theory becomes gapped and $\langle \zeta \rangle = 0$, giving the disordered phase. If $\mu$ is large and negative, the field theory is also gapped but $\langle \zeta \rangle \neq 0$ so that the $\mathbb{Z}_N^{(0)}$ global symmetry is spontaneously broken. At some critical value of $\mu$, there is a direction transition that separates these two gapped phases [56–59]. The $N = 2$ transition is in the 3d Ising universality class, and the

| Trivial | $\mathbb{Z}_N^{(0)}$ SSB |
|---|---|
| $\langle \zeta \rangle = 0$ | $\langle \zeta \rangle \neq 0$ |
| $\langle \tilde{\zeta} \rangle \neq 0$ | $\langle \tilde{\zeta} \rangle \neq 0$ |
| $\mathbb{Z}_N$ topological | $G \to 1$ |
| $\langle \zeta \rangle = 0$ | $\langle \zeta \rangle \neq 0$ |
| $\langle \tilde{\zeta} \rangle = 0$ | $\langle \tilde{\zeta} \rangle = 0$ |

**Figure 3:** The phases and transitions captured by the theory of Eq. (5.5), which has global symmetry $G = \mathbb{Z}_N^{(0)} \times \mathbb{Z}_N^{(1)}$. The field $\zeta$ is the $\mathbb{Z}_N^{(0)}$ order parameter, and $\tilde{\zeta}$ is a magnetic monopole. In our lattice model, Eq. (2.1), these phases and transitions appear in the vicinity of $\Theta = 0$, where the $\mathbb{Z}_N$ spin model is decoupled from the $\mathbb{Z}_N$ lattice gauge theory. The $\mathbb{Z}_N^{(0)}$ and $\mathbb{Z}_N^{(1)}$ symmetries can independently be broken or not in the low energy limit. All other transitions in the lattice model are images of the $\Theta = 0$ transitions under $\mathcal{S}$ and $\mathcal{T}$ transformations.

$N = 3$ case turns out to be weakly first order [56, 63, 65]. The $N \geq 4$ transition is consistent with the universality class of the $XY$ model [56, 58, 59, 62, 65, 66]. The last term in Eq. (5.1) is the prototype of a dangerous irrelevant operator—it is irrelevant at the critical point but is important in the symmetry-breaking phase.

Next, we describe the transition for the decoupled $\mathbb{Z}_N$ gauge theory at $\Theta = 0$. This transition is exchanged with the transition for the $\mathbb{Z}_N$ spin model under duality, $\mathcal{S}$. Thus, the field theory that describes the transition for the $\mathbb{Z}_N$ gauge theory is obtained by gauging the $\mathbb{Z}_N^{(0)}$ symmetry of Eq. (5.1). Under duality the action for the resulting field theory becomes

$$S_{\text{gauge}}[\tilde{\zeta}, c_\mu; B_{\mu\nu}] = \widetilde{S}_{\text{spin}}[\tilde{\zeta}; c_\mu] + \frac{iN}{2\pi} \int c \wedge B. \tag{5.2}$$

where $\tilde{\zeta}$ is a dynamical complex scalar field, $c_\mu$ is a dynamical $\mathbb{Z}_N$ one-form gauge field, and $\widetilde{S}_{\text{spin}}$ takes the same form as Eq. (5.1) but with different coefficients for the terms in the action. We have also introduced a coupling to a $\mathbb{Z}_N$ two-form background field $B_{\mu\nu}$. The field $\tilde{\zeta}$ is not gauge invariant on its own since it must be attached to a line operator of $c_\mu$. Physically, $\tilde{\zeta}$ represents a magnetic monopole, and the attached string is a Dirac string of magnetic flux.

When the monopole $\tilde{\zeta}$ condenses, the (magnetic) gauge field $c_\mu$ is Higgsed, resulting in

the confining phase. In the phase in which $\langle \tilde{\zeta} \rangle = 0$, and the effective action is

$$S_{\text{eff}} = \frac{i}{2\pi} \int \tilde{\rho} \wedge (d\tilde{\varphi} - Nc) + \frac{iN}{2\pi} \int c \wedge B, \tag{5.3}$$

where $\tilde{\rho}_{\mu\nu}$ is a dynamical $U(1)$ two-form gauge field and $\tilde{\varphi}$ is a dynamical $2\pi$ periodic scalar. Integrating out $\tilde{\varphi}$ constrains $\tilde{\rho} = -da$, where $a_\mu$ is a $U(1)$ one-form gauge field, giving an action of

$$S_{\text{BF}} = \frac{iN}{2\pi} \int c \wedge (da + B), \tag{5.4}$$

which is (2+1)d BF theory coupled to a background field that probes the electric $\mathbb{Z}_N^{(1)}$ symmetry. A topological $\mathbb{Z}_N$ gauge theory is left, giving us the deconfined phase. Since Eq. (5.1) and Eq. (5.2) are related to one another by discrete gauging, their transitions will essentially be in the same universality class, except for topological data. In particular, whether the transition is first order or continuous will be the same for each $N$. Further, the critical exponents of Eq. (5.1) and Eq. (5.2) are all the same, but the local operators allowed for Eq. (5.2) must be invariant under the $\mathbb{Z}_N^{(0)}$ gauge symmetry.

Putting Eqs. (5.1) and (5.2) together, we obtain the action,

$$S_{\Theta=0} = S_{\text{spin}}[\zeta; A_\mu] + S_{\text{gauge}}[\tilde{\zeta}, c_\mu; B_{\mu\nu}] = S_{\text{spin}}[\zeta; A_\mu] + \widetilde{S}_{\text{spin}}[\tilde{\zeta}; c_\mu] + \frac{iN}{2\pi} \int c \wedge B, \tag{5.5}$$

which captures the phases and phase transitions accessible to our lattice model, Eq. (2.13), at $\Theta = 0$ as the coupling constants $e$ and $g$ are varied. These transitions are represented in Figure 3.

If $\Theta$ is nonzero but small enough, we expect that the phase diagram in Figure 3 will still be valid for fixed $\Theta$ as $e$ and $g$ are varied. However, one may one wonder whether the *transitions* are still in the same universality class as the $\Theta = 0$ transitions (in the cases in which the $\Theta = 0$ transitions are continuous). A small, nonzero $\Theta$ could in principle lead to different universality classes or first-order transitions. Determining the fate of these transitions for small, nonzero $\Theta$ is beyond the scope of this work. However, the transitions at $\Theta = 0$ will map to transition curves at $\Theta \neq 0$ under $\mathcal{S}$ and $\mathcal{T}$ transformations. These examples are reminiscent of the duality web constructions of CFTs in (2+1)d [67–69] and (1+1)d [69, 70]. Because $\mathcal{S}$ and $\mathcal{T}$ transformations only change topological data, the images of the $\Theta = 0$ transitions under $\mathcal{S}$ and $\mathcal{T}$ will essentially be in the same universality class. In the following subsections, we consider some specific examples of transitions that occur in our lattice model by examining fields theories resulting from acting with $\mathcal{S}$ and $\mathcal{T}$ transformations on Eq. (5.5).

| $(N, p)$ | $\mathbb{Z}_N^{(0)}$ SSB |
|---|---|
| $\langle \zeta \rangle = 0$ | $\langle \zeta \rangle \neq 0$ |
| $\langle \tilde{\zeta} \rangle \neq 0$ | $\langle \tilde{\zeta} \rangle \neq 0$ |
| | |
| $\mathbb{Z}_N$ topological | Trivial |
| $\langle \zeta \rangle = 0$ | $\langle \zeta \rangle \neq 0$ |
| $\langle \tilde{\zeta} \rangle = 0$ | $\langle \tilde{\zeta} \rangle = 0$ |

**Figure 4:** The phase diagram for the field theory of Eq. (5.8). There are four phases: an $(N, p)$ oblique phase, phases in which either $\mathbb{Z}_N^{(0)}$ or $\mathbb{Z}_N^{(1)}$ is broken but the other is preserved, and a trivial phase that preserves the full symmetry $\mathbb{Z}_N^{(0)} \times \mathbb{Z}_N^{(1)}$. Here, $\zeta$ represents a monopole with unit magnetic charge, and $\tilde{\zeta}$ describes a composite operator consisting of a $\mathbb{Z}_N^{(0)}$ order parameter and a charge $p$ monopole.

## 5.2 Transitions involving oblique and trivial phases

In our first example, we act on Eq. (5.5) with $\mathcal{S}\mathcal{T}^p$, where $p \in \mathbb{Z}$, giving an action of

$$S = S_{\text{spin}}[\zeta; c_\mu] + \widetilde{S}_{\text{spin}}[\tilde{\zeta}; \tilde{c}_\mu] + \frac{iN}{2\pi} \int (\tilde{c} \wedge b + p\, c \wedge b + b \wedge A + c \wedge B) \qquad (5.6)$$

where $\zeta$ and $\tilde{\zeta}$ are dynamical complex scalars, $c_\mu$ and $\tilde{c}_\mu$ are dynamical $\mathbb{Z}_N$ one-form gauge fields, $b_{\mu\nu}$ is a dynamical $\mathbb{Z}_N$ two-form gauge field, $A_\mu$ is a background $\mathbb{Z}_N$ one-form gauge field, and $B_{\mu\nu}$ is a background $\mathbb{Z}_N$ two-form gauge field. We can simplify this action by integrating out the gauge field $b_{\mu\nu}$, giving the constraint,

$$\tilde{c}_\mu = -p\, c_\mu - A_\mu. \qquad (5.7)$$

Using this constraint and taking $\tilde{\zeta}$ to its complex conjugate (i.e., $\tilde{\zeta} \to \tilde{\zeta}^*$), we arrive at the action,

$$S = S_{\text{spin}}[\zeta; c_\mu] + \widetilde{S}_{\text{spin}}[\tilde{\zeta}; p\, c_\mu + A_\mu] + \frac{iN}{2\pi} \int c \wedge B. \qquad (5.8)$$

At $\Theta = 0$, in Eq. (5.5), $\zeta$ was the $\mathbb{Z}_N^{(0)}$ order parameter, and $\tilde{\zeta}$ was the field for a magnetic monopole. In Eq. (5.8), $\zeta$ and $\tilde{\zeta}$ represent the images of these objects under an $\mathcal{S}\mathcal{T}^p$ transformation. Specifically, $\zeta$ is now a charge 1 magnetic monopole, which must be attached to a string of $c_\mu$, and $\tilde{\zeta}$ is a composite of a $\mathbb{Z}_N^{(0)}$ order parameter and a charge $p$ monopole, as signaled by its coupling to both $c_\mu$ and $A_\mu$.

We now analyze the phases captured by Eq. (5.8). In the phase where $\langle \zeta \rangle \neq 0$ and $\langle \tilde{\zeta} \rangle = 0$, the gauge field $c_\mu$ is Higgsed, and the $\mathbb{Z}_N^{(0)}$ order parameter is uncondensed, leaving a trivial state with confinement and no symmetry breaking.

In the phase with $\langle \zeta \rangle = \langle \tilde{\zeta} \rangle = 0$, the monopoles and $\mathbb{Z}_N$ spins are uncondensed. The gauge field $c_\mu$ is still deconfined, and following the same logic as in Section 5.1, we find that this phase is described by BF theory, Eq. (5.4). Electric charges condense, and the $\mathbb{Z}_N^{(1)}$ symmetry is fully broken, resulting in topological order.

If $\langle \zeta \rangle \neq 0$ and $\langle \tilde{\zeta} \rangle \neq 0$, then $c_\mu$ is Higgsed by condensation of monopoles $\zeta$, but condensation of $\tilde{\zeta}$ results in $\mathbb{Z}_N^{(0)}$ symmetry breaking. Although $\tilde{\zeta}$ is a composite of the $\mathbb{Z}_N^{(0)}$ order parameter and charge $p$ monopoles, the simultaneous condensation of charge 1 monopoles in this phase screens the effect of the charge $p$ monopoles in $\tilde{\zeta}$.

Finally, we turn to the last phase, which has $\langle \zeta \rangle = 0$ but $\langle \tilde{\zeta} \rangle \neq 0$. The composites of $\mathbb{Z}_N$ spins and charge $p$ monopoles are condensed, leading to the $(N, p)$ oblique phase. Deep within this phase, the gauge field $p\, c_\mu + A_\mu$ obeys the constraint,

$$d\varphi + p\, c + A = 0, \tag{5.9}$$

where $\varphi$ is the phase of $\tilde{\zeta}$. We can introduce a Lagrange multiplier $b_{\mu\nu}$ for this constraint, giving an effective theory of

$$S_{\text{oblique}} = \frac{iN}{2\pi} \int b \wedge (d\varphi + p\, c + A) + \frac{iN}{2\pi} \int c \wedge (da + B), \tag{5.10}$$

where $a_\mu$ is a Lagrange multiplier that constrains $c_\mu$ to be a $\mathbb{Z}_N$ gauge field. This action matches Eq. (2.37), the effective field theory for the $(N, p)$ oblique phase.

All these phases and transitions captured by Eq. (5.8) are summarized in Figure 4. Note that to pass from the oblique phase to the trivial phase, one will generically encounter an intermediate phase in which $\mathbb{Z}_N^{(0)}$ or $\mathbb{Z}_N^{(1)}$ is broken but not the other. A direct transition between the oblique phase and the trivial phase requires fine tuning. A similar phase diagram appears in Ref. [71]. Indeed, for $N = 2$ and $p = 1$, Eq. (5.8) may be regarded as a continuum version of the lattice model in Ref. [71] that has two matter fields and a strictly imposed Gauss law.

To summarize, the field theory in Eq. (5.8) captures transitions between four gapped phases: a trivial phase, a $\mathbb{Z}_N^{(0)}$ symmetry breaking phase, a topologically ordered phase with $\mathbb{Z}_N^{(1)}$ broken, and the $(N, p)$ oblique phase. The transitions between these gapped states are driven by condensation of a magnetic monopole $\zeta$ and/or the composite operator $\tilde{\zeta}$, which represents a bound state of a $\mathbb{Z}_N^{(0)}$ order parameter and a charge $p$ monopole. To pass from the oblique phase to the trivial phase, one will generically encounter an intermediate phase in which $\mathbb{Z}_N^{(0)}$ or $\mathbb{Z}_N^{(1)}$ is broken but not the other. A direct transition between the oblique phase and the trivial phase requires fine tuning since these two phases are driven by the condensation of operators that are mutually local.

| $(N, p+1)$ | $\mathbb{Z}_N^{(0)}$ SSB |
|---|---|
| $\langle \zeta \rangle = 0$ | $\langle \zeta \rangle \neq 0$ |
| $\langle \tilde{\zeta} \rangle \neq 0$ | $\langle \tilde{\zeta} \rangle \neq 0$ |
| | |
| $\mathbb{Z}_N$ topological | $(N, p)$ |
| $\langle \zeta \rangle = 0$ | $\langle \zeta \rangle \neq 0$ |
| $\langle \tilde{\zeta} \rangle = 0$ | $\langle \tilde{\zeta} \rangle = 0$ |

**Figure 5:** The phases and transitions described by the field theory of Eq. (5.13). There are two distinct oblique phases—the $(N, p)$ phase and the $(N, p+1)$ phase. There are also intermediate phases in which either the $\mathbb{Z}_N^{(0)}$ or $\mathbb{Z}_N^{(1)}$ is broken while the other is preserved. The field $\zeta$ is a bound state of a $\mathbb{Z}_N$ spin and a charge $p$ monopole, and $\tilde{\zeta}$ is a composite of a $\mathbb{Z}_N$ spin and a charge $p+1$ monopole.

## 5.3 Transitions involving different oblique phases

Another interesting example arises from acting with $\mathcal{S}\mathcal{T}^p\mathcal{S}\mathcal{T}^{-1}$ on Eq. (5.5). The resulting field theory action is

$$
\begin{aligned}
S = {}& S_{\mathrm{spin}}[\zeta; \tilde{a}_\mu] + \widetilde{S}_{\mathrm{spin}}[\tilde{\zeta}; \tilde{c}_\mu] \\
&+ \frac{iN}{2\pi} \int \left( \tilde{b} \wedge \tilde{c} - \tilde{a} \wedge \tilde{b} + \tilde{a} \wedge b + \tilde{b} \wedge c + p\, b \wedge c + b \wedge A + c \wedge B \right),
\end{aligned}
\tag{5.11}
$$

where $\zeta$ and $\tilde{\zeta}$ are dynamical complex scalar fields, the fields $c_\mu$, $\tilde{a}_\mu$, and $\tilde{c}_\mu$ are dynamical $\mathbb{Z}_N$ one-form gauge fields, the fields $b_{\mu\nu}$ and $\tilde{b}_{\mu\nu}$ are dynamical $\mathbb{Z}_N$ two-form gauge fields, $A_\mu$ is a background $\mathbb{Z}_N$ one-form gauge field, and $B_{\mu\nu}$ is a background $\mathbb{Z}_N$ two-form gauge field. We recognize that $\tilde{b}_{\mu\nu}$ and $b_{\mu\nu}$ are Lagrange multipliers for the respective constraints,

$$
\tilde{c}_\mu = \tilde{a}_\mu - c_\mu, \qquad \tilde{a}_\mu = -p\,c - A.
\tag{5.12}
$$

Using these constraints and taking $\zeta \to \zeta^*$ and $\tilde{\zeta} \to \tilde{\zeta}^*$, we can simplify the action to

$$
S = S_{\mathrm{spin}}[\zeta; p\,c_\mu + A_\mu] + S_{\mathrm{spin}}[\tilde{\zeta}; (p+1)c_\mu + A_\mu] + \frac{iN}{2\pi} \int c \wedge B.
\tag{5.13}
$$

The $\mathcal{S}\mathcal{T}^p\mathcal{S}\mathcal{T}^{-1}$ transformation turns $\zeta$ into a composite operator of a $\mathbb{Z}_N$ spin and a charge $p$ monopole. The field $\tilde{\zeta}$ is now a bound state of a $\mathbb{Z}_N$ spin and a charge $p+1$ monopole. To form a gauge invariant operator, each of these fields must be attached to a line operator of $c_\mu$, which represents a Dirac string of magnetic flux.

The analysis of which phases and transitions are captured by Eq. (5.13) is similar to that of Sections 5.1 and 5.2, so we will summarize the results briefly. If $\zeta$ condenses and $\tilde{\zeta}$ does

not, then by the same reasoning as in Section 5.2, the $(N, p)$ oblique phase results. Similarly, if $\tilde{\zeta}$ condenses and $\zeta$ does not, then we obtain the $(N, p+1)$ oblique phase. If both $\zeta$ and $\tilde{\zeta}$ are uncondensed, then a deconfined gauge field $c_\mu$ remains, resulting in $\mathbb{Z}_N$ topological order with the $\mathbb{Z}_N^{(1)}$ global symmetry fully broken at low energies. If both $\zeta$ and $\tilde{\zeta}$ condense, then one of these fields Higgses $c_\mu$, and the other gives $\mathbb{Z}_N^{(0)}$ symmetry breaking. These phases and transitions are summarized in Figure 5.

## 6 Gapped boundary states

### 6.1 SPT boundaries: General considerations

Given that an oblique phase has SPT order and topological order, it is natural to consider possible boundary states. To make the analysis simpler, we first consider the case in which the bulk has no symmetry breaking or topological order but has only SPT order. Specifically, we consider boundary states for the SPT in Eq. (2.34). Then, we will make the background fields for the classical SPT action dynamical, which will turn the bulk into an oblique phase, and we will accordingly obtain possible boundary states for that phase.

The precise bulk action we consider on the (2+1)d spacetime manifold $X$ is

$$ S = \frac{iN}{2\pi} \int_X B \wedge d\varphi + \frac{iN}{2\pi} \int_X A \wedge da + \frac{iNp}{2\pi} \int_X A \wedge B, \qquad (6.1) $$

where $\varphi$ is a dynamical $2\pi$ periodic scalar, $a_\mu$ is a dynamical $U(1)$ one-form gauge field, $A_\mu$ is a background $U(1)$ one-form gauge field, and $B_{\mu\nu}$ is a background $U(1)$ two-form gauge field. If we integrate out $\varphi$ and $a_\mu$, these fields constrain $A_\mu$ and $B_{\mu\nu}$, respectively, to be $\mathbb{Z}_N$ gauge fields, leading to the response of Eq. (2.34). Boundary theories consistent with the SPT order must clearly have $\mathbb{Z}_N$ zero-form and one-form symmetries with a mixed anomaly between them, but we need to determine more precisely what this anomaly should be using the SPT action.

If we have a boundary $\partial X$, then the surface operator that acts with the $\mathbb{Z}_N^{(0)}$ symmetry in the (2+1)d bulk can end on $\partial X$ at a line. We denote this symmetry operator by $U_0(\Sigma, \Gamma)$ where $\Sigma$ is a surface in the bulk, and $\Gamma = \partial \Sigma$ is on the boundary $\partial X$. Similarly, the line operator $U_1(\gamma, \mathcal{P})$ that acts with the $\mathbb{Z}_N^{(1)}$ symmetry in the bulk can end at a point $\mathcal{P}$ on $\partial X$, which we take to be not along $\Gamma$. Suppose we compute a correlation function for $U_0(\Sigma, \Gamma)$ and $U_1(\gamma, \mathcal{P})$. As depicted in Figure 6, consider deforming $\gamma$, which has endpoint $\mathcal{P}$, to another curve $\gamma'$ with endpoint $\mathcal{P}'$ by crossing the endpoint through $\Gamma$. Using our interpretation of the SPT response in Section 4, we know that $U_1(\gamma, \mathcal{P})$ must carry charge $p$ under the $\mathbb{Z}_N^{(0)}$

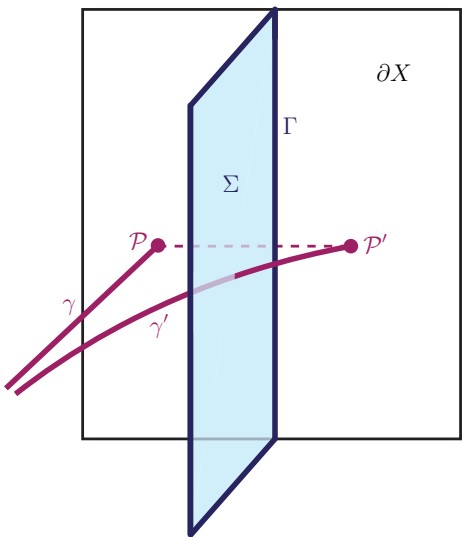

**Figure 6:** Illustration of how to detect the anomaly for the SPT protected by $\mathbb{Z}_N^{(0)} \times \mathbb{Z}_N^{(1)}$ global symmetry when the SPT is on a manifold $X$ with a boundary $\partial X$. The operator $U_0(\Sigma, \Gamma)$ that acts with the $\mathbb{Z}_N^{(0)}$ symmetry is supported on a surface $\Sigma$ in the bulk and ends at the line $\Gamma$ on $\partial X$. The $\mathbb{Z}_N^{(1)}$ symmetry operator $U_1(\gamma, \mathcal{P})$ is supported on a line $\gamma$ in the bulk that ends at a point $\mathcal{P}$ on $\partial X$. If we deform $(\gamma, \mathcal{P}) \to (\gamma', \mathcal{P}')$ using the dashed path on $\partial X$, we obtain a phase difference in correlation functions (see Eq. (6.2)).

symmetry. Correlation functions of the symmetry operators change as

$$\langle U_0(\Sigma, \Gamma) \, U_1(\gamma', \mathcal{P}')\rangle = e^{2\pi i \, p/N} \, \langle U_0(\Sigma, \Gamma) \, U_1(\gamma, \mathcal{P})\rangle \,, \tag{6.2}$$

picking up a phase factor. This expression establishes the anomaly of the bulk when a boundary is introduced.

We can cancel this anomaly by introducing a boundary theory with local operators $\mathcal{V}(\mathcal{P})$ that act with a $\mathbb{Z}_N^{(1)}$ symmetry and line operators $\mathcal{W}(\Gamma)$ that act with a $\mathbb{Z}_N^{(0)}$ symmetry. Upon canonically quantizing the boundary theory, if these operators obey

$$\mathcal{V} \, \mathcal{W} = e^{-2\pi i \, p/N} \, \mathcal{W} \mathcal{V}, \qquad \mathcal{V}^N = \mathcal{W}^N = 1, \tag{6.3}$$

then we can dress $U_0(\Sigma, \Gamma)$ with $\mathcal{W}(\Gamma)$ and decorate $U_0(\gamma, \mathcal{P})$ with $\mathcal{V}(\mathcal{P})$. For these dressed operators, the anomalous phase factors of Eqs. (6.2) and (6.3) will cancel one another so that the dressed operators are fully topological. Eq. (6.3) is the precise characterization of the mixed 't Hooft anomaly for the boundary theory. This boundary anomaly is similar to the anomaly that arises at the (2+1)d boundary of a (3+1)d $\mathbb{Z}_N$ one-form SPT [72]. As we will see in some examples, if the boundary state is gapped, the anomaly of Eq. (6.3) implies that the $\mathbb{Z}_N^{(0)}$ symmetry is spontaneously broken along $\partial X$.

## 6.2 SPT boundaries: Examples

### 6.2.1 $\mathbb{Z}_N^{(0)} \times \mathbb{Z}_N^{(0)}$ symmetry

Now that we have developed general criteria for SPT boundaries, we shall construct some concrete examples of boundary theories. If $X$ is a closed manifold, then the SPT action, Eq. (6.1), is invariant mod $2\pi i$ under gauge transformations,

$$A \to A + d\xi, \qquad B \to B + d\lambda, \qquad \varphi \to \varphi - p\,\xi, \qquad a \to a + d\chi - p\,\lambda, \qquad (6.4)$$

where $\xi$ and $\chi$ are $2\pi$ periodic scalars and $\lambda_\mu$ is a $U(1)$ one-form gauge field. However, if $X$ has a boundary, then the action changes by a boundary term,

$$\Delta S = \frac{iN}{2\pi} \int_{\partial X} \left( \lambda\, d\varphi + \xi\, da - p\,\xi\, d\lambda \right). \qquad (6.5)$$

To cancel this anomaly from the bulk, we must add extra degrees of freedom on $\partial X$. A suitable boundary action to cancel the anomaly is

$$S_{\partial X}[A, B] = \frac{iN}{2\pi} \int_{\partial X} \left( -p\,\alpha\, d\phi + \alpha\, d\tilde\phi + \tilde\alpha\, d\phi + B\,(\tilde\phi - \varphi) - A\,(\tilde\alpha - a) \right) + S_0[\Phi, A; C, B],$$
$$(6.6)$$

$$S_0[\Phi, A; C, B] = \frac{i}{2\pi} \int_{\partial X} \left[ (NB - dC)\,\bar\phi + (NA - d\Phi)\,\bar\alpha \right], \qquad (6.7)$$

where we have left the wedge products implicit for brevity. The fields $\phi$, $\tilde\phi$, and $\bar\phi$ are dynamical $2\pi$ periodic scalars while $\alpha_\mu$, $\tilde\alpha_\mu$, and $\bar\alpha_\mu$ are dynamical $U(1)$ one-form gauge fields. The fields $\phi$, $\tilde\phi$, $\alpha_\mu$, $\tilde\alpha_\mu$, and $\bar\alpha_\mu$ all live soley on the boundary $\partial X$. Here, $\Phi$ is a background $2\pi$ periodic scalar, and $C_\mu$ is a background $U(1)$ one-form gauge field. The term $S_0$ ensures that $A = d\Phi/N$ and $B = dC/N$ are one-form and two-form $\mathbb{Z}_N$ gauge fields respectively (see Appendix A). These boundary degrees of freedom transform under the gauge symmetry of Eq. (6.4) as

$$\alpha \to \alpha - \lambda, \qquad \phi \to \phi - \xi, \qquad \tilde\phi \to \tilde\phi - p\,\xi, \qquad \tilde\alpha \to \tilde\alpha - p\,\lambda, \qquad \bar\alpha \to \bar\alpha - d\chi, \qquad (6.8)$$
$$C \to C + N\lambda, \qquad \Phi \to \Phi + N\xi,$$

and $\bar\phi$ is invariant. We can simplify this boundary state by integrating by parts in the bulk so that the bulk SPT is

$$S_{\mathrm{SPT}}[A, B] = \frac{iNp}{2\pi} \int_X A \wedge B - \frac{iN}{2\pi} \int_X \varphi \wedge dB + \frac{iN}{2\pi} \int_X a \wedge dA. \qquad (6.9)$$

The boundary action then simplifies to

$$S_{\text{bdry}}[A, B] = \frac{iN}{2\pi} \int_{\partial X} \left( -p\,\alpha\,d\phi + \alpha\,d\tilde{\phi} + \tilde{\alpha}\,d\phi + B\,\tilde{\phi} - A\,\tilde{\alpha} \right) + S_0[\Phi, A; C, B]. \qquad (6.10)$$

We are now ready to analyze the physics of this state.

We begin with the symmetries of the state, Eq. (6.10), in terms of our criteria from Section 6.1. This analysis is simpler if we define a gauge field $\tilde{a}_\mu$ as as

$$\tilde{a} = \tilde{\alpha} - p\,\alpha. \qquad (6.11)$$

Then the boundary action is

$$S_{\text{bdry}}[A, B] = \frac{iN}{2\pi} \int_{\partial X} \left( \alpha\,d\tilde{\phi} + \tilde{a}\,d\phi + B\,\tilde{\phi} - A\,(\tilde{a} + p\,\alpha) \right) + S_0[\Phi, A; C, B]. \qquad (6.12)$$

The boundary theory decouples into two independent copies of (1+1)d BF theory at level $N$. While the bulk theory, Eq. (6.1), has global symmetry $G = \mathbb{Z}_N^{(0)} \times \mathbb{Z}_N^{(1)}$, the boundary theory has a global symmetry of $G_{\text{bdry}} = G \times G$. The two $\mathbb{Z}_N^{(0)}$ symmetries act as

$$\phi \to \phi + \frac{2\pi}{N}, \qquad \tilde{\phi} \to \tilde{\phi} + \frac{2\pi}{N}, \qquad (6.13)$$

and the two $\mathbb{Z}_N^{(1)}$ symmetries act as

$$\alpha \to \alpha + \frac{\eta}{N}, \qquad \tilde{a} \to \tilde{a} + \frac{\tilde{\eta}}{N}, \qquad (6.14)$$

where $\eta$ and $\tilde{\eta}$ are flat connections, $d\eta = d\tilde{\eta} = 0$, that satisfy $\oint \eta \in 2\pi\mathbb{Z}$ and $\oint \tilde{\eta} \in 2\pi\mathbb{Z}$. The boundary theory, Eq. (6.12), has local operators,

$$V_\phi(\mathcal{P}) = e^{i\phi(\mathcal{P})}, \qquad V_{\tilde{\phi}}(\mathcal{P}) = e^{i\tilde{\phi}(\mathcal{P})}, \qquad (6.15)$$

and loop operators,

$$W_\alpha(\Gamma) = \exp\left( i \oint_\Gamma \alpha \right), \qquad W_{\tilde{a}}(\Gamma) = \exp\left( i \oint_\Gamma \tilde{a} \right), \qquad (6.16)$$

that realize two copies of the $\mathbb{Z}_N$ clock and shift algebra,

$$V_{\tilde{\phi}} W_\alpha = e^{2\pi i/N} W_\alpha V_{\tilde{\phi}}, \qquad V_\phi W_{\tilde{a}} = e^{2\pi i/N} W_{\tilde{a}} V_\phi, \qquad (6.17)$$
$$(V_\phi)^N = (V_{\tilde{\phi}})^N = (W_\alpha)^N = (W_{\tilde{a}})^N = 1,$$

which can be interpreted to mean that the zero-form symmetry $\mathbb{Z}_N^{(0)} \times \mathbb{Z}_N^{(0)} \subset G_{\text{bdry}}$ is spontaneously broken on $\partial X$.

To see that this boundary state has the correct anomaly structure, Eq. (6.3), if we define the loop operator,

$$W_{\widetilde{\alpha}}(\Gamma) = \exp\left(i \oint_{\Gamma} \widetilde{\alpha}\right) = W_{\widetilde{a}}(\Gamma) W_{\alpha}(\Gamma)^p, \tag{6.18}$$

then we observe that the symmetry operators for the subgroup $G = \mathbb{Z}_N^{(0)} \times \mathbb{Z}_N^{(1)} \subset G \times G$ probed by the background fields, $A_\mu$ and $B_{\mu\nu}$, are $\mathcal{V} = V_{\widetilde{\phi}}$ and $\mathcal{W} = (W_{\widetilde{\alpha}})^{-1}$, which indeed obey Eq. (6.3), signaling the 't Hooft anomaly for the symmetries of the boundary state.

### 6.2.2 $\mathbb{Z}_{Np}^{(0)}$ symmetry

Another possible boundary state for the SPT bulk, Eq. (6.9), is given by (1+1)d BF theory at level $Np$,

$$\widetilde{S}_{\text{bdry}}[A, B] = \frac{iNp}{2\pi} \int_{\partial X} (\alpha \, d\phi + B \, \phi - A \, \alpha) + S_0[\Phi, A; C, B], \tag{6.19}$$

where $\phi$ is a dynamical $2\pi$ periodic scalar, $\alpha_\mu$ and is a dynamical $U(1)$ one-form gauge field, both of which are defined only on the boundary $\partial X$. The term $S_0$ is the same as defined in Eq. (6.7). This boundary state is distinct from Eq. (6.10), but we have reused notation for $\phi$ and $\alpha_\mu$ in Eq. (6.19) since the gauge symmetry transformations act on these fields in the same way as in Eq. (6.8).

Next, we analyze the symmetries of Eq. (6.19) and compare with our general requirements for boundary theories established in Section 6.1. This (1+1)d BF theory has symmetry $\widetilde{G}_{\text{bdry}} = \mathbb{Z}_{Np}^{(0)} \times \mathbb{Z}_{Np}^{(1)}$, which certainly contains $G = \mathbb{Z}_N^{(0)} \times \mathbb{Z}_N^{(1)}$ as a subgroup. The local operators,

$$V_\phi(\mathcal{P}) = e^{i\phi(\mathcal{P})}, \tag{6.20}$$

and the line operators,

$$W_\alpha(\Gamma) = \exp\left(i \oint_{\Gamma} \alpha\right), \tag{6.21}$$

obey the $\mathbb{Z}_{Np}$ clock and shift algebra,

$$V_\phi W_\alpha = e^{2\pi i/Np} W_\alpha V_\phi, \tag{6.22}$$

signaling the spontaneous breaking of the full $\mathbb{Z}_{Np}^{(0)}$ symmetry. The symmetry operators for the $\mathbb{Z}_N^{(0)} \times \mathbb{Z}_N^{(1)}$ subgroup of $\widetilde{G}_{\text{bdry}}$ that couples to the background fields, $A_\mu$ and $B_{\mu\nu}$, are $\mathcal{V} = (V_\phi)^p$ and $\mathcal{W} = (W_\alpha)^{-p}$, which indeed satisfy Eq. (6.3).

## 6.3 Boundary states for oblique phases

Now that we have established general criteria for boundary states of $G = \mathbb{Z}_N^{(0)} \times \mathbb{Z}_N^{(1)}$ SPTs and provided some examples of them, we can easily develop boundary states of oblique phases, which have a bulk effective field theory of Eq. (2.37). We simply need to gauge the $G$ symmetry, making the background fields that probe this symmetry dynamical. We will take this approach for both SPT boundary states introduced in Section 6.2.

### 6.3.1 Electric boundary condition

We begin with the gauged descendent of the boundary state in Section 6.2.1, replacing $A_\mu$ and $B_{\mu\nu}$ in Eq. (6.10) by dynamical $U(1)$ gauge fields, which we denote $c_\mu$ and $b_{\mu\nu}$ respectively. Correspondingly, we replace $\Phi$ and $C_\mu$ with dynamical fields $\tilde{\varphi}$ and $\tilde{c}_\mu$. The action at the boundary $\partial X$ is now

$$S_{\text{elec}} = \frac{iN}{2\pi} \int_{\partial X} \left( -p\,\alpha\,d\phi + \alpha\,d\tilde{\phi} + \tilde{\alpha}\,d\phi + b\,\tilde{\phi} - c\,\tilde{\alpha} \right) + S_0[\tilde{\varphi}, c; \tilde{c}, b], \tag{6.23}$$

where $S_0$ is the same as in Eq. (6.7) except that the all the fields in this term are now dynamical. While the SPT boundary, Eq. (6.10) has global symmetry $G \times G$, where $G = \mathbb{Z}_N^{(0)} \times \mathbb{Z}_N^{(1)}$, we have now gauged $G$, leaving a symmetry of $(G \times G)/G \simeq G$. We refer to this state as the electric boundary condition since the $G$ symmetry of the state includes the $\mathbb{Z}_N^{(1)}$ electric symmetry and the $\mathbb{Z}_N^{(0)}$ symmetry of the bulk oblique state.

To determine the physics of this state, we must identify the gauge invariant operators under the gauge transformations in Eqs. (3.7) and (6.8) (with the background fields replaced by their corresponding dynamical fields). A gauge invariant local operator is

$$e^{iN\phi(\mathcal{P}) + i\tilde{\varphi}(\mathcal{P})}, \tag{6.24}$$

where $\mathcal{P}$ is a point. However, this operator is rendered trivial by the equation of motion for $\tilde{\alpha}_\mu$. Similarly, the operator

$$\exp\left( iN \oint_\Gamma \alpha + i \oint_\Gamma \tilde{c} \right), \tag{6.25}$$

which is defined on a loop $\Gamma$, is also trivial by the equation of motion for $\tilde{\phi}$. In gauging the $G \subset G \times G$ symmetry of Eq. (6.10), we have now projected out several operators.

The nontrivial gauge invariant operators are

$$\widetilde{V}(\mathcal{P}) = e^{i\tilde{\phi}(\mathcal{P}) - ip\,\phi(\mathcal{P})}, \qquad \widetilde{W}(\Gamma) = \exp\left( i \oint_\Gamma \tilde{\alpha} - ip \oint_\Gamma \alpha \right), \tag{6.26}$$

where $\mathcal{P}$ is a point on $\partial X$ and $\Gamma$ is a loop on $\partial X$. These operators realize the symmetry $G$ projectively,

$$\widetilde{V} \, \widetilde{W} = e^{-2\pi i p/N} \, \widetilde{W} \, \widetilde{V}, \qquad (\widetilde{V})^N = (\widetilde{W})^N = 1. \tag{6.27}$$

Note that $(\widetilde{V})^{N/L}$ and $(\widetilde{W})^{N/L}$ commute with all other boundary operators. We recall from Section 3 that the $\mathbb{Z}_N^{(0)} \times \mathbb{Z}_N^{(1)}$ symmetry is broken in the bulk to the subgroup $\mathbb{Z}_{N/L}^{(0)} \times \mathbb{Z}_{N/L}^{(1)}$, where $L = \gcd(N, p)$. The operators $(\widetilde{V})^{N/L}$ and $(\widetilde{W})^{N/L}$ correspond to operators of the bulk. If we mod out by these bulk operators, Eq. (6.27) implies that the remaining $\mathbb{Z}_{N/L}^{(0)}$ global symmetry is spontaneously broken at the boundary. The $\mathbb{Z}_{N/L}^{(1)}$ symmetry is still preserved at the boundary $\partial X$ since this symmetry cannot be spontaneously broken in (1+1)d.

### 6.3.2   Magnetic boundary condition

We now apply a similar analysis to the SPT boundary state discussion in Section 6.2.2. We make the background fields of this state dynamical. The resulting boundary state for the oblique phase is

$$S_{\text{mag}} = \frac{iNp}{2\pi} \int_{\partial X} (\alpha \, d\phi + b \, \phi - c \, \alpha) + S_0[\tilde{\varphi}, c; \tilde{c}, b], \tag{6.28}$$

where $\tilde{\varphi}$ is a dynamical $2\pi$ periodic scalar, $c_\mu$ and $\tilde{c}_\mu$ are dynamical $U(1)$ one-form gauge fields, and $b_{\mu\nu}$ is a dynamical $U(1)$ two-form gauge field. The rest of the fields are the same as in Section 6.2.2, and $S_0$ is the same as in Eq. (6.7) but with dynamical fields.

Recall that the original global symmetry of the SPT state, Eq. (6.19), was $\mathbb{Z}_{Np}^{(0)} \times \mathbb{Z}_{Np}^{(1)}$. Now that we have gauged the $\mathbb{Z}_N^{(0)} \times \mathbb{Z}_N^{(1)}$ symmetry, the symmetry group of the resulting state, Eq. (6.28) is $\mathbb{Z}_p^{(0)} \times \mathbb{Z}_p^{(1)}$. Since this boundary state does not contain the symmetry $\mathbb{Z}_{N/L}^{(0)} \times \mathbb{Z}_{N/L}^{(1)}$ as a subgroup, it is not a boundary condition that is consistent with the SPT order for this symmetry discussed in Section 4. Nonetheless, it is a valid boundary state of the effective field theory, Eq. (2.37), for this bulk oblique phase, so a natural question is how to understand the physical meaning of the $\mathbb{Z}_p^{(0)} \times \mathbb{Z}_p^{(1)}$ symmetry. In our microscopic lattice model, Eq. (2.1), we have condensed charge $N$ spin sources and charge $N$ electric matter so that we are working with $\mathbb{Z}_N$ variables. Within the $(N, p)$ oblique phase, the charge $N$ spins become bound states with charge $p$ magnetic monopoles, and the charge $N$ electric matter bound to vortices of vorticity $p$ condense. The same physics can arise from first condensing the charge $p$ magnetic variables and then forming bound state with charge $N$ electric variables. In this case, there is a $\mathbb{Z}_p^{(0)} \times \mathbb{Z}_p^{(1)}$ that is broken to $\mathbb{Z}_{p/L}^{(0)} \times \mathbb{Z}_{p/L}^{(1)}$ in the bulk oblique phase. The boundary state, Eq. (6.28), preserves this $\mathbb{Z}_p^{(0)} \times \mathbb{Z}_p^{(1)}$ symmetry, so we refer to this boundary state as the magnetic boundary conditon.

We now examine the physics of this state by determining the gauge invariant operators. Like in Section 6.3.1, some of the operators of the theory in Section 6.2.2 are now projected out. These trivialized operators are

$$e^{i\,\varphi(\mathcal{P})-ip\,\phi(\mathcal{P})}, \qquad \exp\left(i\oint_{\Gamma} a - ip\oint_{\Gamma}\alpha\right), \tag{6.29}$$

where $\mathcal{P}$ is a point and $\Gamma$ is a loop, and we recall that $\varphi$ and $a_\mu$ are fields from the bulk oblique phase, Eq. (2.37). The nontrivial genuine local operators and loop operators are respectively,

$$\widetilde{\mathcal{V}}(\mathcal{P}) = e^{i\,\tilde{\varphi}(\mathcal{P})+iN\,\phi(\mathcal{P})}, \qquad \widetilde{\mathcal{W}}(\Gamma) = \exp\left(i\oint_{\Gamma}\tilde{\alpha} + iN\oint_{\Gamma}\alpha\right). \tag{6.30}$$

These boundary operators obey

$$\widetilde{\mathcal{V}}\,\widetilde{\mathcal{W}} = e^{2\pi i\,N/p}\,\widetilde{\mathcal{W}}\,\widetilde{\mathcal{V}}, \qquad (\widetilde{\mathcal{V}})^p = (\widetilde{\mathcal{W}})^p = 1, \tag{6.31}$$

which implies that the $\mathbb{Z}_p^{(0)}$ magnetic global symmetry is spontaneously broken. Similar to our analysis in Section 6.3.1, if we mod out by the bulk operators, we find that a $\mathbb{Z}_{p/L}^{(0)}$ symmetry is spontaneously broken at the boundary.

# 7  Oblique phases: Hamiltonian lattice model

Having discussed both the bulk and boundary physics of an oblique phase, to provide a complementary perspective, we develop a class of Hamiltonian lattice models whose ground states are oblique phases. We will then reiterate some of the previously discussed physics in Sections 3 and 6 but now using the Hamiltonian formalism. We work on a two-dimensional square lattice with sites $r$. Each link $\ell$ of the lattice is labeled by a site $r$ and a spatial direction $j \in \{x, y\}$. We place $\mathbb{Z}_N$ gauge field operators, $\tau_j^z(r)$ and $\tau_j^x(r)$, on links. These unitary operators obey

$$[\tau_j^x(r)]^N = [\tau_j^z(r)]^N = 1, \qquad \tau_j^z(r)\,\tau_{j'}^x(r') = \omega^{\delta(\ell-\ell')}\,\tau_{j'}^x(r')\,\tau_j^z(r), \tag{7.1}$$

where $\omega = e^{2\pi i/N}$ and $\delta(\ell - \ell') = \delta(r - r')\,\delta(j - j')$ is the Kronecker delta for links. We additionally introduce $\mathbb{Z}_N$ matter operators $\sigma^z(R)$ and $\sigma^x(R)$ on sites $R$ of the dual lattice (at the centers of plaquettes of the direct lattice). Similarly, these operators are unitary and satisfy

$$[\sigma^x(R)]^N = [\sigma^z(R)]^N = 1, \qquad \sigma^z(R)\,\sigma^x(R') = \omega^{\delta(R-R')}\,\sigma^x(R')\,\sigma^z(R), \tag{7.2}$$

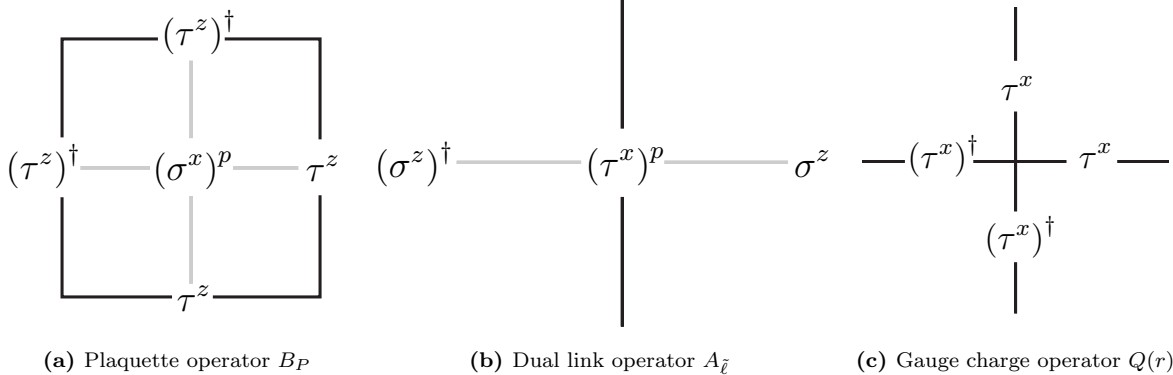

**(a)** Plaquette operator $B_P$  **(b)** Dual link operator $A_{\tilde{\ell}}$  **(c)** Gauge charge operator $Q(r)$

**Figure 7:** Depictions of (7a) the plaquette operator $B_P$, (7b) the dual link operator $A_{\tilde{\ell}}$, and (7c) the gauge charge operator $Q(r)$. The darker lines indicate links of the lattice while the lighter lines are links of the dual lattice.

where $\delta(R - R')$ is the Kronecker delta for two dual sites $R, R'$.

Before presenting the Hamiltonian, it is useful to define the following operators, which are also shown pictorially in Figure 7. For a plaquette $P$ with a lower left corner at lattice site $r$, the plaquette operator $B_P$ is defined as

$$B_P = \tau_x^z(r)\,\tau_y^z(r + e_x)\,\tau_x^z(r + e_y)^\dagger\,\tau_y^z(r)^\dagger\,[\sigma^x(r + (e_x + e_y)/2)]^p, \tag{7.3}$$

where $p \in \mathbb{Z}$. Next, for a dual link $\tilde{\ell}$ oriented in the positive $e_j$ direction (where $j \in \{x, y\}$), the dual link operator $A_{\tilde{\ell}}$ is

$$A_{\tilde{\ell}} = \begin{cases} \sigma^z(R)\,\sigma^z(R - e_x)^\dagger\,[\tau_y^x(R - (e_x + e_y)/2)]^p & \text{for } e_j = e_x, \\ \sigma^z(R)\,\sigma^z(R - e_y)^\dagger\,[\tau_x^x(R - (e_x + e_y)/2)]^p & \text{for } e_j = e_y. \end{cases} \tag{7.4}$$

Here, $R$ is defined as the rightmost endpoint of the dual link if $\tilde{\ell}$ is oriented in the positive $x$-direction, and $R$ is the uppermost end point of the dual link if $\tilde{\ell}$ is oriented in the positive $y$-direction. Finally, we define the $\mathbb{Z}_N$ gauge charge operator $Q(r)$ at a lattice site $r$ as

$$Q(r) = \tau_x^x(r)\,\tau_x^x(r - e_x)^\dagger\,\tau_y^x(r)\,\tau_y^x(r - e_y)^\dagger. \tag{7.5}$$

Using these operators, we are now prepared to describe the Hamiltonian.

The Hamiltonian whose ground state realizes an oblique phase is

$$H = -\sum_P \left(B_P + B_P^\dagger\right) - \sum_{\tilde{\ell}} \left(A_{\tilde{\ell}} + A_{\tilde{\ell}}^\dagger\right). \tag{7.6}$$

A Hamiltonian of this type was first studied in the $(N, p) = (2, 1)$ case on the triangular lattice by Yoshida [37]. In that example, the ground state is a $\mathbb{Z}_2^{(0)} \times \mathbb{Z}_2^{(1)}$ SPT. Here, we will

analyze this Hamiltonian for generic $(N, p)$ on the square lattice. This Hamiltonian is also a natural generalization of similar models in (1+1)d [73, 74].

Since we are working with a gauge theory, the gauge charge $Q(r)$ and the Hamiltonian $H$ commute,

$$[H, Q(r)] = 0. \tag{7.7}$$

Any state $|\Psi\rangle$ in the physical Hilbert space is constrained to obey Gauss's law,

$$Q(r) |\Psi\rangle = |\Psi\rangle, \tag{7.8}$$

for every site $r$ on the lattice.

The Hamiltonian also has two global symmetries—a $\mathbb{Z}_N^{(0)}$ zero-form symmetry and a $\mathbb{Z}_N^{(1)}$ one-form symmetry. The respective symmetry operators that commute with the Hamiltonian are

$$U = \prod_R \sigma^x(R), \qquad T(\widetilde{\Gamma}) = \prod_{(r,j) \in \widetilde{\Gamma}} \tau_j^x(r). \tag{7.9}$$

The $\mathbb{Z}_N^{(0)}$ symmetry operator $U$ is a product over all sites $R$ on the dual lattice, and the $\mathbb{Z}_N^{(1)}$ symmetry operator, an 't Hooft loop $T(\widetilde{\Gamma})$, is a product over all links intersecting a noncontractible loop $\widetilde{\Gamma}$ on the dual lattice. These operators are the lattice analogues of the symmetry operators, Eq. (3.12), of the field theory in Section 3. The order parameter for the $\mathbb{Z}_N^{(0)}$ symmetry is $\sigma^z(R)$ since it transforms nontrivially under this symmetry,

$$U^{-1} \sigma^z(R) U = \omega \sigma^z(R). \tag{7.10}$$

Likewise, the Wilson loop operator $W(\Gamma)$,

$$W(\Gamma) = \prod_{(r,j) \in \Gamma} \tau_j^z(r), \tag{7.11}$$

for a noncontractible loop $\Gamma$, is the operator that transforms nontrivially under the one-form symmetry,

$$T(\widetilde{\Gamma})^{-1} W(\Gamma) T(\widetilde{\Gamma}) = \omega W(\Gamma), \tag{7.12}$$

where $\Gamma$ and $\widetilde{\Gamma}$ are noncontractible loops that intersect once. The lattice Hamiltonian, Eq. (7.6), indeed has the global symmetry $G = \mathbb{Z}_N^{(0)} \times \mathbb{Z}_N^{(1)}$.

Next, we determine the fate of this global symmetry $G$ at low energies. Each term of the Hamiltonian, Eq. (7.6), commutes with every other term, so a ground state $|\psi_0\rangle$ must satisfy

$$Q(r) |\psi_0\rangle = B_P |\psi_0\rangle = B_P^\dagger |\psi_0\rangle = A_{\widetilde{\ell}} |\psi_0\rangle = A_\ell^\dagger |\psi_0\rangle = |\psi_0\rangle \tag{7.13}$$

for all sites $r$, dual links $\tilde{\ell}$, and plaquettes $P$. From these conditions, we can demonstrate that the global symmetry $G$ is broken to the subgroup $\mathbb{Z}_{N/L}^{(0)} \times \mathbb{Z}_{N/L}^{(1)}$, where $L = \gcd(N, p)$, at low energies. To determine whether the $\mathbb{Z}_N^{(0)}$ symmetry is spontaneously broken, we examine the two-point functions for the order parameter. A subset of these correlation functions can be expressed as string of $(A_{\tilde{\ell}})^M$ operators for some integer $M$. If $M$ is an integer multiple of $N/L$, then such a string operator can be related to a product of local operators at the endpoints of the string. Indeed, because $Np/L$ is a multiple of $N$, the string connecting the endpoints is trivial. This statement in mathematical form is

$$\langle \psi_0 | \, \sigma^z(R)^{N/L} \left( \sigma^z(R')^\dagger \right)^{N/L} | \psi_0 \rangle = \langle \psi_0 | \prod_{\tilde{\ell} \in \tilde{\gamma}} (A_{\tilde{\ell}})^{N/L} | \psi_0 \rangle = 1, \qquad (7.14)$$

where the product is over all dual links $\tilde{\ell}$ in a curve $\tilde{\gamma}$ in the dual lattice from $R'$ to $R$. For simplicity, we have assumed without loss of generality that $R = R' + k_x \, e_x + k_y \, e_y$, where $k_x$ and $k_y$ are positive integers. We have also used the constraint in Eq. (7.13) that $A_{\tilde{\ell}}$ leaves $|\psi_0\rangle$ invariant for any dual link $\tilde{\ell}$. This correlation function, Eq. (7.14), is the analogue of Eq. (3.11) in the field theory of Section 3. Its nonzero expectation value implies long-ranged order in $\sigma^z(R)^{N/L}$, so the $\mathbb{Z}_N^{(0)}$ symmetry is spontaneously broken to $\mathbb{Z}_{N/L}^{(0)}$.

We now similarly analyze the $\mathbb{Z}_N^{(1)}$ global symmetry by examining the expectation value of a contractible Wilson loop,

$$W(\gamma) = \prod_{(r,j) \in \gamma_+} \tau_j^z(r) \prod_{(r,j) \in \gamma_-} \tau_j^z(r)^\dagger. \qquad (7.15)$$

In the above notation, we mean that $W(\gamma)$ is a product over oriented links in a single contractible loop $\gamma = \gamma_+ \cup \gamma_-$. The curve $\gamma$ can be divided into a set of links $\gamma_+$ that are oriented in the $+e_x$ or $+e_y$ direction and a distinct set of links $\gamma_-$ that point in the $-e_x$ or $-e_y$ direction. In our Wilson loop operator $W(\gamma)$, we use $\tau_j^z(r)$ on every link in $\gamma_+$ and $\tau_j^z(r)^\dagger$ on every link in $\gamma_-$. For a ground state $|\psi_0\rangle$, we have

$$\langle \psi_0 | \, W(\gamma)^{N/L} | \psi_0 \rangle = \langle \psi_0 | \prod_{P \in \Sigma} (B_P)^{N/L} | \psi_0 \rangle = 1, \qquad (7.16)$$

where the product is over plaquettes $P$ of a surface $\Sigma$ bounded by $\gamma = \partial\Sigma$. Here, we have used the constraint in Eq. (7.13) that $B_P$ leaves the ground state invariant for any plaquette $P$. This Wilson loop $W(\gamma)^{N/L}$ is an analogue of the genuine loop operator in Eq. (3.10). Eq. (7.16) implies that the $\mathbb{Z}_N^{(1)}$ symmetry is broken to $\mathbb{Z}_{N/L}^{(1)}$ since $W(\Gamma)^{N/L}$ is deconfined.

There is another way of demonstrating that the $\mathbb{Z}_N^{(0)} \times \mathbb{Z}_N^{(1)}$ global symmetry is broken to $\mathbb{Z}_{N/L}^{(0)} \times \mathbb{Z}_{N/L}^{(1)}$. Note that the Hamiltonian, Eq. (7.6), commutes with the unitary operators

$$\sigma^z(R)^{N/L}, \qquad W(\Gamma)^{N/L}, \qquad (7.17)$$

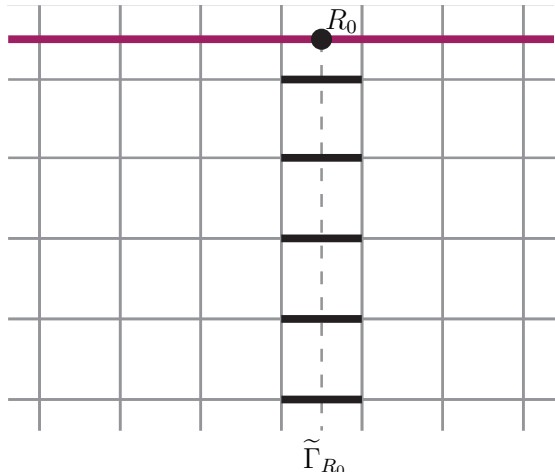

**Figure 8:** Action of the symmetry operators when there is a lattice boundary. The 't Hooft line $T(\widetilde{\Gamma}_{R_0})^L$ along $\widetilde{\Gamma}_{R_0}$, depicted by the dashed line, ends at a point $R_0$ on the boundary. The symmetry operator $U^L$ for the zero-form symmetry acts on the boundary along the red line, which intersects $R_0$, resulting in the projective algebra of Eq. (7.23). We take the boundary Hamiltonian to have link terms $A_{\tilde{\ell}}$ but not plaquette terms $B_P$.

where $\Gamma$ is a noncontractible loop and $W(\Gamma)$ is defined in Eq. (7.11). The operator $\sigma^z(R)^{N/L}$ acts with a $\mathbb{Z}_L$ two-form symmetry on $U$, and the Wilson loop $W(\Gamma)^{N/L}$ acts with a $\mathbb{Z}_L$ one-form symmetry on $T(\widetilde{\Gamma})$. The symmetry operators $\sigma^z(R)^{N/L}$ and $U$ obey the clock and shift algebra,

$$\sigma^z(R)^{N/L}\, U = e^{2\pi i/L}\, U\, \sigma^z(R)^{N/L}, \tag{7.18}$$

which implies a ground state degeneracy associated with the symmetry breaking $\mathbb{Z}_N^{(0)} \to \mathbb{Z}_{N/L}^{(0)}$. To diagnose the one-form symmetry breaking, we place the lattice model on a torus and consider Wilson loops $W(\Gamma_j)$ and 't Hooft loops $T(\widetilde{\Gamma}_j)$, where $\Gamma_j$ and $\widetilde{\Gamma}_j$ are oriented loops on the direct and dual lattices, respectively, in the $e_j$ direction (with $j \in \{x, y\}$). These operators also obey a clock and shift algebra,

$$W(\Gamma_x)^{N/L}\, T(\widetilde{\Gamma}_y) = e^{2\pi i/L}\, T(\widetilde{\Gamma}_y)\, W(\Gamma_x)^{N/L},$$
$$W(\Gamma_y)^{N/L}\, T(\widetilde{\Gamma}_x) = e^{2\pi i/L}\, T(\widetilde{\Gamma}_x)\, W(\Gamma_y)^{N/L}, \tag{7.19}$$

so we conclude that there is topological order due to one-form symmetry breaking, $\mathbb{Z}_N^{(1)} \to \mathbb{Z}_{N/L}^{(1)}$. The full ground state degeneracy on the torus is then $L^3$, where a factor $L^2$ is from topological order and a factor $L$ is from zero-form symmetry breaking.

Finally, we diagnose the SPT order for the remaining unbroken subgroup $\mathbb{Z}_{N/L}^{(0)} \times \mathbb{Z}_{N/L}^{(1)}$ by introducing a boundary. As illustrated in Figure 8, we choose the boundary so that

the lattice model is in a half-space extending into the $-e_y$ direction, and the vacuum is in the other half-space extending in the $+e_y$ direction. We take the boundary Hamiltonian to have link terms $A_{\tilde{\ell}}$ but not plaquette terms $B_P$. The symmetry operators for the unbroken symmetries are

$$U^L = \prod_R \sigma^x(R)^L, \qquad T(\widetilde{\Gamma})^L = \prod_{(r,j)\in\tilde{\Gamma}} \tau_j^x(r)^L. \tag{7.20}$$

We can write $k\,p = L \bmod N$ for some integer $k$ because $L = \gcd(N, p)$. Consider a line $\widetilde{\Gamma}_{R_0}$, depicted in Figure 8, that ends on the system boundary at point $R_0$ and extends infinitely into the bulk. Because $A_{\tilde{\ell}}$ leaves the ground state invariant for each dual link $\tilde{\ell}$, for each link $\ell = (r, j)$ intersecting $\widetilde{\Gamma}_{R_0}$ we have

$$\tau_j^x(r)^L \,|\psi_0\rangle = \tau_j^x(r)^{kp}\,|\psi_0\rangle = \left[\sigma^z(R)^\dagger \sigma^z(R')\right]^k |\psi_0\rangle, \tag{7.21}$$

where $R$ and $R'$ are the endpoints of the dual link intersecting $\ell$. The 't Hooft operator $T(\widetilde{\Gamma}_{R_0})^L$ then acts on the ground state $|\psi_0\rangle$ as

$$T(\widetilde{\Gamma}_{R_0})^L \,|\psi_0\rangle = \left[\sigma^z(R_0)^\dagger\right]^k |\psi_0\rangle. \tag{7.22}$$

Therefore, the one-form symmetry operator $T(\widetilde{\Gamma}_{R_0})^L$ now has nontrivial commutation relations with $U^L$ at the boundary. We find a projective representation,

$$T(\widetilde{\Gamma}_{R_0})^L \, U^L |\psi_0\rangle = e^{-2\pi i\,kL/N} \, U^L \, T(\widetilde{\Gamma}_{R_0})^L \,|\psi_0\rangle, \tag{7.23}$$

which implies that there is a mixed 't Hooft anomaly for the unbroken $\mathbb{Z}_{N/L}^{(0)} \times \mathbb{Z}_{N/L}^{(1)}$ symmetry, and hence, there is SPT order for this symmetry. This $\mathbb{Z}_{N/L}$ clock and shift algebra, Eq. (7.23), implies that the state has $\mathbb{Z}_{N/L}^{(0)}$ spontaneous symmetry breaking at the boundary in addition to the symmetry breaking in the bulk. Hence, the boundary condition is the analogue of the state studied in Section 6.3.1. To make this connection more direct, we note that the commutation relations of $T(\widetilde{\Gamma}_{R_0})^p$ and $U^p$ match those of $\widetilde{V}$ and $\widetilde{W}$ in Eq. (6.27).

To summarize, the low energy physics of the Hamiltonian, Eq. (7.6), is that of an oblique phase. There is a $\mathbb{Z}_N^{(0)} \times \mathbb{Z}_N^{(1)}$ global symmetry, which is broken in the bulk to a possibly nontrivial subgroup $\mathbb{Z}_{N/L}^{(0)} \times \mathbb{Z}_{N/L}^{(1)}$, where $L = \gcd(N, p)$. The residual subgroup symmetry $\mathbb{Z}_{N/L}^{(0)} \times \mathbb{Z}_{N/L}^{(1)}$ has SPT order. This lattice Hamiltonian approach provides a complementary perspective to our results derived using the lattice partition function and the effective field theory in previous sections.

# 8    SPT boundary criticality

The boundary states we have discussed thus far are all gapped and spontaneously break a zero-form global symmetry. Because a one-form symmetry cannot be spontaneously broken in (1+1)d, a boundary state of a (2+1)d $\mathbb{Z}_N^{(0)} \times \mathbb{Z}_N^{(1)}$ SPT must either spontaneously break the $\mathbb{Z}_N^{(0)}$ symmetry or be gapless. In this section, we will present a gapless boundary state for a generic $\mathbb{Z}_N^{(0)} \times \mathbb{Z}_N^{(1)}$ SPT. We will then examine symmetry-allowed perturbations that can be added to this gapless boundary state and lead to a gapped phase when these perturbations are relevant. In these cases, the gapless phase can be interpreted as a critical point (or critical line) between distinct gapped boundary states of the SPT.

We consider a generic $\mathbb{Z}_N^{(0)} \times \mathbb{Z}_N^{(1)}$ SPT, Eq. (6.9), on an open manifold $X$. The action along the boundary $\partial X$ is

$$
S = \int_{\partial X} d^2 x \left[ \frac{K}{2} \left( \partial_x \phi + A_x \right)^2 + \frac{\widetilde{K}}{2} (\partial_\mu \tilde{\phi} + p A_\mu)^2 + \frac{1}{4 e_\alpha^2} \left( f_{\mu\nu}^{(\alpha)} + B_{\mu\nu} \right)^2 + \frac{1}{2(2\pi)^2 K} \left( \partial_x \vartheta \right)^2 \right.
$$
$$
\left. + \frac{i}{2\pi} (\partial_t \phi + A_t)(\partial_x \vartheta) \right] + \frac{iN}{2\pi} \int_{\partial X} \left( \alpha \, d\tilde{\phi} + B \, \tilde{\phi} - p \, A \, \alpha \right) + S_0[\Phi, A; C, B],
$$

$$(8.1)$$

where $\phi$, $\tilde{\phi}$, and $\vartheta$ are dynamical $2\pi$ periodic scalars, $\alpha_\mu$ is a dynamical $U(1)$ gauge field, and $f_{\mu\nu}^{(\alpha)} = \partial_\mu \alpha_\nu - \partial_\nu \alpha_\mu$ is the field strength of $\alpha_\mu$. We have also introduced the Luttinger parameters $K$ and $\widetilde{K}$ for $\phi$ and $\tilde{\phi}$ respectively, and the scalar $\vartheta$ is the dual field of $\phi$. The term $S_0$ is the same as defined in Eq. (6.7). Under the gauge symmetry, Eq. (6.4), the boundary fields of Eq. (8.1) change in the same way as in Eq. (6.8), and the newly introduced field $\vartheta$ is invariant. If the boundary fields of Eq. (8.1) transform in this way, then the bulk action, Eq. (6.9), and the boundary action, Eq. (8.1), are together gauge invariant.

We begin by discussing the global symmetries of the boundary theory, Eq. (8.1). There is a $U(1)$ momentum zero-form global symmetry, denoted $U(1)_m^{(0)}$, which acts on $\phi$ as

$$
\phi \to \phi + c_m, \tag{8.2}
$$

where $c_m \in \mathbb{R}$ is a constant. A $U(1)$ winding zero-form symmetry, denoted $U(1)_w^{(0)}$, acts on $\vartheta$ as

$$
\vartheta \to \vartheta + c_w, \tag{8.3}
$$

where $c_w \in \mathbb{R}$ is a constant. A $\mathbb{Z}_N^{(0)}$ global symmetry acts on $\tilde{\phi}$ as

$$
\tilde{\phi} \to \tilde{\phi} + \frac{2\pi}{N}. \tag{8.4}
$$

Finally, we have a $\mathbb{Z}_N^{(1)}$ global symmetry,

$$\alpha \to \alpha + \frac{\eta}{N}, \tag{8.5}$$

where $\eta$ is a flat connection, $d\eta = 0$, with quantized cycles $\oint \eta \in 2\pi\mathbb{Z}$. The one-form background field $A_\mu$ probes the $\mathbb{Z}_N^{(0)}$ diagonal subgroup of the $U(1)^{(0)} \times \mathbb{Z}_N^{(0)}$ that acts on $\phi$ and $\tilde{\phi}$ as

$$\phi \to \phi + \frac{2\pi}{N}, \qquad \tilde{\phi} \to \tilde{\phi} + \frac{2\pi p}{N}, \tag{8.6}$$

and the two-form background field $B_{\mu\nu}$ probes the one-form symmetry of Eq. (8.5).

This boundary theory has a gapped sector and a gapless sector, which can be seen by integrating out $\alpha_\mu$ and $\vartheta$, giving an effective action of

$$S_{\text{eff}} = \int_{\partial X} d^2x \left[ \frac{K}{2}(\partial_\mu\phi + A_\mu)^2 + \frac{\widetilde{K}}{2}(\partial_\mu\tilde{\phi} + p\, A_\mu)^2 + \frac{1}{2}\left(\frac{Ne_\alpha}{2\pi}\right)^2 \min_{z \in \mathbb{Z}} \left(\tilde{\phi} + \frac{p\,\Phi}{N} - \frac{2\pi z}{N}\right)^2 \right]$$
$$+ \frac{iN}{2\pi} \int_{\partial X} B\,\tilde{\phi} + S_0[\Phi, A; C, B]. \tag{8.7}$$

The gauge field induces a mass term for $\tilde{\phi}$, as in the massless Schwinger model [75]. In the extreme IR limit, $e_\alpha^2 \to \infty$, we obtain the constraint,

$$\tilde{\phi} + \frac{p\,\Phi}{N} = \frac{2\pi z}{N}, \qquad z \in \mathbb{Z}, \tag{8.8}$$

After using this constraint, the effective action that remains is

$$S_{\text{eff}} = \int_{\partial X} d^2x \, \frac{K}{2}(\partial_\mu\phi + A_\mu)^2 + \frac{i}{2\pi} \int_{\partial X} B\,(2\pi z - p\,\Phi) + S_0[\Phi, A; C, B]. \tag{8.9}$$

If we turn off the background fields, this action represents a single compact boson with Luttinger parameter $K$. The boundary theory thus consists of a decoupled compact boson and a gapped sector with $\mathbb{Z}_N^{(0)}$ symmetry breaking. Although these sectors are decoupled here, because the background field $A_\mu$ couples to both the gapped and gapless sectors, they will not necessarily be decoupled in gapped phases when we add perturbations.

We now determine the fate of the gapless boundary theory after adding perturbations that preserve a subgroup of the full global symmetry. For the resulting state to be a valid boundary state of the $\mathbb{Z}_N^{(0)} \times \mathbb{Z}_N^{(1)}$ SPT, the perturbations must preserve the $\mathbb{Z}_N^{(0)} \times \mathbb{Z}_N^{(1)}$ subgroup probed by the background fields $A_\mu$ and $B_{\mu\nu}$. We first focus on the case in which

only this minimal subgroup is preserved by the additional perturbations. The operator of the lowest scaling dimension that explicitly breaks the $U(1)$ winding symmetry is

$$\widetilde{\mathcal{O}} = -\tilde{w}\cos(\vartheta), \tag{8.10}$$

where $\tilde{w}$ is a real coupling constant. This operator has a scaling dimension of $\widetilde{\Delta} = \pi K$. On the other hand, we cannot form any local gauge invariant operators from the dual of $\tilde{\phi}$ because its corresponding winding symmetry is gauged. The remaining allowed operators can be composed from $\phi$ and $\tilde{\phi}$. These operators take the form

$$\mathcal{O}_{q,\tilde{q}} = -w_{q,\tilde{q}}\cos\left[(Nq + p\,\tilde{q})\phi - \tilde{q}\,\tilde{\phi} + q\,\Phi\right], \tag{8.11}$$

where $q, \tilde{q} \in \mathbb{Z}$ and $w_{q,\tilde{q}}$ is another real coupling constant. The scaling dimension of this operator is

$$\Delta_{q,\tilde{q}} = \frac{(Nq + p\,\tilde{q})^2}{4\pi K}. \tag{8.12}$$

From the definition $L = \gcd(N, p)$, there exist integers $r$ and $k$ such that

$$Nr + p\,k = L, \tag{8.13}$$

and $L$ is the smallest positive integer that can be formed from a linear combination of $N$ and $p$ with integer coefficients. Thus, in this family of operators, $\mathcal{O}_{r,k}$ has the lowest scaling dimension, which is

$$\Delta_{r,k} = \frac{L^2}{4\pi K}. \tag{8.14}$$

However, the choice of $r$ and $k$ is not unique. Indeed, for a given solution of $r$ and $k$, a different solution is given by $r' = r + p/L$ and $k' = k - N/L$. We therefore must add multiple operators of the same scaling dimension as $\mathcal{O}_{r,k}$. If we fix a particular $r$ and $k$, then the sum of these operators with the same scaling dimension is

$$\mathcal{O} = -\sum_{j=0}^{L-1} t_j \cos\left[L\,\phi - \left(k - \frac{N}{L}j\right)\tilde{\phi} + \left(r + \frac{p}{L}j\right)\Phi\right], \tag{8.15}$$

where $t_j$ is a real constant. Because $\tilde{\phi}$ is gapped, each operator in this sum is equivalent in the critical theory, Eq. (8.1), but in order to determine the physics of the gapped phase in which $\mathcal{O}$ is relevant, we must include all of these operators. We need to consider only a finite number of operators in this sum because the constraint, Eq. (8.8), ensures that the index $j$ is periodic with period $L$.

**(a)** Boundary with $\mathbb{Z}_N^{(0)} \times \mathbb{Z}_N^{(1)}$  **(b)** Boundary with $\mathbb{Z}_N^{(0)} \times \mathbb{Z}_N^{(0)} \times \mathbb{Z}_N^{(1)}$  **(c)** Boundary with $\mathbb{Z}_{Np}^{(0)} \times \mathbb{Z}_N^{(1)}$

**Figure 9:** Examples of $\mathbb{Z}_N^{(0)} \times \mathbb{Z}_N^{(1)}$ SPT boundary transitions captured by our gapless state, Eq. (8.1), as a function of the Luttinger parameter $K$ if different global symmetries are preserved. In all cases, the gapless phase is Eq. (8.1), and the phase for $K < 2/\pi$ has $\mathbb{Z}_N^{(0)}$ symmetry breaking. If the only symmetry of the gapless theory preserved is the $\mathbb{Z}_N^{(0)} \times \mathbb{Z}_N^{(1)}$ symmetry probed by the background fields, then for $L = \gcd(N, p) \geq 4$, the gapless state is a critical point or line separating gapped phases of $\mathbb{Z}_N^{(0)}$ symmetry breaking with different order parameters (9a). If we instead preserve the $\mathbb{Z}_N^{(0)} \times \mathbb{Z}_N^{(0)} \times \mathbb{Z}_N^{(1)}$ subgroup (9b) or the $\mathbb{Z}_{Np}^{(0)} \times \mathbb{Z}_N^{(1)}$ subgroup (9c) of the gapless phase, then we obtain $\mathbb{Z}_N^{(0)} \times \mathbb{Z}_N^{(0)}$ or $\mathbb{Z}_{Np}^{(0)}$ symmetry breaking phases respectively at large $K$.

The operator $\widetilde{\mathcal{O}}$ is relevant for

$$K < \frac{2}{\pi}, \tag{8.16}$$

and the operator $\mathcal{O}$ is relevant for

$$K > \frac{L^2}{8\pi}. \tag{8.17}$$

We thus find that the gapless phase is stable to these perturbations that preserve only the minimal $\mathbb{Z}_N^{(0)} \times \mathbb{Z}_N^{(1)}$ symmetry probed by the background fields $A_\mu$ and $B_{\mu\nu}$ for

$$\frac{2}{\pi} \leq K \leq \frac{L^2}{8\pi}, \qquad L = \gcd(N, p) \geq 4. \tag{8.18}$$

For $L = 4$, the gapless state is a critical point that separates two gapped phases, and for $L > 4$, the gapless state represents a critical line.

We now determine the gapped phases that arise when $\widetilde{\mathcal{O}}$ or $\mathcal{O}$ is relevant. When $\widetilde{\mathcal{O}}$ is relevant, the local operators that do not commute with $e^{i\vartheta}$ become trivial in the low energy limit, gapping the field $\phi$. The remaining effective action is

$$S_{\text{SSB}} = \frac{iN}{2\pi} \int_{\partial X} \left( \alpha \, d\tilde{\phi} + B \, \tilde{\phi} - p \, A \, \alpha \right) + S_0[\Phi, A; C, B], \tag{8.19}$$

indicating that the $\mathbb{Z}_N^{(0)}$ symmetry that acts on $\tilde{\phi}$ is spontaneously broken.

If $\mathcal{O}$ is relevant, the physics is determined by minimizing the potential in Eq. (8.15) for each $j \in \mathbb{Z}$. If we take the background field to be trivial, we must have

$$L \phi - \left( k - \frac{N}{L} j \right) \tilde{\phi} = 2\pi k_j, \tag{8.20}$$

where $k_j \in \mathbb{Z}$. Subtracting these equations for different $j$, we find that

$$\tilde{\phi} = \frac{2\pi L(k_1 - k_0)}{N} \in \frac{2\pi}{N/L}\mathbb{Z}, \qquad k_j = j(k_1 - k_0) + k_0, \tag{8.21}$$

which is already consistent with the constraint, Eq. (8.8), imposed by the gauge field $\alpha_\mu$. Using that $L\phi - k\tilde{\phi} = 2\pi k_0$, we can solve for $\phi$ to obtain

$$\phi = \frac{2\pi}{N}\left(\frac{N}{L}k_0 + k(k_1 - k_0)\right) \in \frac{2\pi}{N}\mathbb{Z}. \tag{8.22}$$

From $\frac{N}{L}r + \frac{p}{L}k = 1$, we know that $\gcd(N/L, k) = 1$, so there always exist choices of $k_0$ and $k_1$ that lead to

$$\left\langle e^{i\phi} \right\rangle = e^{2\pi i \tilde{z}/N}, \tag{8.23}$$

for any $\tilde{z} \in \mathbb{Z}$. We thus conclude that a $\mathbb{Z}_N^{(0)}$ global symmetry that acts on $\phi$ is spontaneously broken in this phase. Note that the expectation value of $e^{i\tilde{\phi}}$ is not independent since

$$\left\langle e^{ip\phi} \right\rangle = \left\langle e^{i\tilde{\phi}} \right\rangle. \tag{8.24}$$

This gapped phase therefore has $N$ ground states and represents the spontaneous symmetry breaking of the $\mathbb{Z}_N^{(0)}$ subgroup probed by $A_\mu$ (cf. Eq. (8.6)).

To summarize, if we deform the gapless boundary state, Eq. (8.1), by operators that preserve the $\mathbb{Z}_N^{(0)} \times \mathbb{Z}_N^{(1)}$ that is probed by the background fields $A_\mu$ and $B_{\mu\nu}$ but explicitly break the other symmetries, then we obtain the phase diagram depicted in Figure 9a. For $L = \gcd(N, p) \geq 4$, the gapless state is robust to all these deformations for

$$\frac{2}{\pi} \leq K \leq \frac{L^2}{8\pi}. \tag{8.25}$$

The gapless phase has enhanced symmetries: an emergent $U(1)$ winding symmetry and an enhancement of $\mathbb{Z}_N^{(0)}$ to a $U(1)$ momentum symmetry acting on $\phi$ and a $\mathbb{Z}_N^{(0)}$ symmetry acting on $\tilde{\phi}$. For $K < 2/\pi$, the operator $\widetilde{\mathcal{O}}$ is relevant, gapping $\phi$ but leaving the gapped sector that spontaneously breaks the $\mathbb{Z}_N^{(0)}$ symmetry acting on $\tilde{\phi}$. For $K > L^2/8\pi$, the diagonal $\mathbb{Z}_N^{(0)}$ symmetry acting on $\phi$ and $\tilde{\phi}$ is spontaneously broken.

We can access other kinds of boundary phase transitions if we consider only perturbations that preserve a larger subgroup of the symmetry of the gapless state. Suppose we additionally preserve the $\mathbb{Z}_N^{(0)}$ symmetry that acts on $\tilde{\phi}$ only (cf. Eq. (8.4)). We can still add the perturbation $\widetilde{\mathcal{O}}$, which leads to the same $\mathbb{Z}_N^{(0)}$ symmetry breaking as before. But the operator composed of $\phi$ and/or $\tilde{\phi}$ with the lowest scaling dimension is now

$$\mathcal{O}_{1,0} = -u_{1,0}\cos(N\phi + \Phi), \tag{8.26}$$

which has scaling dimension $\Delta_{1,0} = N^2/4\pi K$. When this operator is relevant, the effective field theory deep within this phase is

$$S_{\text{bdry}}[A, B] = \frac{iN}{2\pi} \int_{\partial X} \left( \alpha \, d\tilde{\phi} + \tilde{a} \, (d\phi + A) + B \, \tilde{\phi} - p \, A \, \alpha \right) + S_0[\Phi, A; C, B], \qquad (8.27)$$

where $\tilde{a}_\mu$ is a new $U(1)$ one-form gauge field serving as a Lagrange multiplier that strictly enforces the constraint that $\mathcal{O}_{1,0}$ imposes energetically. We see that this action is the same as Eq. (6.12), so the resulting state in this boundary phase is the SPT boundary of Section 6.2.1 that spontaneously breaks $\mathbb{Z}_N^{(0)} \times \mathbb{Z}_N^{(0)}$. For $N \geq 4$, the gapless theory separates this $\mathbb{Z}_N^{(0)} \times \mathbb{Z}_N^{(0)}$ symmetry breaking phase from a state that spontaneously breaks $\mathbb{Z}_N^{(0)}$, as summarized in Figure 9b.

If we instead preserve the $\mathbb{Z}_{Np}^{(0)}$ symmetry acting on $\phi$ and $\tilde{\phi}$ as

$$\phi \to \phi + \frac{2\pi}{Np}, \qquad \tilde{\phi} \to \tilde{\phi} + \frac{2\pi}{N}, \qquad (8.28)$$

and the $\mathbb{Z}_N^{(1)}$ symmetry, then the operator composed from $\phi$ and/or $\tilde{\phi}$ with the lowest scaling dimension is

$$\mathcal{O}_{0,1} = -u_{0,1} \cos(p \, \phi - \tilde{\phi}), \qquad (8.29)$$

which has scaling dimension $\Delta_{0,1} = p^2/4\pi K$. When $\mathcal{O}_{0,1}$ is relevant, $\tilde{\phi}$ becomes pinned to $p\,\phi$, so the effective field theory deep within this phase is

$$\widetilde{S}_{\text{bdry}}[A, B] = \frac{iNp}{2\pi} \int_{\partial X} (\alpha \, d\phi + B \, \phi - A \, \alpha) + S_0[\Phi, A; C, B], \qquad (8.30)$$

which is the gapped state studied in Section 6.2.2 that spontaneously breaks the $\mathbb{Z}_{Np}^{(0)}$ symmetry. In this case, as depicted in Figure 9c, the gapless theory separates a $\mathbb{Z}_N^{(0)}$ symmetry breaking phase from a phase with spontaneously broken $\mathbb{Z}_{Np}^{(0)}$ for $p \geq 4$.

Finally, one may wonder what happens if we make the background fields of Eq. (8.1) dynamical so that we obtain a boundary state of the $(N, p)$ oblique phase. This process results in an orbifolded boundary theory, which is gapless. However, the orbifolding changes the Luttinger parameter as $K \to K/N^2$, which modifies the scaling dimension of $\widetilde{\mathcal{O}}$ to $\pi K/N^2$ while keeping the scaling dimensions of the $\mathcal{O}_{q,\tilde{q}}$ operators the same. Thus, there is always a relevant operator that destabilizes the gapless boundary state.

# 9 $\mathbb{Z}_N$ axion model and non-invertible symmetry

Having now discussed the rich phases realized in our lattice model, Eq. (2.1), in this section we explore some natural generalizations of this lattice model with more exotic kinds of

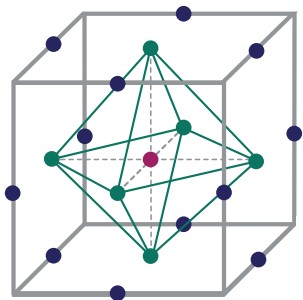

**Figure 10:** Depiction of the degrees of freedom in the lattice model, Eq. (9.3), on a single cube of the direct lattice. The $\mathbb{Z}_N$ gauge field $k_\mu(r)$ is defined on links $\ell$ of the direct lattice, labeled by the blue points. The $\mathbb{Z}_N$ spin variables $k(R)$ are on sites $R$ of the dual lattice, which are depicted by red points. The $\mathbb{Z}_N$ axion $\tilde{k}(\mathcal{R})$ lives on the green points at the centers of plaquettes of the direct lattice. The green links connect nearest neighbors of sites $\mathcal{R}$ where the $\mathbb{Z}_N$ axion is defined.

intertwined symmetries. We previously demonstrated in Section 2.3 that although our lattice model is invariant under $\Theta \to \Theta + 2\pi$, this periodicity in $\Theta$ is broken by background fields for the global symmetry $G = \mathbb{Z}_N^{(0)} \times \mathbb{Z}_N^{(1)}$. This property is reminiscent of an 't Hooft anomaly, but it is not the same because $\Theta$ is a coupling constant rather than a dynamical variable. If we then generalize our lattice model by promoting $\Theta$ to a dynamical matter variable whose periodicity is now a genuine global symmetry, then this new global symmetry can have a mixed 't Hooft anomaly with the original $G = \mathbb{Z}_N^{(0)} \times \mathbb{Z}_N^{(1)}$ symmetry.

In Section 9.1, we promote $\Theta$ to a matter field that transforms under a new $\mathbb{Z}_N^{(0)}$ global symmetry that has a mixed 't Hooft anomaly with $G$, and we examine some of the consequences. We refer to this model as a $\mathbb{Z}_N$ axion model in analogy to the $U(1)$ axion that couples to a theta term in (3+1)d. In Section 9.2, we consider a related lattice model in which the global symmetry $G$ of the $\mathbb{Z}_N$ axion model is gauged. Because of the mixed 't Hooft anomaly, the $\mathbb{Z}_N^{(0)}$ symmetry associated with the $\mathbb{Z}_N$ axion is anomalously broken. However, it can also be viewed as a non-invertible global symmetry, and we demonstrate that this lattice model has an interesting phase in which the non-invertible symmetry is spontaneously broken, which leads to nontrivial fusion rules for domain walls. These domain walls separate different oblique phases and are directly related to the boundary states we discussed previously in Section 6.3.1.

## 9.1 $\mathbb{Z}_N$ axion model: Mixed anomalies

Before generalizing our original lattice model, it is helpful to express the original model, Eq. (2.13), in a form that generalizes more easily. If we sum over the $n$ and $n_\mu$, then by the

Poisson summation formula, $\varphi$ and $a_\mu$ in Eq. (2.13) are effectively constrained so that

$$\varphi(R) \to \frac{2\pi k(R)}{N}, \qquad a_\mu(r) \to \frac{2\pi k_\mu(r)}{N}, \tag{9.1}$$

where $k, k_\mu \in \mathbb{Z}$. The action in Eq. (2.13) then reduces down to

$$S = \frac{1}{4}\left(\frac{2\pi}{Ne}\right)^2 \sum_P (\Delta_\mu k_\nu - \Delta_\nu k_\mu - N s_{\mu\nu} - B_{\mu\nu})^2 + \frac{1}{2}\left(\frac{2\pi}{Ng}\right)^2 \sum_{\tilde{\ell}} (\Delta_\mu k - N s_\mu - A_\mu)^2$$
$$+ \frac{i\Theta}{2N} \sum_{r,R} \varepsilon_{\mu\nu\lambda} (\Delta_\mu k - N s_\mu - A_\mu)(\Delta_\nu k_\lambda - \Delta_\lambda k_\nu - N s_{\nu\lambda} - B_{\nu\lambda}). \tag{9.2}$$

Now we are ready to generalize this model. We promote the parameter $\Theta$ to a dynamical variable. Specifically, we make the replacement $\Theta \to N\theta(\mathcal{R})$, where $\theta(\mathcal{R}) = \frac{2\pi\tilde{k}(\mathcal{R})}{N} \in \frac{2\pi}{N}\mathbb{Z}$ is now a dynamical matter field, which call a $\mathbb{Z}_N$ axion. The $\mathbb{Z}_N$ axion is defined on sites $\mathcal{R}$ of a new lattice at the centers plaquettes of the direct lattice (see Figure 10). The modified partition function is

$$Z_{\text{axion}}[A_\mu, B_{\mu\nu}] = \sum_{\{k, \tilde{k}, k_\mu, s_\mu, s_{\mu\nu}\}} e^{-S[k, \tilde{k}, k_\mu, s_\mu, s_{\mu\nu}; A_\mu, B_{\mu\nu}]},$$

$$S = \frac{1}{4}\left(\frac{2\pi}{Ne}\right)^2 \sum_P (\Delta_\mu k_\nu - \Delta_\nu k_\mu - N s_{\mu\nu} - B_{\mu\nu})^2$$
$$+ \frac{1}{2}\left(\frac{2\pi}{Ng}\right)^2 \sum_{\tilde{\ell}} (\Delta_\mu k - N s_\mu - A_\mu)^2 - J\sum_{\langle \mathcal{R}, \mathcal{R}'\rangle} \cos\left(\frac{2\pi}{N}[\tilde{k}(\mathcal{R}) - \tilde{k}(\mathcal{R}')]\right)$$
$$+ \frac{i\pi}{N} \sum_{r,R,\mathcal{R}} \varepsilon_{\mu\nu\lambda} \tilde{k}(\mathcal{R})(\Delta_\mu k - A_\mu)(\Delta_\nu k_\lambda - \Delta_\lambda k_\nu - B_{\nu\lambda}), \tag{9.3}$$

where $J > 0$ is a ferromagnetic coupling and the sum over $\langle \mathcal{R}, \mathcal{R}'\rangle$ is a sum over nearest neighbors.

In addition to the $G = \mathbb{Z}_N^{(0)} \times \mathbb{Z}_N^{(1)}$ global symmetry of the original lattice model, Eq. (9.3) has an additional $\mathbb{Z}_N^{(0)}$ global symmetry,

$$\theta(\mathcal{R}) \to \theta(\mathcal{R}) + \frac{2\pi}{N}, \tag{9.4}$$

where $\theta(\mathcal{R}) = 2\pi\tilde{k}(\mathcal{R})/N$. We refer to this symmetry as $(\mathbb{Z}_N^{(0)})_{\text{axion}}$. The full global symmetry of Eq. (9.3) is then

$$\mathcal{G} = (\mathbb{Z}_N^{(0)})_{\text{axion}} \times G. \tag{9.5}$$

As discussed at the beginning of this section, we note that the $(\mathbb{Z}_N^{(0)})_{\text{axion}}$ global symmetry descends from the periodicity of $\Theta \sim \Theta + 2\pi$ in the original lattice model, Eq. (2.1), which is the transformation $\mathcal{T}$. However, as we found in Section 2.3, although $\Theta$ has $2\pi$ periodicity if there are no background fields for $G$, if we introduce nontrivial background fields, $A_\mu$ and $B_{\mu\nu}$, for the $G = \mathbb{Z}_N^{(0)} \times \mathbb{Z}_N^{(1)}$ global symmetry, the action will change under $\mathcal{T}$ by a term that depends only on the background fields, Eq. (2.20). In our axion model, Eq. (9.3), this observation implies that there is a mixed 't Hooft anomaly for $(\mathbb{Z}_N^{(0)})_{\text{axion}}$ and $G$. Indeed, if we introduce nontrivial background gauge fields for $G$, then acting with the $(\mathbb{Z}_N^{(0)})_{\text{axion}}$ symmetry, Eq. (9.4) changes the action by Eq. (2.20), indicating that there is a mixed anomaly.

A consequence of this mixed anomaly is that the model has no trivial phase that preserves the full symmetry group $\mathcal{G}$. We will not attempt to determine the full phase diagram of the model, but in the interest of finding more exotic phases not already present in the original model, Eq. (2.13), we concentrate on the phase that spontaneously breaks $(\mathbb{Z}_N^{(0)})_{\text{axion}}$ but preserves $G$. Such a phase occurs in the limit in which $g^2$, $e^2$, and $J$ are all large. In this phase, there are $N$ degenerate ground states, and in each ground state, the order parameter $e^{i\theta(\mathcal{R})}$ acquires a nonzero expectation value,

$$\left\langle e^{i\theta(\mathcal{R})} \right\rangle = e^{2\pi i\, p/N}, \tag{9.6}$$

for some integer $p \in \mathbb{Z}$ that labels the distinct ground states. For a given ground state labeled by $p$, we obtain the same state of the original lattice model, Eq. (2.13), at $\Theta = 2\pi p$ and large $g^2$ and $e^2$. As discussed in Section 2.4, this state is an SPT protected by $G$ with action,

$$S_{\text{SPT}} = \frac{iNp}{2\pi} \int A \wedge B - \frac{iN}{2\pi} \int \varphi \wedge dB + \frac{iN}{2\pi} \int a \wedge dA, \tag{9.7}$$

in the notation for continuum fields. Here, $\varphi$ is a dynamical $2\pi$ periodic scalar and $a_\mu$ is a dynamical $U(1)$ one-form gauge field. Integrating over these two dynamical fields constrains the background $U(1)$ one-form gauge field $A_\mu$ and background $U(1)$ two-form gauge field $B_{\mu\nu}$ to be $\mathbb{Z}_N$ background gauge fields.

In Appendix E.1, we derive that the effective field theory in the continuum for this phase at large $g^2$, $e^2$, and $J$ is

$$S = \frac{iN}{2\pi} \int \theta \wedge \left( d\beta + \frac{N}{2\pi} A \wedge B \right) - \frac{iN}{2\pi} \int \varphi \wedge dB + \frac{iN}{2\pi} \int a \wedge dA, \tag{9.8}$$

where $\theta$ is a dynamical $2\pi$ periodic scalar, $\beta_{\mu\nu}$ is a dynamical $U(1)$ two-form gauge field, and the other fields are the same as in Eq. (9.7). To see that Eq. (9.8) reproduces the correct

physics of this phase, we first turn off the background fields, setting $A_\mu = B_{\nu\lambda} = 0$. The action in Eq. (9.8) then reduces to

$$S_{\text{SSB}} = \frac{iN}{2\pi} \int \theta \wedge d\beta, \tag{9.9}$$

which is the effective field theory for a phase with spontaneously broken $(\mathbb{Z}_N^{(0)})_{\text{axion}}$ symmetry. Indeed, integrating out $\beta_{\mu\nu}$ implements the constraint that $\theta = 2\pi p/N$ for some $p \in \mathbb{Z}$, giving $e^{i\theta}$ a nonzero expectation value. If we now introduce nontrivial background gauge fields, then setting $\theta = 2\pi p/N$ for a given $p$ results in the SPT action of Eq. (9.7).

## 9.2 Gauged model: Non-invertible symmetry breaking

We now consider a generalization of the model in the previous section in which we gauge the $G = \mathbb{Z}_N^{(0)} \times \mathbb{Z}_N^{(1)}$ symmetry, promoting the background gauge fields that probe this symmetry into dynamical fields. The partition function for this lattice model is

$$\mathcal{Z}_{\text{axion}}[A_\mu, B_{\mu\nu}] = \sum_{\{c_\mu, b_{\mu\nu}\}} Z_{\text{axion}}[c_\mu, b_{\mu\nu}] \exp\left(-\frac{i\pi}{N} \sum_{r,R} \varepsilon_{\mu\nu\lambda} \left(c_\mu B_{\nu\lambda} + b_{\mu\nu} A_\lambda\right)\right), \tag{9.10}$$

where $c_\mu \in \mathbb{Z}$ and $b_{\mu\nu} \in \mathbb{Z}$ are locally flat dynamical $\mathbb{Z}_N$ gauge fields, $A_\mu$ and $B_{\mu\nu}$ are locally flat background $\mathbb{Z}_N$ gauge fields, and $Z_{\text{axion}}$ is defined in Eq. (9.3). We can equivalently think of $\mathcal{Z}_{\text{axion}}$ is the image of $Z_{\text{axion}}$ under $\mathcal{S}$ as defined in Eq. (2.26).

Turning to the global symmetries of this new model, Eq. (9.10), we note that gauging $G$ in Eq. (9.3) results in a dual symmetry $\widetilde{G} = \mathbb{Z}_N^{(0)} \times \mathbb{Z}_N^{(1)}$, which is probed by the background fields $A_\mu$ and $B_{\mu\nu}$ in Eq. (9.10). Because $(\mathbb{Z}_N^{(0)})_{\text{axion}}$ has a mixed anomaly with $G$ in Eq. (9.3), the $(\mathbb{Z}_N^{(0)})_{\text{axion}}$ symmetry is now anomalously broken in the gauged model, Eq. (9.10). However, as demonstrated in Appendix E.2, although the partition function is not invariant under

$$\theta(\mathcal{R}) \to \theta(\mathcal{R}) + \frac{2\pi p}{N}, \qquad p \in \mathbb{Z}, \tag{9.11}$$

it will be invariant if we also act with the transformation $\mathcal{S}\mathcal{T}^{-p}\mathcal{S}$ (and give minus signs to the background fields), where we recall that $\mathcal{S}$ and $\mathcal{T}$ are defined in Eq. (2.26). Because this combined operation involves duality $\mathcal{S}$, it is not a conventional symmetry, but it may be viewed as a non-invertible symmetry. Non-invertible symmetries also arise in (3+1)d axion models [22, 76, 77].

To be explicit, the non-invertible symmetry is associated with a surface operator (or defect) that is topological. For example, as shown in Appendix E.2, if we place a defect for

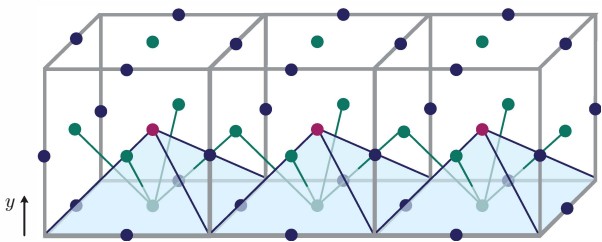

**Figure 11:** Depiction of the domain wall for the $\mathbb{Z}_N$ axion. This domain wall is the non-invertible defect of Eq. (9.12). The vertical direction is the $y$-direction, and the plaquettes at the bottoms of the blue pyramids define the $y = 0$ plane. The $y = 1/2$ plane intersects the red points at the peaks of the pyramids. We have depicted three cubes, but of course, the domain wall extends infinitely in the imaginary time $t$ and $x$ directions. The domain wall lies on the blue triangles. The axion variables at the ends of the green links that intersect a blue triangle differ by a $(\mathbb{Z}_N^{(0)})_{\text{axion}}$ transformation.

the non-invertible symmetry along a surface $\Sigma$, then the action of the defect is

$$
S_d = -J \sum_{\langle \mathcal{R}, \mathcal{R}' \rangle \in \Sigma} \cos\left( \theta(\mathcal{R}) - \theta(\mathcal{R}') + \frac{2\pi p}{N} \right) + \frac{2\pi i p}{N} \sum_{\{r, R\} \in \Sigma} \varepsilon_{\mu\nu} \left( k \, \Delta_\mu k_\nu - c_\mu \, k_\nu - \frac{1}{2} k \, b_{\mu\nu} \right)
$$
$$
- \frac{2\pi i}{N} \sum_{\{r, R\} \in \Sigma} \varepsilon_{\mu\nu} \left( -p \, \phi \, \Delta_\mu \alpha_\nu + \phi \, \Delta_\mu \tilde{\alpha}_\nu + \tilde{\phi} \, \Delta_\mu \alpha_\nu + \tilde{\alpha}_\mu \, c_\nu + \frac{1}{2} \, \tilde{\phi} \, b_{\mu\nu} \right).
$$
$$
(9.12)
$$

To be more concrete, we choose $\Sigma$ to be the surface of triangles depicted in Figure 11, which are just above plaquettes in the plane $y = 0$ in the direct lattice. In $S_d$ the sum over $\langle \mathcal{R}, \mathcal{R}' \rangle \in \Sigma$ means we sum over links connecting axion sites $\mathcal{R}$ and $\mathcal{R}'$, where $\mathcal{R}$ is below $\Sigma$ (in the plane $y = 0$) and $\mathcal{R}'$ is above $\Sigma$ (in the plane $y = 1/2$). The gauge fields $\alpha_\mu, \tilde{\alpha}_\mu \in \mathbb{Z}$ lie along links in the plane $y = 0$, and the lattice variables $\phi, \tilde{\phi} \in \mathbb{Z}$ are on dual sites in the plane $y = 1/2$. The Euclidean spacetime indices are over imaginary time $t$ and the spatial direction $x$. The first line of Eq. (9.12) indicates that $\theta(\mathcal{R}) = 2\pi \tilde{k}(\mathcal{R})/N$ increases by $2\pi p/N$ if $\theta$ crosses from above the domain wall to below it. The meaning of the second line of Eq. (9.12) is that observables that cross from above to below the domain wall are acted upon by a $\mathcal{ST}^{-p}\mathcal{S}$ transformation.

This non-invertible symmetry is especially important in the phase that is the gauged descendant of Eq. (9.8). In Eq. (9.8), because of the mixed anomaly between $(\mathbb{Z}_N^{(0)})_{\text{axion}}$ and $G$, the domain walls for the $(\mathbb{Z}_N^{(0)})_{\text{axion}}$ symmetry separate distinct $G$ SPT states. If we now gauge $G$, we obtain the phase at large $g^2$, $e^2$, and $J$ in Eq. (9.10). The operator $e^{i\theta}$ still acquires a nonzero expectation value, but a domain wall $\mathcal{U}$ that interpolates between different expectation values of $e^{i\theta}$ now separates two distinct oblique phases, which generically have

topological order, ordinary spontaneous symmetry breaking, and $G$ SPT order. Just as with the boundary states in Section 6, we must then decorate the domain wall $\mathcal{U}$ with additional dynamical degrees of freedom, resulting in nontrivial fusion rules for $\mathcal{U}$. Since $e^{i\theta}$ acquires a nonzero expectation value, this phase may be regarded as having a spontaneously broken non-invertible symmetry.

We shall proceed to develop these ideas in more detail. The $G$ gauged version of Eq. (9.8) is

$$
S = \frac{iN}{2\pi} \int \theta \wedge \left( d\beta + \frac{N}{2\pi} c \wedge b \right) - \frac{iN}{2\pi} \int \varphi \wedge db + \frac{iN}{2\pi} \int a \wedge dc, \tag{9.13}
$$

where $c_\mu$ is a dynamical $U(1)$ one-form gauge field, $b_{\mu\nu}$ is a dynamical $U(1)$ two-form gauge field, and the other fields are the same as defined in Eq. (9.8). This action, Eq. (9.13), is invariant under the unusual gauge transformation,

$$
c \to c + d\xi, \qquad b \to b + d\lambda, \qquad \varphi \to \varphi - \frac{N}{2\pi}\theta\,\xi, \qquad a \to a + d\chi - \frac{N}{2\pi}\theta\,\lambda,
$$
$$
\beta \to \beta + d\tilde{\lambda} - \frac{N}{2\pi}\xi \wedge b + \frac{N}{2\pi}c \wedge \lambda - \frac{N}{2\pi}\xi \wedge d\lambda, \tag{9.14}
$$

where $\xi$ is a $2\pi$ periodic scalar while $\lambda_\mu$ and $\tilde{\lambda}_\mu$ are $U(1)$ one-form gauge fields. Note that integrating out $\beta_{\mu\nu}$ imposes the constraint that $\theta = 2\pi p/N$, where $p \in \mathbb{Z}$. For a given value of $p$, the action, Eq. (9.13), reduces to the action for the effective field theory of an oblique phase, Eq. (2.37). In the path integral, we are then summing over all oblique phases for a given $N$.

We now turn to the global symmetries of Eq. (9.13), starting with the invertible zero-form and one-form symmetries. The $\mathbb{Z}_N^{(0)}$ global symmetry acts as,

$$
\varphi \to \varphi + \frac{2\pi}{N}, \tag{9.15}
$$

and is associated with the symmetry operator,

$$
U(\Sigma) = \exp\left( i \oint_\Sigma b \right), \tag{9.16}
$$

where $\Sigma$ is a surface. There is also a $\mathbb{Z}_N^{(1)}$ global symmetry,

$$
a \to a + \frac{\eta}{N}, \tag{9.17}
$$

where $\eta$ is a locally flat connection, $d\eta = 0$, with quantized fluxes $\oint \eta \in 2\pi\mathbb{Z}$. The symmetry operator for the $\mathbb{Z}_N^{(1)}$ symmetry is

$$
W_c(\Gamma) = \exp\left( i \oint_\Gamma c \right). \tag{9.18}
$$

These two global symmetries form the $\widetilde{G} = \mathbb{Z}_N^{(0)} \times \mathbb{Z}_N^{(1)}$ symmetry of the lattice model, Eq. (9.10).

The non-invertible symmetry is more subtle. In the effective field theory, Eq. (9.8), for the normal $(\mathbb{Z}_N^{(0)})_{\text{axion}}$ symmetry breaking phase, the symmetry operator for a $(\mathbb{Z}_N^{(0)})_{\text{axion}}$ transformation is

$$\widetilde{U}(\Sigma)^p = \exp\left(ip \oint_\Sigma \beta\right), \qquad p \in \mathbb{Z}, \tag{9.19}$$

where $\Sigma$ is a closed surface. However, if we now gauge $G$ to obtain the theory, Eq. (9.13), we now impose gauge transformations, Eq. (9.14). Consequently, $\widetilde{U}(\Sigma)$ is no longer gauge invariant. Furthermore, we cannot obtain a gauge invariant operator by simply dressing $\widetilde{U}(\Sigma)$ with $\theta$, $\varphi$, $a_\mu$, $c_\mu$, and $b_{\mu\nu}$. Instead, as in Section 6, we must decorate $\widetilde{U}(\Sigma)$ with additional degrees of freedom. The resulting surface operator is

$$\mathcal{U}_p(\Sigma) = \int \mathcal{D}\phi \, \mathcal{D}\tilde{\phi} \, \mathcal{D}\alpha \, \mathcal{D}\tilde{\alpha} \exp\left(ip \oint_\Sigma \beta + \frac{iN}{2\pi} \oint_\Sigma \left(p\, \alpha\, d\phi - \tilde{\alpha}\, d\phi - \alpha\, d\tilde{\phi} - b\, \tilde{\phi} + c\, \tilde{\alpha}\right)\right), \tag{9.20}$$

where $\phi$ and $\tilde{\phi}$ are dynamical $2\pi$ periodic scalar fields while $\alpha_\mu$ and $\tilde{\alpha}_\mu$ are dynamical $U(1)$ one-form gauge fields, all of which are defined solely on $\Sigma$. The surface operator $\mathcal{U}_p(\Sigma)$ is the continuum analogue of Eq. (9.12). These fields transform under the gauge symmetry, Eq. (9.14), as

$$\phi \to \phi - \xi, \qquad \alpha \to \alpha - \lambda, \qquad \tilde{\phi} \to \tilde{\phi} - p\,\xi, \qquad \tilde{\alpha} \to \tilde{\alpha} - p\,\lambda, \tag{9.21}$$

so that the surface operator $\mathcal{U}_p(\Sigma)$ gauge invariant. Note that the $(1+1)$d field theory we have introduced along $\Sigma$ resembles the boundary state of Section 6.3.1. This resemblance is not an accident since $\mathcal{U}_p(\Sigma)$ can be regarded as a boundary between two distinct oblique states.

As demonstrated in Appendix E.3, the fusion rules for $\mathcal{U}_p(\Sigma)$ are

$$\mathcal{U}_{p_1}(\Sigma) \times \mathcal{U}_{p_2}(\Sigma) = \left(\mathcal{Z}_N\right)^2 \mathcal{U}_{p_1+p_2}(\Sigma), \tag{9.22}$$

where $\mathcal{Z}_N$ is the partition function for $(1+1)$d BF theory at level $N$, which describes a state with spontaneously broken $\mathbb{Z}_N^{(0)}$ global symmetry along the surface $\Sigma$. More explicitly, we define

$$\mathcal{Z}_N = \int \mathcal{D}\phi \, \mathcal{D}\alpha \, \exp\left(-\frac{iN}{2\pi} \oint_\Sigma \alpha\, d\phi\right), \tag{9.23}$$

where $\phi$ is a $2\pi$ periodic scalar and $\alpha_\mu$ is a $U(1)$ one-form gauge field. In particular, for $p_1 = p$ and $p_2 = -p$, we obtain

$$\mathcal{U}_p(\Sigma) \times \mathcal{U}_p(\Sigma)^\dagger = \mathcal{U}_p(\Sigma) \times \mathcal{U}_{-p}(\Sigma) = \left(\mathcal{Z}_N\right)^2 \mathcal{C}_N(\Sigma) \neq 1 \tag{9.24}$$

so that $\mathcal{U}_p(\Sigma)$ is in fact non-invertible. Here, we have defined

$$\mathcal{C}_N(\Sigma) = \int \mathcal{D}\phi \, \mathcal{D}\tilde{\phi} \, \mathcal{D}\alpha \, \mathcal{D}\tilde{\alpha} \, \exp\left[-\frac{iN}{2\pi} \oint_\Sigma \left(\alpha \, d\tilde{\phi} + \tilde{\alpha} \, d\phi + b \, \tilde{\phi} - c \, \tilde{\alpha}\right)\right], \tag{9.25}$$

which is a surface defect along which we gauge $\widetilde{G} = \mathbb{Z}_N^{(0)} \times \mathbb{Z}_N^{(1)}$. Thus, $U(\Sigma)$ and $W_c(\Gamma)$, which are the operators that act with this symmetry, may freely appear from or disappear into this defect $\mathcal{C}_N$.

We now consider how $\mathcal{U}_p(\Sigma)$ acts on other operators. If $\Sigma$ is a closed surface surrounding the point $\mathcal{P}$, then $\mathcal{U}_p(\Sigma)$ acts on $e^{i\theta(\mathcal{P})}$ as

$$\left\langle \mathcal{U}_p(\Sigma) \, e^{i\theta(\mathcal{P})} \cdots \right\rangle = e^{2\pi i \, p/N} \left\langle e^{i\theta(\mathcal{P})} \cdots \right\rangle, \tag{9.26}$$

where the $\cdots$ denote other operators that lie outside of $\Sigma$. Hence, $\mathcal{U}_p(\Sigma)$ acts on $e^{i\theta(\mathcal{P})}$ just as the $(\mathbb{Z}_N^{(0)})_{\text{axion}}$ symmetry operator $\widetilde{U}(\Sigma)^p$ does in the ungauged theory, Eq. (9.8). Because $e^{i\theta(\mathcal{P})}$ takes a nonzero expectation value in the phase with effective field theory, Eq. (9.13), we can regard this phase as having a spontaneously broken non-invertible symmetry.

To better understand the implications of this non-invertible symmetry breaking, we compute the ground state degeneracy on a closed manifold $X$ of genus $g_h$. Recall from Section 3 that the $(N, p)$ oblique state has a ground state degeneracy of $L^{2g_h+1}$, where $L = \gcd(N, p)$. One factor of $L$ may be attributed to zero-form symmetry breaking, $\mathbb{Z}_N^{(0)} \to \mathbb{Z}_{N/L}^{(0)}$, and the remaining factor $L^{2g_h}$ is from topological order, or equivalently, discrete one-form symmetry breaking, $\mathbb{Z}_N^{(1)} \to \mathbb{Z}_{N/L}^{(1)}$. When $e^{i\theta(\mathcal{P})}$ acquires an expectation value, all possible $(N, p)$ oblique states for a given $N$ and different $p \bmod N$ become degenerate. The full ground state degeneracy $\mathcal{D}_{\text{GS}}(X)$ in this phase is therefore

$$\mathcal{D}_{\text{GS}}(X) = \sum_{p=0}^{N-1} [\gcd(N, p)]^{2g_h+1}, \tag{9.27}$$

where the sum over $p$ is now a consequence of the non-invertible symmetry breaking.

We can also consider how $\mathcal{U}_p(\Sigma)$ acts on non-local operators. The other symmetry operators, $U(\Sigma)$ and $W_c(\Gamma)$, can disappear into $\mathcal{U}_p(\Sigma)$ so that

$$\left\langle \mathcal{U}_p(\Sigma) W_c(\Gamma) \cdots \right\rangle = \left\langle \mathcal{U}_p(\Sigma) U(\Sigma) \cdots \right\rangle = \left\langle \mathcal{U}_p(\Sigma) \cdots \right\rangle, \tag{9.28}$$

where $\Gamma$ is a loop on $\Sigma$ and the $\cdots$ denote other operators not along $\Sigma$.

The action of $\mathcal{U}_p$ on the remaining operators, which can be formed using $\varphi$ and $a_\mu$, are rather complicated in the general case, so we will consider some simple examples. Suppose

we have two (2+1)d regions, $X_1$ and $X_2$, in spacetime, and the expectation value of $e^{i\theta(\mathcal{P})}$ is given by

$$\left\langle e^{i\theta(\mathcal{P})} \right\rangle = \begin{cases} 1, & \mathcal{P} \in X_1, \\ e^{2\pi i \, p/N}, & \mathcal{P} \in X_2, \end{cases} \tag{9.29}$$

so that at the interface $\Sigma$ between $X_1$ and $X_2$ there is a domain wall $\mathcal{U}_p(\Sigma)$. The operators,

$$V(\mathcal{P}_1) = e^{i\varphi(\mathcal{P}_1)}, \qquad W_a(\Gamma_1) = \exp\left( i \oint_{\Gamma_1} a \right), \tag{9.30}$$

can be defined for a point $\mathcal{P}_1$ and line $\Gamma_1$ that both lie in $X_1$. However, if we move these operators through $\mathcal{U}_p(\Sigma)$, they become

$$V(\mathcal{P}_1) \to V(\mathcal{P}_2) \exp\left( ip \int_{\gamma_2} c \right), \qquad W_a(\Gamma_1) \to W_a(\Gamma_2) \exp\left( ip \int_{\Sigma_2} b \right), \tag{9.31}$$

where $\gamma_2$ is a curve in $X_2$ from the point $\mathcal{P}_2 \in X_2$ to the surface $\Sigma$ and $\Sigma_2$ is an open surface that connects to a loop on $\Sigma$ to the loop $\Gamma_2$ in $X_2$. The local operator is now attached to a string, and the Wilson loop is attached to a surface unless $p = 0 \mod N$. We can interpret $X_1$ as a region in which local operators of spin charge $N$ and loop operators of electric charge $N$ are condensed. In region $X_2$, we have the oblique phase $(N, p)$, so the condensed local operators here have spin charge $N$ and magnetic charge $p$, and the condensed loop operators have electric charge $N$ and vorticity $p$. Thus, while the Wilson loop is deconfined in $X_1$, where electric charges are condensed, it is confined in $X_2$ because the local operators that are condensed have nontrivial magnetic charge. Similarly, the local operator $V$ picks up a string because the condensation of vortices in $X_2$ makes the $\mathbb{Z}_N$ spins energetically costly.

To summarize, the $(\mathbb{Z}_N^{(0)})_{\text{axion}}$ global symmetry of Eq. (9.3) becomes a non-invertible symmetry in Eq. (9.10). At small $\tilde{g}^2$ but large $g^2$ and $e^2$, we obtain phases with a spontaneously broken non-invertible symmetry, which is signaled by a nonzero expectation value of the $\mathbb{Z}_N$ axion operator $e^{i\theta(\mathcal{P})}$. For a given expectation value of the axion, we obtain an oblique phase, which is characterized by ordinary symmetry breaking, topological order, and mixed SPT order for discrete zero-form and one-form symmetries. The gauged $\mathbb{Z}_N$ axion model, Eq. (9.13), indeed provides a rich platform for studying the interplay of different kinds of generalized global symmetries.

## 10    Discussion

We have presented in this work a new lattice model that exhibits a rich phase diagram with various patterns of symmetry breaking and SPT order for $\mathbb{Z}_N$ zero-form and $\mathbb{Z}_N$ one-form

global symmetries. In this model a system with a global ("zero-form") $\mathbb{Z}_N$ symmetry (i.e. a $\mathbb{Z}_N$ clock model) on a 3d Euclidean lattice is coupled to a $\mathbb{Z}_N$ gauge theory only through a topological term, which is similar to a theta term in (3+1)d. In this 3d theory the topological term is defined as a local coupling on the lattice, which is a vexing problem for the theta term in (3+1)d. This local interaction is achieved by placing the gauge fields on links of the direct lattice and the matter on sites of the dual lattice. The $\mathbb{Z}_N$ zero-form global symmetry of the matter field (in the clock model) is not gauged, and it is not charged under the $\mathbb{Z}_N$ gauge field. Thus, this model is very different than a conventional theory of a matter and a gauge field coupled together.

An important feature of our lattice model is that it has duality relations. Using duality and our knowledge of the model in the limit where the clock model and gauge theory are decoupled, we can place constraints on the phase diagram and deduce the existence of certain phases and phase transitions. For example, we can demonstrate that the model naturally leads to different kinds of oblique phases that generically have ordinary symmetry breaking, topological order, and SPT order all intertwined together. The oblique phases arise from the interplay of zero-form and one-form global symmetries by condensing bound states of the order parameter for one symmetry and the disorder operator for the other symmetry. The physics of these gapped phases is described by an effective field theory consisting of a dynamical one-form field coupled to a dynamical two-form field. Using this effective theory, we have examined response to probes and developed several gapped boundary states for these oblique phases and their cousins that have only SPT order protected by $\mathbb{Z}_N$ zero-form and one-form symmetries. In the SPT case, we developed a gapless boundary state that can describe a boundary phase transition between distinct gapped boundary states of the same bulk SPT phase. We have also extended our lattice model to have a non-invertible symmetry that is intertwined with $\mathbb{Z}_N$ zero-form and one-form symmetries. This generalized model includes a phase in which the non-invertible symmetry is spontaneously broken, leading to domain walls with exotic fusion rules.

We have focused in this work on Abelian generalized symmetries. While higher-form global symmetries are necessarily Abelian, it should be possible to develop more general models in which a non-Abelian zero-form symmetry is intertwined with an Abelian higher-form symmetry. Furthermore, in non-Abelian topological orders, there are non-invertible loop operators, physically corresponding to worldlines of non-Abelian anyons. Investigating such models could offer new insights into higher-form non-invertible symmetries and their interplay with non-Abelian symmetries.

Another promising direction for future work is to explore non-invertible one-form sym-

metries in related lattice models. The $\mathbb{Z}_N$ axion model introduced in Section 9.1 has a mixed anomaly involving two $\mathbb{Z}_N$ zero-form symmetries and a $\mathbb{Z}_N$ one-form symmetry. By gauging the one-form symmetry and one of the zero-form symmetries, we obtained a model with a non-invertible zero-form symmetry in Section 9.2. We could instead gauge the two zero-form symmetries to obtain a lattice model with a non-invertible one-form symmetry. This lattice model will have phases with non-Abelian topological order, which are characterized by a discrete non-Abelian gauge group [78–81]. It would be interesting to examine these phases from the generalized symmetries point of view.

## Acknowledgements

We thank Fiona Burnell, Junyi Cao, Hart Goldman, and Raman Sohal for many insightful comments. This work was supported in part by the grant of the National Science Foundation DMR 2225920 at the University of Illinois.

## A    Review of $\mathbb{Z}_N$ gauge fields

Here, we review the physics of $\mathbb{Z}_N$ gauge fields in continuum field theories. (See also Ref. [82, 83].) As stated in Section 2.1, a $\mathbb{Z}_N$ gauge field must be locally flat in the continuum. Let us review why, focusing on $\mathbb{Z}_N$ one-form gauge fields. (The argument for higher-form gauge fields is analogous.) Consider a Wilson loop $W_A(\gamma)$ for a $\mathbb{Z}_N$ gauge field $A_\mu$,

$$W_A(\gamma) = \exp\left(i \oint_\gamma A\right), \tag{A.1}$$

which is an $N$th root of unity. In the continuum, if $\gamma$ is a contractible loop, then we can consider how $W_A(\gamma)$ behaves as we continuously shrink $\gamma$ to a point. Under such a process, we should find that the Wilson loop continuously approaches 1: $W_A(\gamma) \to 1$. But because $\mathbb{Z}_N$ is discrete, this kind of behavior can only happen if $W_A(\gamma) = 1$ for *any* contractible $\gamma$. If we have an Wilson loop along an infinitesimal loop $\gamma_\epsilon$ in the $\mu\nu$-plane of area $\epsilon^2$, then we can relate the field strength $F_{\mu\nu} = \partial_\mu A_\nu - \partial_\nu A_\mu$ to this Wilson loop as

$$W_A(\gamma_\epsilon) = 1 + i\,\epsilon^2\,F_{\mu\nu} + \mathcal{O}(\epsilon^4). \tag{A.2}$$

We then have that $F_{\mu\nu} = 0$ locally—the condition of local flatness for $A_\mu$. In a lattice gauge theory, we do not have this restriction because no Wilson loop can be contracted to a loop smaller than a lattice plaquette.

We now discuss how to represent these gauge fields in the continuum. As described above, a $\mathbb{Z}_N$ one-form gauge field $A_\mu$ and a $\mathbb{Z}_N$ two-form gauge field $B_{\mu\nu}$ are locally flat, so we have $dA = dB = 0$ locally. But globally, these fields can have nontrivial fluxes over noncontractible cycles, satisfying

$$\oint_\Gamma A \in \frac{2\pi}{N}\mathbb{Z}, \qquad \oint_\Sigma B \in \frac{2\pi}{N}\mathbb{Z}, \tag{A.3}$$

where $\Gamma$ is a noncontractible loop and $\Sigma$ is a noncontractible surface. We can then represent $A_\mu$ and $B_{\mu\nu}$ as

$$A = \frac{d\Phi}{N}, \qquad B = \frac{dC}{N}, \tag{A.4}$$

where $\Phi$ is a $2\pi$ periodic scalar and $C_\mu$ is a $U(1)$ one-form gauge field.

Next, we discuss how to couple a field theory to background $\mathbb{Z}_N$ gauge fields. Consider a continuum field theory in a $D$-dimensional Euclidean spacetime represented by an action $S[\phi, a_\mu]$ that is a local functional of a dynamical scalar field $\phi$ and a dynamical $U(1)$ gauge field $a_\mu$. Suppose this theory has a $\mathbb{Z}_N$ zero-form global symmetry,

$$\phi \to e^{2\pi i/N}\phi, \tag{A.5}$$

and a $\mathbb{Z}_N$ one-form global symmetry,

$$a \to a + \frac{\eta}{N}, \tag{A.6}$$

where $\eta$ is a locally flat connection, $d\eta = 0$, with quantized cycles $\oint \eta \in 2\pi\mathbb{Z}$. We can probe these symmetries by coupling to background gauge fields as

$$S_{\text{probed}} = S[e^{-i\Phi/N}\phi, a_\mu - (C_\mu/N)] + \frac{i}{2\pi}\int [\alpha \wedge (NB - dC) + \beta \wedge (NA - d\Phi)], \tag{A.7}$$

where $\alpha_\mu$ and $\beta_{\mu\nu}$ are dynamical $U(1)$ one-form and two-form gauge fields respectively, $\Phi$ is a background $2\pi$ periodic scalar, $A_\mu$ and $C_\mu$ are background $U(1)$ one-form gauge fields, and $B_{\mu\nu}$ is a background $U(1)$ two-form gauge field. We impose the gauge transformations,

$$A \to A + d\xi, \qquad \Phi \to \Phi + N\xi, \qquad \phi \to e^{i\xi}\phi, \tag{A.8}$$
$$B \to B + d\lambda, \qquad C \to C + d\chi + N\lambda, \qquad a \to a + \lambda,$$

where $\xi$ and $\chi$ are $2\pi$ periodic scalar fields and $\lambda_\mu$ is a $U(1)$ one-form gauge field. The Lagrange multipliers $\alpha$ and $\beta$ implement the contraints of Eq. A.4, turning $A_\mu$ and $B_{\mu\nu}$ into $\mathbb{Z}_N$ one-form and two-form background gauge fields respectively. We thus say that in the

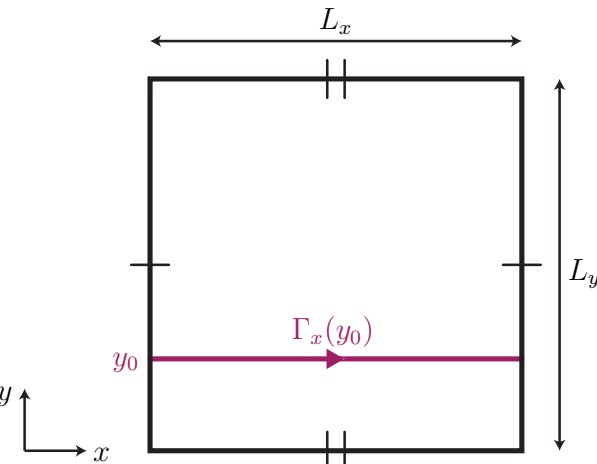

**Figure 12:** Depiction of twisted boundary conditions for a one-form global symmetry. The square above represents the $xy$-plane, which is a torus, so the opposite sides of the square are identified. We introduce a nontrivial background field $B_{xy}$. If we put a Wilson loop along the noncontractible loop $\Gamma_x(y_0)$ and adjust $y_0$ so that the Wilson loop sweeps around the $y$-direction once, then the Wilson loop will be acted on by the one-form global symmetry probed by the two-form background field.

continuum a $\mathbb{Z}_N$ one-form gauge field is formed by a pair $(\Phi, A_\mu)$ and a $\mathbb{Z}_N$ two-form gauge field is formed by the pair $(C_\mu, B_{\mu\nu})$.

Finally, we discuss the physical interpretation of probing the $\mathbb{Z}_N$ global symmetries of Eq. (A.7) with background fields. This notion is the same as introducing a defect that establishes twisted boundary conditions, which we now review. For concreteness, we work in 2+1 dimensions. Suppose we choose a configuration of $A_\mu$ such that

$$W_A(\Gamma_x) = \exp\left( i \oint_{\Gamma_x} A \right) = e^{2\pi i k/N}, \tag{A.9}$$

where $k \in \mathbb{Z}$ and $\Gamma_x$ is a noncontractible loop around the $x$-direction. This background field configuration for $A_\mu$ introduces a twisted boundary condition for the $\mathbb{Z}_N$ order parameter $\phi(t, x, y)$,

$$\phi(t, x + L_x, y) = e^{2\pi i k/N} \phi(t, x, y), \tag{A.10}$$

where $L_x$ is the distance of the $x$-direction. The symmetry operator in (2+1)d is a surface operator $U(\Sigma)$. The holonomy, Eq. (A.10), can be established by inserting a defect $U(\Sigma_{yt})^k$ along the $yt$-plane $\Sigma_{yt}$ since $U(\Sigma_{yt})^k$ is a domain wall along the $y$-directon. Thus, the background field configuration is the same as introducing this defect.

A similar picture exists for the two-form background gauge field $B_{\mu\nu}$, which may be less familar. Suppose the $xy$-plane forms a torus, which we will label $\Sigma_{xy}$. We choose a

configuration of $B_{\mu\nu}$ such that

$$U_B(\Sigma_{xy}) = \exp\left(i \oint_{\Sigma_{xy}} B\right) = e^{2\pi i k/N}, \tag{A.11}$$

where $k \in \mathbb{Z}$. We then place a Wilson loop,

$$W(\Gamma_x(y_0)) = \exp\left(i \oint_{\Gamma_x(y_0)} a\right), \tag{A.12}$$

along a noncontractible loop $\Gamma_x(y_0)$ in the $xy$-plane. The loop $\Gamma_x(y_0)$ winds around the $x$-direction, and has $y$-coordinate $y_0$ (see Figure 12). The background field configuration, Eq. (A.11), introduces a twisted boundary condition,

$$W(\Gamma_x(y_0 + L_y)) = e^{2\pi i k/N} W(\Gamma_x(y_0)) \tag{A.13}$$

where $L_y$ is the distance along the $y$-direction. In (2+1)d, the symmetry operator for a one-form symmetry is a loop operator, $T(\widetilde{\Gamma})$, respresenting the wordline of a magnetic flux. This nontrivial holonomy, Eq. (A.13), for the Wilson loop can also be achieved by inserting $T(\widetilde{\Gamma}_t)^k$ along a noncontracible loop $\widetilde{\Gamma}_t$ in the time direction. Hence, introducing the two-form background field is the same as adding a defect of this kind.

# B  Lattice model duality

Here, we elaborate on the calculation for the duality of Eq. (2.13) to Eq. (2.21), thus establishing the relation for the $\mathcal{S}$ transformation in Eq. (2.26). We also find two other actions dual to Eq. (2.13). The duality calculation is similar to those in Refs. [7, 53]. We start from the action in Eq. (2.13) (and leave the flatness constraints of the background fields implicit). First, using the Poisson summation formula, we sum over $n_\mu$, which simply replaces $a_\mu \to 2\pi k_\mu/N$, where $k_\mu \in \mathbb{Z}$, so we obtain

$$\begin{aligned}
S = {} & \frac{1}{4e^2}\left(\frac{2\pi}{N}\right)^2 \sum_P (\Delta_\mu k_\nu - \Delta_\nu k_\mu - N s_{\mu\nu} - B_{\mu\nu})^2 + \frac{1}{2g^2}\sum_{\tilde{\ell}} (\omega_\mu[A])^2 - iN\sum_R n(R)\varphi(R) \\
& + \frac{iN\Theta}{8\pi^2}\left(\frac{2\pi}{N}\right)\sum_{r,R} \varepsilon_{\mu\nu\lambda}\, \omega_\mu[A]\, (\Delta_\nu k_\lambda - \Delta_\lambda k_\nu - N s_{\nu\lambda} - B_{\nu\lambda}).
\end{aligned} \tag{B.1}$$

We then dualize the $\mathbb{Z}_N$ gauge field by making the replacement

$$\Delta_\mu k_\nu - \Delta_\nu k_\mu - N s_{\mu\nu} - B_{\mu\nu} \to \mathcal{F}_{\mu\nu} \tag{B.2}$$

for some new dynamical field $\mathcal{F}_{\mu\nu} \in \mathbb{R}$ on the plaquettes. We must then also introduce a Lagrange multiplier $\tilde{v}_\mu \in \mathbb{R}$ on links of the dual lattice that implements a constraint for the replacement, Eq. (B.2). The resulting action is

$$
\begin{aligned}
S = {} & \frac{1}{4e^2}\left(\frac{2\pi}{N}\right)^2 \sum_P \mathcal{F}_{\mu\nu}^2 + \frac{1}{2g^2}\sum_{\tilde{\ell}}(\omega_\mu[A])^2 + \frac{iN\Theta}{8\pi^2}\left(\frac{2\pi}{N}\right)\sum_{r,R}\varepsilon_{\mu\nu\lambda}\,\omega_\mu[A]\,\mathcal{F}_{\nu\lambda} \\
& - iN\sum_R n(R)\varphi(R) - \frac{i}{2}\sum_{r,R}\varepsilon_{\mu\nu\lambda}\left(\Delta_\mu k_\nu - \Delta_\nu k_\mu - Ns_{\mu\nu} - B_{\mu\nu} - \mathcal{F}_{\mu\nu}\right)\tilde{v}_\lambda.
\end{aligned}
\tag{B.3}
$$

Next, we sum over $s_{\mu\nu}$, which simply replaces $\tilde{v}_\mu \to 2\pi c_\mu/N$, where $c_\mu \in \mathbb{Z}$. We also integrate out $\mathcal{F}_{\mu\nu}$, which yields

$$
\begin{aligned}
S = {} & \sum_{\tilde{\ell}}\left[\frac{1}{2g^2}(\omega_\mu[A])^2 + \frac{e^2}{2}\left(-c_\mu - \frac{N\Theta}{4\pi^2}\omega_\mu[A]\right)^2\right] - iN\sum_R n(R)\varphi(R) \\
& - \frac{2\pi i}{N}\sum_{r,R}\varepsilon_{\mu\nu\lambda}\frac{1}{2}\left(\Delta_\mu k_\nu - \Delta_\nu k_\mu - B_{\mu\nu}\right)c_\lambda.
\end{aligned}
\tag{B.4}
$$

To make this result more transparent, we can introduce new integer fields $\tilde{k}$ and $\tilde{s}_\mu$ on dual sites and dual links respectively with the gauge symmetries,

$$
\tilde{k} \to \tilde{k} + \chi, \qquad \tilde{s}_\mu \to \tilde{s}_\mu + \xi_\mu, \qquad c_\mu \to c_\mu + \Delta_\mu\chi - N\,\xi_\mu,
\tag{B.5}
$$

for $\chi,\xi_\mu \in \mathbb{Z}$. Indeed, these gauge symmetries ensure that we have not added extra degrees of freedom since $\tilde{k}$ and $\tilde{s}_\mu$ can be removed by gauge fixing $c_\mu$. The resulting action is

$$
\begin{aligned}
S = {} & \frac{e^2}{2}\sum_{\tilde{\ell}}\left(\Delta_\mu\tilde{k} - N\tilde{s}_\mu - c_\mu - \frac{N\Theta}{4\pi^2}\omega_\mu[A]\right)^2 - iN\sum_R n(R)\varphi(R) + \frac{1}{2g^2}\sum_{\tilde{\ell}}(\omega_\mu[A])^2 \\
& - \frac{2\pi i}{N}\sum_{r,R}\varepsilon_{\mu\nu\lambda}\left(k_\mu\,\Delta_\nu c_\lambda - \frac{1}{2}B_{\mu\nu}\,c_\lambda\right).
\end{aligned}
\tag{B.6}
$$

We then replace $2\pi\tilde{k}/N$ with a real field $\tilde{\varphi} \in \mathbb{R}$ and introduce $\tilde{n} \in \mathbb{Z}$ on dual sites to constrain $\tilde{\varphi} \in \frac{2\pi}{N}\mathbb{Z}$. We obtain

$$
\begin{aligned}
S = {} & \frac{1}{2}\left(\frac{eN}{2\pi}\right)^2\sum_{\tilde{\ell}}\left(\tilde{\omega}_\mu[c] - \frac{\Theta}{2\pi}\omega_\mu[A]\right)^2 + \frac{1}{2g^2}\sum_{\tilde{\ell}}(\omega_\mu[A])^2 \\
& - \frac{2\pi i}{N}\sum_{r,R}\varepsilon_{\mu\nu\lambda}\left(k_\mu\,\Delta_\nu c_\lambda - \frac{1}{2}B_{\mu\nu}\,c_\lambda\right) - iN\sum_R[n(R)\varphi(R) + \tilde{n}(R)\tilde{\varphi}(R)],
\end{aligned}
\tag{B.7}
$$

where $\tilde{\omega}_\mu[c]$ is as defined in Eq. (2.22). From Eq. (B.7), we see that our spin-gauge model is equivalent to two coupled $\mathbb{Z}_N$ spin models, but the $\mathbb{Z}_N$ zero-form symmetry of one of the spin models is gauged, and the associated $\mathbb{Z}_N$ gauge field is constrained to be locally flat. The remainder of the calculation entails dualizing $\varphi$ into a gauge field $\tilde{a}_\mu$. This computation is analogous to the one above and leads to Eq. (2.21).

Following a similar process, we can also dualize only the matter field $\varphi$ in the original action, Eq. (2.13), to obtain a model of two coupled gauge theories. The resulting dual action is

$$
S = \frac{1}{4}\left(\frac{gN}{2\pi}\right)^2 \sum_P \left(\tilde{f}_{\mu\nu}[b] - \frac{\Theta}{2\pi}f_{\mu\nu}[B]\right)^2 + \frac{1}{4e^2}\sum_P (f_{\mu\nu}[B])^2 - \frac{\pi i}{N}\sum_{r,R}\varepsilon_{\mu\nu\lambda}\left(\Delta_\mu \tilde{k} - A_\mu\right)b_{\nu\lambda}
$$
$$
- iN\sum_\ell \left(n_\mu\, a_\mu + \tilde{n}_\mu\, \tilde{a}_\mu\right),
$$

(B.8)

where $f_{\mu\nu}[B]$ and $\tilde{f}_{\mu\nu}[b]$ are as defined in Eq. (2.14) and Eq. (2.22) respectively. Here, we have $\tilde{n}_\mu, \tilde{s}_{\mu\nu}, b_{\mu\nu}, \tilde{k}(R) \in \mathbb{Z}$ and $\tilde{a}_\mu \in \mathbb{R}$ is the gauge field dual to $\varphi(R)$. We have obtained two coupled $\mathbb{Z}_N$ gauge theories, but one of the $\mathbb{Z}_N^{(1)}$ one-form symmetries is gauged with a locally flat $\mathbb{Z}_N$ two-form gauge field so that the global symmetry is $G = \mathbb{Z}_N^{(0)} \times \mathbb{Z}_N^{(1)}$, as in the original lattice model, Eq. (2.1).

## C  Coulomb gas and duality

To determine the phase structure of the model, Eq. (2.1), it is useful to integrate out $a_\mu$ and $\varphi$ to obtain an effective action for $n_\mu$, $n$, $m_\mu$, and $m$. In doing so, we lose some information about the topological data within each phase, but we can still learn about the phase diagram and which objects are condensed in each phase. (Here, we turn off the background fields,

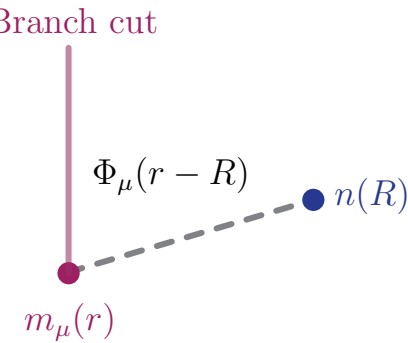

**Figure 13:** Statistical interaction in the Coulomb gas. The last term in the Coulomb gas action, Eq. (C.1), is a statistical interaction of spins and vortices. If $m_\mu(r)$ is a straight line along the Euclidean time direction and $n(R)$ is in the $xy$-plane, then $\Phi_\mu(r - R)$ is the angle between the branch cut of $m_\mu(r)$ and the shortest line from $m_\mu(r)$ to $n(R)$.

setting $A_\mu = B_{\nu\lambda} = 0$ mod $N$.) The partition function after integrating out $a_\mu$ and $\varphi$ is then

$$Z = \sum_{\{n, n_\mu, m, m_\mu\}} \delta(\Delta_\mu n_\mu)\, \delta(\Delta_\mu m_\mu)\, e^{-S_{\mathrm{CG}}[n, n_\mu, a_\mu, \varphi, s_\mu, s_{\mu\nu}]},$$

$$S_{\mathrm{CG}} = \frac{2\pi^2}{g^2} \sum_{r, r'} m_\mu(r)\, \mathcal{K}(r - r')\, m_\mu(r') + \frac{2\pi^2}{e^2} \sum_{R, R'} m(R)\, \mathcal{K}(R - R')\, m(R')$$

$$+ \frac{e^2 N^2}{2} \sum_{r, r'} \left( n_\mu(r) + \frac{\Theta}{2\pi}\, m_\mu(r) \right) \mathcal{K}(r - r') \left( n_\mu(r') + \frac{\Theta}{2\pi}\, m_\mu(r') \right) \tag{C.1}$$

$$+ \frac{g^2 N^2}{2} \sum_{R, R'} \left( n(R) + \frac{\Theta}{2\pi}\, m(R) \right) \mathcal{K}(R - R') \left( n(R') + \frac{\Theta}{2\pi}\, m(R') \right)$$

$$- iN \sum_{R, r} m(R)\, \Phi_\mu(R - r)\, n_\mu(r) + iN \sum_{r, R} m_\mu(r)\, \Phi_\mu(r - R)\, n(R).$$

Here, $\mathcal{K}(r) \sim 1/4\pi r$ is the lattice Green function for the Laplacian $-\Delta^2$ in 3d, and we define

$$\Phi_\mu(r - R) = 2\pi\, \varepsilon_{\mu\nu\lambda}\, u_\nu\, (u \cdot \Delta)^{-1} \Delta_\lambda^{(r)} \mathcal{K}(r - R), \tag{C.2}$$

where $u_\mu$ is a unit vector. We refer to the above action $S_{\mathrm{CG}}$ as the Coulomb gas action since the first three lines of $S_{\mathrm{CG}}$ represent Coulomb interactions between the loops or local operators. The terms in the last line of Eq. (C.1) are statistical interactions. For example, if $m_\mu$ is a straight line along the Euclidean time direction, then the last term in the Coulomb gas action is the angle in the $xy$-plane between the branch cut of the vortex (determined by the unit vector $u_\mu$) and the shortest line from $m_\mu$ to $n$ (See Figure 13).

One useful aspect of the Coulomb gas action is that it makes certain invariances of the partition function manifest. To make these dualities more explicit, we define the complex

coupling constant $\tau$ and the coupling ratio $\kappa$ as in Eq. (2.24). In these variables, the Coulomb gas action takes the form

$$
\begin{aligned}
S_{\text{CG}} = {} & \frac{2\pi i N \kappa}{\tau - \bar{\tau}} \sum_{r, r'} [n_\mu(r) + \tau\, m_\mu(r)]\, \mathcal{K}(r - r')\, [n_\mu(r') + \bar{\tau}\, m_\mu(r')] \\
& + \frac{2\pi i N}{\kappa(\tau - \bar{\tau})} \sum_{R, R'} [n(R) + \tau\, m(R)]\, \mathcal{K}(R - R')\, [n(R') + \bar{\tau}\, m(R')] \\
& - iN \sum_{R, r} m(R)\, \Phi_\mu(R - r)\, n_\mu(r) + iN \sum_{r, R} m_\mu(r)\, \Phi_\mu(r - R)\, n(R),
\end{aligned}
\tag{C.3}
$$

where $\bar{\tau}$ is the complex conjugate of $\tau$. The Coulomb gas partition function is then manifestly invariant under

$$
\begin{aligned}
\mathbf{S}: && \tau &\mapsto -\frac{1}{\tau}, && (n_\mu, m_\mu) &\mapsto (-m_\mu, n_\mu), && (n, m) &\mapsto (-m, n), \\
\mathbf{T}: && \tau &\mapsto \tau + 1, && (n_\mu, m_\mu) &\mapsto (n_\mu - m_\mu, m_\mu), && (n, m) &\mapsto (n - m, m),
\end{aligned}
\tag{C.4}
$$

while keeping $\kappa$ fixed. Invariance under $\mathbf{T}$ is simply the $2\pi$ periodicity of $\Theta$, and $\mathbf{S}$ is the analogue of Kramers-Wannier duality in (1+1)d or electromagnetic duality in (3+1)d. Together, $\mathbf{S}$ and $\mathbf{T}$ generate the modular group $\text{PSL}(2, \mathbb{Z})$, which is the set of mappings,

$$
\tau \mapsto \frac{a\tau + b}{c\tau + d},
\tag{C.5}
$$

such that $a, b, c, d \in \mathbb{Z}$ and $ad - bc \neq 0$. These modular transformations are reminiscent of those derived in Ref. [41].

These transformations, Eq. (C.4), are less precise analogues of $\mathcal{S}$ and $\mathcal{T}$ in Eq. (2.26) since they ignore topological properties. However, they are still useful in determining what objects condense in a given phase. The transformations in Eq. (C.4) imply that the $\mathcal{S}$ and $\mathcal{T}$ transformations act on charges of operators as in Eq. (2.28). As discussed in Section 2.3, these transformations allow us to relate the charges of the condensing objects in one phase to another.

# D Effective field theories at weak coupling

## D.1 $\mathbb{Z}_N$ clock model: Ordered phase

Let us review the effective field theories for the nontrivial phases at $\Theta = 0$. We begin with the ordered phase of the $\mathbb{Z}_N$ clock model. The part of the action in Eq. (2.13) for the spin

model is

$$S_{\text{spin}} = \frac{1}{2g^2} \sum_{\tilde{\ell}} (\Delta_\mu \varphi - 2\pi s_\mu - 2\pi A_\mu/N)^2 - iN \sum_R n(R)\varphi(R). \tag{D.1}$$

Summing over $n(R)$ effectively constrains $\varphi(R) \to 2\pi \tilde{k}(R)/N$, where $\tilde{k}(R) \in \mathbb{Z}$, so we have

$$S_{\text{spin}} = \frac{1}{2} \left( \frac{2\pi}{Ng} \right)^2 \sum_{\tilde{\ell}} (\Delta_\mu \tilde{k} - Ns_\mu - A_\mu)^2. \tag{D.2}$$

The ordered phase arises in the small $g^2$ limit. In the strict $g^2 \to 0$ limit, we obtain the constraint,

$$(\Delta_\mu \tilde{k} - A_\mu) \in N\mathbb{Z}. \tag{D.3}$$

We can then implement this constraint using a dynamical Lagrange multiplier $b_{\mu\nu} \in \mathbb{Z}$, which is defined on plaquettes of the direct lattice. The effective action introducing the constraint is

$$S_{\text{SSB}} = \frac{\pi i}{N} \sum_{r,R} \varepsilon_{\mu\nu\lambda} b_{\mu\nu} (\Delta_\lambda \tilde{k} - A_\lambda). \tag{D.4}$$

This effective action simply constrains the $\mathbb{Z}_N^{(0)}$ order parameter to take a nonzero expectation value.

The corresponding continuum effective field theory for the ordered phase may be read from Eq. (D.4). The effective theory depends on a dynamical $2\pi$ periodic scalar $\varphi$, a dynamical $U(1)$ two-form gauge field $b_{\mu\nu}$, and a background $\mathbb{Z}_N$ one-form gauge field $A_\mu$, which are related to the variables in Eq. (D.4) by

$$\frac{2\pi \tilde{k}}{N} \to \varphi, \qquad \frac{2\pi b_{\mu\nu}}{N} \to b_{\mu\nu}, \qquad \frac{2\pi A_\mu}{N} \to A_\mu, \tag{D.5}$$

where we have reused some notation because of the natural corresspondence with the lattice degrees of freedom. The low energy effective field theory is then

$$S_{\text{SSB}} = \frac{iN}{2\pi} \int b \wedge (d\varphi - A). \tag{D.6}$$

This effective theory has a $\mathbb{Z}_N^{(0)}$ symmetry,

$$\varphi \to \varphi + \frac{2\pi}{N}, \tag{D.7}$$

which acts nontrivially on a local operator,

$$V(\mathcal{P}) = e^{i\varphi(\mathcal{P})}, \tag{D.8}$$

at point $\mathcal{P}$. This symmetry is probed by the background field $A_\mu$. There are also operators defined on surfaces $\Sigma$,

$$U(\Sigma) = \exp\left(i \oint_\Sigma b\right). \tag{D.9}$$

If $\Sigma$ is a fixed timeslice, then $U(\Sigma)$ is interpreted as a symmetry operator for the $\mathbb{Z}_N^{(0)}$ symmetry. If we canonically quantize this theory, we obtain the clock and shift algebra,

$$V U = e^{2\pi i/N} U V, \qquad V^N = U^N = 1. \tag{D.10}$$

If $\Sigma$ stretches along the time direction, then $U(\Sigma)$ represents a domain wall for the $\mathbb{Z}_N^{(0)}$ symmetry. The effective field theory, Eq. (D.6), simply signals the spontaneous breaking of the $\mathbb{Z}_N^{(0)}$ symmetry. We can also interpret Eq. (D.10) to mean that the local operator $V$ acts on the surface operator $U$ with a $\mathbb{Z}_N$ two-form global symmetry. This two-form symmetry is emergent and appears deep within the ordered phase where domain walls are suppressed. The emergence of generalized symmetries within conventional ordered phases is generic [84].

## D.2 $\mathbb{Z}_N$ gauge theory: Deconfined phase

Similarly, we can derive the precise topologogical quantum field theory (TQFT) of the deconfined phase of the $\mathbb{Z}_N$ gauge theory by a calculation analogous to what we did above for the ordered phase of the $\mathbb{Z}_N$ clock model. Using the same notation as in Eq. (2.13), we start with the $\mathbb{Z}_N$ lattice gauge theory,

$$S_{\text{gauge}} = \frac{1}{4e^2} \sum_P (\Delta_\mu a_\nu - \Delta_\nu a_\mu - 2\pi s_{\mu\nu} - 2\pi B_{\mu\nu}/N)^2 - iN \sum_\ell n_\mu\, a_\mu, \tag{D.11}$$

By the Poisson summation formula, summing over $n_\mu$ forces the constraint $a_\mu \to 2\pi k_\mu/N$ where $k_\mu \in \mathbb{Z}$. In analogy with the calculation in Section D.1, we take the $e^2 \to 0$ limit, which constrains

$$\Delta_\mu k_\nu - \Delta_\nu k_\mu - B_{\mu\nu} \in N\mathbb{Z}. \tag{D.12}$$

Introducing a Lagrange multiplier $c_\mu \in \mathbb{Z}$ for this constraint, we obtain

$$S_{\text{BF}} = \frac{\pi i}{N} \sum_{r,R} \varepsilon_{\mu\nu\lambda}\, c_\mu\, (\Delta_\nu k_\lambda - \Delta_\lambda k_\nu - B_{\nu\lambda}). \tag{D.13}$$

This action is a lattice regularization of (2+1)d BF theory at level $N$, encoding the topological order of the deconfined phase.

The analogous TQFT depends on two dynamical $U(1)$ one-form gauge fields and a background $\mathbb{Z}_N$ two-form gauge field $B_{\mu\nu}$, which are related to the lattice variables of Eq. (D.13) by

$$\frac{2\pi k_\mu}{N} \to a_\mu, \qquad \frac{2\pi c_\mu}{N} \to c_\mu, \qquad \frac{2\pi B_{\mu\nu}}{N} \to B_{\mu\nu}. \tag{D.14}$$

The TQFT is BF theory,

$$S_{\mathrm{BF}} = \frac{iN}{2\pi} \int c \wedge (da - B). \tag{D.15}$$

The $\mathbb{Z}_N^{(1)}$ symmetry of the lattice gauge theory acts on $a_\mu$ as

$$a \to a + \frac{\eta}{N}, \tag{D.16}$$

where $\eta$ is a locally flat connection, $d\eta = 0$, with quantized cycles $\oint \eta \in 2\pi\mathbb{Z}$. This symmetry is broken at low energies, leading to the same topological order as the $\mathbb{Z}_N$ toric code. For a loop $\Gamma$ in spacetime, we can define the loop operators,

$$W_a(\Gamma) = \exp\left( i \oint_\Gamma a \right), \qquad W_c(\Gamma) = \exp\left( i \oint_\Gamma c \right), \tag{D.17}$$

which have correlation functions,

$$\langle W_a(\Gamma) W_c(\Gamma') \rangle = \exp\left( \frac{2\pi i}{N} \Phi_{\mathrm{link}}(\Gamma, \Gamma') \right), \tag{D.18}$$

where $\Phi_{\mathrm{link}}(\Gamma, \Gamma')$ is the linking number for loops $\Gamma$ and $\Gamma'$. The operators $W_a(\Gamma)$ and $W_c(\Gamma')$ represent the worldlines of two different species of anyons with bosonic self-statistics but fractional mutual statistics.

## D.3 SPT stacking

Here, we explicitly show that the phases of the lattice model, Eq. (2.1) at $\Theta = 0$ with $\mathbb{Z}_N^{(0)}$ symmetry breaking or $\mathbb{Z}_N^{(1)}$ symmetry breaking (or both) are invariant under a $\mathcal{T}$ transformation. Consider the phase that appears at $\Theta = 0$ in the limit $ge \to 0$. In this phase, the $G = \mathbb{Z}_N^{(0)} \times \mathbb{Z}_N^{(1)}$ symmetry is broken completely. Combining Eq. (D.4) and Eq. (D.13), this phase is effectively described by the lattice action,

$$S = \frac{2\pi i}{N} \sum_{r,R} \varepsilon_{\mu\nu\lambda}\, b_{\mu\nu}\, \Delta_\lambda(\tilde{k} - A_\mu) + \frac{\pi i}{N} \sum_{r,R} \varepsilon_{\mu\nu\lambda}\, c_\mu\, (\Delta_\nu k_\lambda - \Delta_\lambda k_\nu - B_{\nu\lambda}). \tag{D.19}$$

Under a $\mathcal{T}^p$ transformation, the action is mapped to

$$S = \frac{\pi i}{N} \sum_{r,R} \varepsilon_{\mu\nu\lambda}\, b_{\mu\nu}\, \Delta_\lambda(\tilde{k} - A_\mu) + \frac{\pi i}{N} \sum_{r,R} \varepsilon_{\mu\nu\lambda}\, c_\mu\, (\Delta_\nu k_\lambda - \Delta_\lambda k_\nu - B_{\nu\lambda}) + \frac{i\pi p}{N} \sum_{r,R} \varepsilon_{\mu\nu\lambda}\, A_\mu B_{\nu\lambda}. \tag{D.20}$$

Shifting $b_{\mu\nu} \to b_{\mu\nu} + p\,B_{\mu\nu}$ leaves us with

$$S = \frac{\pi i}{N} \sum_{r,R} \varepsilon_{\mu\nu\lambda}\, b_{\mu\nu}\, \Delta_\lambda(\tilde{k} - A_\mu) + \frac{\pi i}{N} \sum_{r,R} \varepsilon_{\mu\nu\lambda}\, c_\mu\, (\Delta_\nu k_\lambda - \Delta_\lambda k_\nu - B_{\nu\lambda}) - \frac{i\pi p}{N} \sum_{r,R} \varepsilon_{\mu\nu\lambda}\, \tilde{k}\, \Delta_\mu B_{\nu\lambda},$$

(D.21)

where we did a summation by parts in the last term to move the lattice derivative onto $B_{\mu\nu}$. Because $B_{\mu\nu}$ is locally flat, the last term of Eq. (D.21) is an integer multiple of $2\pi i$ and therefore may be discarded. Hence, in this weak coupling phase, the $\mathcal{T}^p$ transformation does not affect any physics. A similar argument also shows that either intermediate phase in which $\mathbb{Z}_N^{(0)}$ or $\mathbb{Z}_N^{(1)}$ is broken but not the other is also invariant under $\mathcal{T}^p$.

# E  Axion model calculations

## E.1  Axion SPT model: Effective field theory

Here we derive the lattice version of Eq. (9.8), which is the effective field theory of Eq. (9.3) deep within the phase realized at large $g^2$, $e^2$, and $J$. The $g^2 \to \infty$ and $e^2 \to \infty$ limits are straightforward. The $J \to \infty$ limit results in the constraint,

$$[\tilde{k}(\mathcal{R}) - \tilde{k}(\mathcal{R}')] \in N\mathbb{Z}$$

(E.1)

for any nearest neighbors $\mathcal{R}$ and $\mathcal{R}'$, which implies that the $(\mathbb{Z}_N^{(0)})_{\text{axion}}$ global symmetry is spontaneously broken. We use a Lagrange multiplier $\beta \in \mathbb{Z}$, which lives on triangles intersecting links between nearest neighbors of the sites $\mathcal{R}$ on which the $\tilde{k}(\mathcal{R})$ live (i.e., $\beta$ is defined on the blue triangles depicted in Figure 11.) The effective action becomes

$$S_{\text{eff}} = \frac{i\pi}{N} \sum_{r,R,\mathcal{R}} \varepsilon_{\mu\nu\lambda}\, \tilde{k}\, (\Delta_\mu k - A_\mu)\, (\Delta_\nu k_\lambda - \Delta_\lambda k_\nu - B_{\nu\lambda}) - \frac{i\pi}{N} \sum_{\langle \mathcal{R},\mathcal{R}' \rangle} \beta\, [\tilde{k}(\mathcal{R}) - \tilde{k}(\mathcal{R}')]$$
$$+ \frac{i\pi}{N} \sum_{r,R} \varepsilon_{\mu\nu\lambda}\, \left( -j\, \Delta_\mu B_{\nu\lambda} + 2\, \tilde{k}_\mu\, \Delta_\nu A_\lambda \right).$$

(E.2)

Summing by parts, we obtain

$$S_{\text{eff}} = -\frac{i\pi}{N} \sum_{r,R,\mathcal{R}} \varepsilon_{\mu\nu\lambda}\, \left[ k(\Delta_\mu k_\nu - \Delta_\nu k_\mu - B_{\mu\nu})\Delta_\lambda \tilde{k} + 2A_\mu\, k_\nu\, \Delta_\lambda \tilde{k} - k\, \tilde{k}\, \Delta_\mu B_{\nu\lambda} + 2k_\mu\, \tilde{k}\, \Delta_\nu A_\lambda \right]$$
$$+ \frac{i\pi}{N} \sum_{r,R,\mathcal{R}} \varepsilon_{\mu\nu\lambda}\, \tilde{k}\, A_\mu\, B_{\nu\lambda} + \frac{i\pi}{N} \sum_{r,R} \varepsilon_{\mu\nu\lambda}\, \left( -j\, \Delta_\mu B_{\nu\lambda} + 2\, \tilde{k}_\mu\, \Delta_\nu A_\lambda \right)$$
$$- \frac{i\pi}{N} \sum_{\langle \mathcal{R},\mathcal{R}' \rangle} \beta\, [\tilde{k}(\mathcal{R}) - \tilde{k}(\mathcal{R}')].$$

(E.3)

Because $A_\mu$ and $B_{\mu\nu}$ are locally flat and $\tilde{k}$ obeys the constraint, Eq. (E.1), this action simplifies to

$$S_{\text{eff}} = -\frac{i\pi}{N} \sum_{\langle \mathcal{R}, \mathcal{R}' \rangle} \beta \left[ \tilde{k}(\mathcal{R}) - \tilde{k}(\mathcal{R}') \right] + \frac{i\pi}{N} \sum_{r, R} \varepsilon_{\mu\nu\lambda} \, \tilde{k} \, A_\mu \, B_{\nu\lambda}$$
$$+ \frac{i\pi}{N} \sum_{r, R, \mathcal{R}} \varepsilon_{\mu\nu\lambda} \left( 2 \, \tilde{k}_\mu \, \Delta_\nu A_\lambda - j \, \Delta_\mu B_{\nu\lambda} \right),$$

(E.4)

which is the lattice version of the action in Eq. (9.8). The correspondence between lattice variables and the fields in Eq. (9.8) is

$$\frac{2\pi \tilde{k}}{N} \to \theta, \qquad \frac{2\pi \beta}{N} \to \beta_{\mu\nu}, \qquad \frac{2\pi j}{N} \to \varphi, \qquad \frac{2\pi \tilde{k}_\mu}{N} \to a_\mu,$$
$$\frac{2\pi A_\mu}{N} \to A_\mu, \qquad \frac{2\pi B_{\mu\nu}}{N} \to B_{\mu\nu}.$$

(E.5)

This derivation corroborates the physical argument in the main text that Eq. (9.8) is the correct field theory to describe the phase at large $g^2$, $e^2$, and $J$. For the gauged axion model, discussed in Section 9.2, the derivation on the lattice of the field theory, Eq. (9.13), which describes the phase with non-invertible symmetry breaking is also analogous to the calculation above.

## E.2 Non-invertible symmetry in the lattice model

In this appendix, we demonstrate that the lattice model introduced in Section 9.2 has a non-invertible symmetry that descends from the $(\mathbb{Z}_N^{(0)})_{\text{axion}}$ global symmetry of the model in Section 9.1. Similar calculations were completed in Refs. [19, 21]. First, we consider how the partition function $\mathcal{Z}_{\text{axion}}$ transforms under the transformation $\mathcal{S}\mathcal{T}^{-p}\mathcal{S}$, where $p \in \mathbb{Z}$. Although $\tau = \frac{\Theta}{2\pi} + i\frac{2\pi}{Nge}$ is no longer a constant in this model since we promoted $\Theta \to N\theta$ to a dynamical variable, based on Eq. (2.26), we can still define $\mathcal{S}$ and $\mathcal{T}$ transformations as

$$\mathcal{T}(\mathcal{Z}_{\text{axion}}[A_\mu, B_{\mu\nu}]) = \mathcal{Z}_{\text{axion}}[A_\mu, B_{\mu\nu}] \, e^{-\frac{\pi i}{N} \sum_{r, R} \varepsilon_{\mu\nu\lambda} A_\mu B_{\nu\lambda}},$$
$$\mathcal{S}(\mathcal{Z}_{\text{axion}}[A_\mu, B_{\mu\nu}]) = \sum_{\{a_\mu, b_{\mu\nu}\}} \mathcal{Z}_{\text{axion}}[a_\mu, b_{\mu\nu}] \, e^{-\frac{\pi i}{N} \sum_{r, R} \varepsilon_{\mu\nu\lambda} (a_\mu B_{\nu\lambda} + b_{\mu\nu} A_\lambda)},$$

(E.6)

rather than as actions on $\tau$. We also define a transformation $\mathcal{C}$ that simply changes the sign of the background fields,

$$\mathcal{C}(\mathcal{Z}_{\text{axion}}[A_\mu, B_{\mu\nu}]) = \mathcal{Z}_{\text{axion}}[-A_\mu, -B_{\mu\nu}].$$

(E.7)

After a $\mathcal{CST}^{-p}\mathcal{S}$ transformation, the partition function of Eq. (9.10) is mapped to

$$
\begin{aligned}
&\mathcal{CST}^{-p}\mathcal{S}(\mathcal{Z}_{\text{axion}}[A_\mu, B_{\mu\nu}]) \\
&= \sum_{\substack{\{c_\mu, b_{\mu\nu}, \tilde{a}_\mu, \\ \tilde{b}_{\mu\nu}, \tilde{c}_\mu, \tilde{\beta}_{\mu\nu}\}}} Z_{\text{axion}}[c_\mu, b_{\mu\nu}]\, e^{-\frac{i\pi}{N}\sum_{r,R}\varepsilon_{\mu\nu\lambda}\left(c_\mu\,\tilde{b}_{\nu\lambda}+b_{\mu\nu}\,\tilde{a}_\lambda+\tilde{a}_\mu\,\tilde{\beta}_{\nu\lambda}+\tilde{b}_{\mu\nu}\,\tilde{c}_\lambda-p\,\tilde{c}_\mu\,\tilde{\beta}_{\nu\lambda}-\tilde{c}_\mu\,B_{\nu\lambda}-\tilde{\beta}_{\mu\nu}\,A_\lambda\right)}, \quad \text{(E.8)}
\end{aligned}
$$

where $c_\mu, \tilde{a}_\mu, \tilde{c}_\mu \in \mathbb{Z}$ are locally flat dynamical $\mathbb{Z}_N$ one-form gauge fields, $b_{\mu\nu}, \tilde{b}_{\mu\nu}, \tilde{\beta}_{\mu\nu} \in \mathbb{Z}$ are locally flat dynamical $\mathbb{Z}_N$ two-form gauge fields, $A_\mu \in \mathbb{Z}$ is a locally flat background $\mathbb{Z}_N$ one-form gauge field, and $B_{\mu\nu}$ is a locally flat background $\mathbb{Z}_N$ two-form gauge field.

Summing over $\tilde{b}_{\mu\nu}$ and $\tilde{a}_\mu$ gives the constraints $\tilde{c}_\mu = -c_\mu$ mod $N$ and $\tilde{\beta}_{\mu\nu} = -b_{\mu\nu}$ mod $N$ respectively. After imposing the constraints, we find

$$
\mathcal{CST}^{-p}\mathcal{S}(\mathcal{Z}_{\text{axion}}[A_\mu, B_{\mu\nu}]) = \sum_{\{c_\mu, b_{\mu\nu}\}} Z_{\text{axion}}[c_\mu, b_{\mu\nu}]\, e^{-\frac{i\pi}{N}\sum_{r,R}\varepsilon_{\mu\nu\lambda}(-p\,c_\mu\,b_{\nu\lambda}+c_\mu\,B_{\nu\lambda}+b_{\mu\nu}\,A_\lambda)}. \quad \text{(E.9)}
$$

The $\mathcal{CST}^{-p}\mathcal{S}$ transformation can then be canceled if we also act on $\theta(\mathcal{R})$ with

$$
\theta(\mathcal{R}) \rightarrow \theta(\mathcal{R}) + \frac{2\pi p}{N}, \quad \text{(E.10)}
$$

which was a $(\mathbb{Z}_N^{(0)})_{\text{axion}}$ symmetry transformation in the ungauged theory, Eq. (9.3). Thus, the combination of the transformation, Eq. (E.10), on $\theta(\mathcal{R})$ and $\mathcal{CST}^{-p}\mathcal{S}$ leaves the theory invariant.

This combined operation may be regarded as a non-invertible symmetry, and the symmetry operator may be constructed using the half-gauging procedure [20]. Let $y$ denote a coordinate of the direct lattice so that $y = 0$ defines a plane of plaquettes of the direct lattice. We divide our spacetime into two regions, $y \leq 0$ and $y > 0$. We use the action for our gauged axion model, Eq. (9.10), in the region $y > 0$. For $y \leq 0$, we place the same model but acted on with the non-invertible symmetry. To be precise, we act with $\mathcal{CST}^{-p}\mathcal{S}$ and take $\theta(\mathcal{R}) \rightarrow \theta(\mathcal{R}) + 2\pi p/N$ for every $\mathcal{R}$ on a plaquette of the direct lattice where $y \leq 0$. As established above, the combination of Eq. (E.10) and $\mathcal{CST}^{-p}\mathcal{S}$ leaves our model invariant. A surface defect for the non-invertible symmetry will then be left behind, which intersects all the links between the axion sites in the plane $y = 0$ and their nearest neighbors in the plane $y = 1/2$ (see Figure 11). In the above calculation, when we performed the $\mathcal{CST}^{-p}\mathcal{S}$ transformation, we left implicit the constraints that the $\mathbb{Z}_N$ gauge fields are flat. Here, we will need to keep these constraints explicit since they will be important when there are defects, but for simplicity, we turn off background fields.

Let $\Sigma$ denote the surface between $y = 0$ and $y = 1/2$ where the defect lies (i.e., the blue surface in Figure 11). For brevity, we define the action,

$$
\begin{aligned}
S_0 = & \frac{1}{4}\left(\frac{2\pi}{Ne}\right)^2 \sum_P (\Delta_\mu k_\nu - \Delta_\nu k_\mu - N s_{\mu\nu} - b_{\mu\nu})^2 \\
& + \frac{1}{2}\left(\frac{2\pi}{Ng}\right)^2 \sum_{\tilde{\ell}} (\Delta_\mu k - N s_\mu - c_\mu)^2 - J \sum_{\langle \mathcal{R}, \mathcal{R}' \rangle \notin \Sigma} \cos\left(\frac{2\pi}{N}[\tilde{k}(\mathcal{R}) - \tilde{k}(\mathcal{R}')]\right) \\
& + \frac{i\pi}{N} \sum_{r, R, \mathcal{R}} \varepsilon_{\mu\nu\lambda}\, \tilde{k}(\mathcal{R})\, (\Delta_\mu k - c_\mu)\, (\Delta_\nu k_\lambda - \Delta_\lambda k_\nu - b_{\nu\lambda}) \\
& + \frac{i\pi}{N} \sum_{r, R} \varepsilon_{\mu\nu\lambda}\, (-\tilde{m}\, \Delta_\mu b_{\nu\lambda} + 2\, \tilde{m}_\mu\, \Delta_\nu c_\lambda)\,,
\end{aligned}
\tag{E.11}
$$

where $\tilde{m}, \tilde{m}_\mu \in \mathbb{Z}$ are integers that enforce the local flatness of $b_{\mu\nu}$ and $c_\mu$ respectively. The sum over $\langle \mathcal{R}, \mathcal{R}' \rangle \notin \Sigma$ means that we include nearest neighbors $\mathcal{R}$ and $\mathcal{R}'$ connected by links that do not intersect $\Sigma$. Let $X_-$ denote the half-space $y \leq 0$, where we act with the non-invertible symmetry. The full action is

$$
\begin{aligned}
S = & S_0 - J \sum_{\langle \mathcal{R}, \mathcal{R}' \rangle \in \Sigma} \cos\left(\frac{2\pi}{N}[\tilde{k}(\mathcal{R}) - \tilde{k}(\mathcal{R}')] + \frac{2\pi p}{N}\right) \\
& + \frac{i\pi p}{N} \sum_{\{r, R\} \in X_-} \varepsilon_{\mu\nu\lambda}\, (\Delta_\mu k - c_\mu)(\Delta_\nu k_\lambda - \Delta_\lambda k_\nu - b_{\nu\lambda}) \\
& + \frac{i\pi}{N} \sum_{\{r, R\} \in X_-} \varepsilon_{\mu\nu\lambda}\left(\tilde{a}_\mu\, b_{\nu\lambda} + c_\mu\, \tilde{b}_{\nu\lambda} + \tilde{a}_\mu\, \tilde{\beta}_{\nu\lambda} + \tilde{b}_{\mu\nu}\, \tilde{c}_\lambda - p\, \tilde{c}_\mu\, \tilde{\beta}_{\nu\lambda}\right) \\
& + \frac{i\pi}{N} \sum_{\{r, R\} \in X_-} \varepsilon_{\mu\nu\lambda}\left(2\, \tilde{a}_\mu\, \Delta_\nu \alpha_\lambda + \tilde{b}_{\mu\nu}\, \Delta_\lambda \phi + 2\, \tilde{c}_\mu\, \Delta_\nu \tilde{\alpha}_\lambda + \tilde{\beta}_{\mu\nu}\, \Delta_\lambda \tilde{\phi}\right),
\end{aligned}
\tag{E.12}
$$

where $\phi, \tilde{\phi} \in \mathbb{Z}$ are on sites $R$ of the dual lattice, $\tilde{a}_\mu, \tilde{c}_\mu, \alpha_\mu, \tilde{\alpha}_\mu \in \mathbb{Z}$ are on links $\ell$ of the direct lattice, and $\tilde{b}_{\mu\nu}, \tilde{\beta}_{\mu\nu} \in \mathbb{Z}$ are on plaquettes of the direct lattice. In the second term of Eq. (E.12) we take $\mathcal{R}$ to lie along the plane $y = 0$ and $\mathcal{R}'$ to lie along the plane $y = 1/2$. Summing over $\tilde{b}_{\mu\nu}$ and $\tilde{a}_\mu$ gives the respective constraints,

$$
\tilde{c}_\mu = -(\Delta_\mu \phi + c_\mu), \qquad \tilde{\beta}_{\mu\nu} = -(\Delta_\mu \alpha_\nu - \Delta_\nu \alpha_\mu + b_{\mu\nu}).
\tag{E.13}
$$

Using these constraints, we obtain

$$
\begin{aligned}
S = S_0 &- J \sum_{\langle \mathcal{R}, \mathcal{R}' \rangle \in \Sigma} \cos\left( \frac{2\pi}{N} [\tilde{k}(\mathcal{R}) - \tilde{k}(\mathcal{R}') + p] \right) \\
&+ \frac{i\pi p}{N} \sum_{\{r, R\} \in X_-} \varepsilon_{\mu\nu\lambda} (\Delta_\mu k - c_\mu)(\Delta_\nu k_\lambda - \Delta_\lambda k_\nu - b_{\nu\lambda}) \\
&- \frac{i\pi p}{N} \sum_{\{r, R\} \in X_-} \varepsilon_{\mu\nu\lambda} (\Delta_\mu \phi + c_\mu)(\Delta_\nu \alpha_\lambda - \Delta_\lambda \alpha_\nu + b_{\nu\lambda}) \\
&- \frac{i\pi}{N} \sum_{\{r, R\} \in X_-} \varepsilon_{\mu\nu\lambda} \left( 2 (\Delta_\mu \phi + c_\mu) \Delta_\nu \tilde{\alpha}_\lambda + (\Delta_\mu \alpha_\nu - \Delta_\nu \alpha_\mu + b_{\mu\nu}) \Delta_\lambda \tilde{\phi} \right).
\end{aligned}
\tag{E.14}
$$

If we sum by parts and take $\tilde{m} \to \tilde{m} + p(\phi + k) + \tilde{\phi}$ at every dual site $R$ for $y < 0$ and $\tilde{m}_\mu \to \tilde{m}_\mu + p(\alpha_\mu + k_\mu) + \tilde{\alpha}_\mu$ at every link $\ell$ for $y \leq 0$, we arrive at the original lattice action everywhere except at $y = 0$ and $y = 1/2$. We are left with a defect,

$$
\begin{aligned}
S_d = &- J \sum_{\langle \mathcal{R}, \mathcal{R}' \rangle \in \Sigma} \cos\left( \frac{2\pi [\tilde{k}(\mathcal{R}) - \tilde{k}(\mathcal{R}') + p]}{N} \right) + \frac{2\pi i p}{N} \sum_{\{r, R\} \in \Sigma} \varepsilon_{\mu\nu} \left( k \Delta_\mu k_\nu - c_\mu k_\nu - \frac{1}{2} k b_{\mu\nu} \right) \\
&- \frac{2\pi i}{N} \sum_{\{r, R\} \in \Sigma} \varepsilon_{\mu\nu} \left( (p\phi + \tilde{\phi}) \Delta_\mu \alpha_\nu + \phi \Delta_\mu \tilde{\alpha}_\nu + \frac{1}{2} (\tilde{\phi} + p\phi) b_{\mu\nu} + (\tilde{\alpha}_\mu + p\alpha_\mu) c_\nu \right).
\end{aligned}
\tag{E.15}
$$

We then take $\tilde{\phi} \to \tilde{\phi} - p\phi$ and $\tilde{\alpha}_\mu \to \tilde{\alpha}_\mu - p a_\mu$ along the defect, which gives the defect action of Eq. (9.12). The spacetime indices along the defect are valued in imaginary time $t$ and $x$.

### E.3  Non-invertible symmetry in the field theory

We now explicitly construct the non-invertible symmetry operator, discussed in the context of the lattice model in the previous section, for the effective field theory, Eq. (9.13), to show how we can derive the non-invertible symmetry operator $\mathcal{U}_p(\Sigma)$. We again use the half-gauging method [20]. We divide our spacetime into two regions. For $y > 0$, we use the action, Eq. (9.13). For $y < 0$, we use the same theory but acted on with $\theta \to \theta + 2\pi p/N$ and $\mathcal{CST}^{-p}\mathcal{S}$. (For simplicity, though, we set background fields to zero.) Our action is then

$$
\begin{aligned}
S = &\int_{y>0} \mathcal{L}_{\text{axion}} + ip \int_{y=0} \beta + \int_{y<0} \mathcal{L}_{\text{axion}} + \frac{iN}{2\pi} \int_{y<0} \left( p\, c \wedge b + \tilde{a} \wedge b + c \wedge \tilde{b} + \tilde{a} \wedge \tilde{\beta} + \tilde{b} \wedge \tilde{c} \right) \\
&+ \frac{iN}{2\pi} \int_{y<0} \left( -p\, \tilde{c} \wedge \tilde{\beta} + \tilde{a} \wedge d\alpha + \tilde{b} \wedge d\phi + \tilde{c} \wedge d\tilde{\alpha} + \tilde{\beta} \wedge d\tilde{\phi} \right),
\end{aligned}
\tag{E.16}
$$

where we use the abbreviation,

$$\mathcal{L}_{\text{axion}} = \frac{iN}{2\pi} \left[ \theta \left( d\beta + \frac{N}{2\pi} c \wedge b \right) - \varphi \, db + a \wedge dc \right],$$ (E.17)

and where $\phi$ and $\tilde{\phi}$ are dynamical $2\pi$ periodic scalars, the fields $\tilde{a}_\mu$, $\tilde{c}_\mu$, $\alpha_\mu$, and $\tilde{\alpha}_\mu$ are dynamical $U(1)$ one-form gauge fields, and the fields $\tilde{b}_{\mu\nu}$ and $\tilde{\beta}_{\mu\nu}$ are dynamical $U(1)$ two-form gauge fields. The other fields are the same as defined in Eq. (9.13). Note that the $y = 0$ term and the $c \wedge b$ term arise from the $\theta \to \theta + 2\pi p/N$ transformation.

Integrating out $\tilde{b}_{\mu\nu}$ and $\tilde{a}_\mu$ imposes the respective constraints,

$$\tilde{c} = -(d\phi + c), \qquad \tilde{\beta} = -(d\alpha + b).$$ (E.18)

Using the constraints to integrate out $\tilde{c}_\mu$ and $\tilde{\beta}_{\mu\nu}$, we find

$$S = \int_{y>0} \mathcal{L}_{\text{axion}} + \int_{y<0} \mathcal{L}_{\text{axion}} + \frac{iN}{2\pi} \int_{y<0} \left[ -p \left( c \wedge d\alpha + b \wedge d\phi \right) - c \wedge d\tilde{\alpha} - b \wedge d\tilde{\phi} \right]$$
$$+ i \int_{y=0} \left[ p \beta + \frac{N}{2\pi} \left( -p \, \alpha \wedge d\phi - \tilde{\alpha} \wedge d\phi - \alpha \wedge d\tilde{\phi} \right) \right].$$ (E.19)

Integrating by parts, we obtain

$$S = \int_{y>0} \mathcal{L}_{\text{axion}} + \int_{y<0} \mathcal{L}_{\text{axion}} + \frac{iN}{2\pi} \int_{y<0} \left[ -p \left( \alpha \wedge dc - \phi \, db \right) - \tilde{\alpha} \wedge dc + \tilde{\phi} \, db \right]$$
$$+ i \int_{y=0} \left[ p \beta + \frac{N}{2\pi} \left( -p \, \alpha \wedge d\phi - \tilde{\alpha} \wedge d\phi - \alpha \wedge d\tilde{\phi} + p \, c \wedge \alpha - p \, b \phi + c \wedge \tilde{\alpha} - b \tilde{\phi} \right) \right].$$ (E.20)

We can perform the shifts $\varphi \to \varphi + p \phi + \tilde{\phi}$ and $a_\mu \to a_\mu + p \alpha_\mu + \tilde{\alpha}_\mu$ for $y < 0$, which results same action for $y > 0$ and $y < 0$. We are then left with a defect at $y = 0$,

$$S_{\text{defect}} = i \int_{y=0} \left[ p \beta + \frac{N}{2\pi} \left( -p \, \alpha \wedge d\phi - \tilde{\alpha} \wedge d\phi - \alpha \wedge d\tilde{\phi} + p \, c \wedge \alpha - p \, b \phi + c \wedge \tilde{\alpha} - b \tilde{\phi} \right) \right].$$ (E.21)

To make this result more transparent, we perform the shifts $\tilde{\alpha}_\mu \to \tilde{\alpha}_\mu - p \alpha_\mu$ and $\tilde{\phi} \to \tilde{\phi} - p \phi$. The action for the defect is then

$$S_{\text{defect}} = i \int_{y=0} \left[ p \beta + \frac{N}{2\pi} \left( p \, \alpha \wedge d\phi - \tilde{\alpha} \wedge d\phi - \alpha \wedge d\tilde{\phi} + c \wedge \tilde{\alpha} - b \tilde{\phi} \right) \right].$$ (E.22)

This defect is the same as $\mathcal{U}_p(\Sigma)$, defined in Eq. (9.20), demonstrating that $\mathcal{U}_p(\Sigma)$ is in fact a topological operator.

We now demonstrate the fusion rules for $\mathcal{U}_p(\Sigma)$. Fusing the operators $\mathcal{U}_{p_1}(\Sigma)$ and $\mathcal{U}_{p_2}(\Sigma)$ produces

$$\mathcal{U}_{p_1} \times \mathcal{U}_{p_2} = \int_{\phi_j, \alpha_j, \tilde{\phi}_j, \tilde{\alpha}_j} \exp\left[i \oint_\Sigma (p_1 + p_2)\,\beta + \frac{iN}{2\pi} \oint_\Sigma \left(\alpha_1\,(p_1\,d\phi_1 - d\tilde{\phi}_1) + \alpha_2\,(p_2\,d\phi_2 - d\tilde{\phi}_2)\right)\right.$$
$$\left. + \frac{iN}{2\pi} \oint_\Sigma \left(-\tilde{\alpha}_1\,d\phi_1 - \tilde{\alpha}_2\,d\phi_2 - b\,(\tilde{\phi}_1 + \tilde{\phi}_2) + c\,(\tilde{\alpha}_1 + \tilde{\alpha}_2)\right)\right],$$

(E.23)

where we use the abbreviated notation,

$$\int_{\phi_j, \alpha_j, \tilde{\phi}_j, \tilde{\alpha}_j} = \int \prod_{j=1}^{2} \mathcal{D}\phi_j \mathcal{D}\tilde{\phi}_j \mathcal{D}\alpha_j \mathcal{D}\tilde{\alpha}_j,$$

(E.24)

for the functional integrals. To simplify the result, we perform the change of variables,

$$\phi = \phi_1, \qquad \tilde{\phi} = \tilde{\phi}_1 + \tilde{\phi}_2, \qquad \phi_- = \phi_2 - \phi_1, \qquad \bar{\phi} = \tilde{\phi}_2 - p_2\,\phi_1,$$
$$\alpha = \alpha_1, \qquad \tilde{\alpha} = \tilde{\alpha}_1 + \tilde{\alpha}_2, \qquad \alpha_- = \alpha_2 - \alpha_1, \qquad \bar{\alpha} = \tilde{\alpha}_2 - p_2\,\alpha_2.$$

(E.25)

The fusion rule for the operators then becomes

$$\mathcal{U}_{p_1} \times \mathcal{U}_{p_2} = \int_{\phi, \tilde{\phi}, \alpha, \tilde{\alpha}} \exp\left[i(p_1 + p_2) \oint_\Sigma \beta + \frac{iN}{2\pi} \oint_\Sigma \left((p_1 + p_2)\alpha\,d\phi - \alpha\,d\tilde{\phi} - \tilde{\alpha}\,d\phi - b\,\tilde{\phi} + c\,\tilde{\alpha}\right)\right]$$
$$\times \int_{\phi_-, \bar{\phi}, \alpha_-, \bar{\alpha}} \exp\left(-\frac{iN}{2\pi} \oint_\Sigma \left(\alpha_-\,d\bar{\phi} + \bar{\alpha}\,d\phi_-\right)\right).$$

(E.26)

We then see that result decomposes into $\mathcal{U}_{p_1+p_2}(\Sigma)$ and two decoupled $(1+1)$d states with $\mathbb{Z}_N^{(0)}$ spontaneous symmetry breaking. We denote the $(1+1)$d $\mathbb{Z}_N^{(0)}$ symmetry breaking partition function as

$$\mathcal{Z}_N = \int_{\phi_-, \bar{\alpha}} \exp\left(-\frac{iN}{2\pi} \oint_\Sigma \bar{\alpha}\,d\phi_-\right).$$

(E.27)

We then find that the fusion rules are given by

$$\mathcal{U}_{p_1}(\Sigma) \times \mathcal{U}_{p_2}(\Sigma) = (\mathcal{Z}_N)^2\,\mathcal{U}_{p_1+p_2}(\Sigma).$$

(E.28)

Specializing to $p_1 = p$ and $p_2 = -p$, we find

$$\mathcal{U}_p(\Sigma) \times \mathcal{U}_p(\Sigma)^\dagger = \mathcal{U}_p(\Sigma) \times \mathcal{U}_{-p}(\Sigma) = (\mathcal{Z}_N)^2\,\mathcal{C}_N(\Sigma) \neq 1$$

(E.29)

so that $\mathcal{U}_p(\Sigma)$ is in fact non-invertible. Here, we have defined

$$\mathcal{C}_N(\Sigma) = \int_{\phi,\tilde{\phi},\alpha,\tilde{\alpha}} \exp\left[ -\frac{iN}{2\pi} \oint_\Sigma \left( \alpha\, d\tilde{\phi} + \tilde{\alpha}\, d\phi + b\,\tilde{\phi} - c\,\tilde{\alpha} \right) \right], \tag{E.30}$$

which is a surface defect along which we gauge $\widetilde{G} = \mathbb{Z}_N^{(0)} \times \mathbb{Z}_N^{(1)}$. Indeed, integrating out $\alpha_\mu$ and $\phi$ turn $\tilde{\phi}$ and $\tilde{\alpha}_\mu$ into $\mathbb{Z}_N$ fields respectively. The fields $\tilde{\phi}$ and $\tilde{\alpha}_\mu$ are then coupled to $b_{\mu\nu}$ and $c_\mu$, respectively, so that integrating over $\tilde{\phi}$ and $\tilde{\alpha}_\mu$ amounts to summing over all possible Wilson surfaces for $b_{\mu\nu}$,

$$U(\Sigma)^q = \exp\left( iq \oint_\Sigma b \right), \qquad q \in \mathbb{Z}, \tag{E.31}$$

and all possible Wilson lines for $c_\mu$,

$$W_c(\Gamma)^{\tilde{q}} = \exp\left( i\tilde{q} \oint_\Gamma c \right), \qquad \tilde{q} \in \mathbb{Z}, \tag{E.32}$$

where $\Gamma$ is a loop along $\Sigma$. Recall that Eq. (9.13) has a global symmetry $\widetilde{G} = \mathbb{Z}_N^{(0)} \times \mathbb{Z}_N^{(1)}$, and $U(\Sigma)$ is the symmetry operator for $\mathbb{Z}_N^{(0)}$ while $W_c(\Gamma)$ acts with $\mathbb{Z}_N^{(1)}$. Along the defect $\mathcal{C}_N(\Sigma)$, we are then summing over all possible symmetry operators for $\widetilde{G}$, which means we are gauging this symmetry along $\Sigma$. Thus, $U(\Sigma)$ or $W_c(\Gamma)$ may freely disappear along the defect $\mathcal{C}_N(\Sigma)$.

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
