# Peer review of "Intertwined order of generalized global symmetries"

_SciPost Physics_

## Round 1 · Referee Report · Anonymous (Referee 1) · 2025-1-21

Strengths

1- describes a very interesting novel model demonstrating an intriguing interplay between two different discrete symmetries 2- analyzes the complete phase diagram, using both lattice and continuum methods 3- the resulting picture holds together very nicely and combines many topics of great current interest: generalizeds symmetries, topological phases and dualities

Weaknesses

1- In the end the findings are probably not all that surprising. The main claim to novelty is that the two symmetries interact in a quite unusual way, the model definitely does not just describe two completely independent symmetries. There is some truth to this. Also, it is just overall a very interesting and novel model even if there may be no great new "mechanisitic insights", so this is not really that much of a weakness

Report

This is a very nice and interesting model, describing in all detail the interplay between two symmetries by coupling a "gauge theory" to a "matter theory" by an interesting topological interaction term. The resulting picture of the complete phase diagram, using both lattice and continuum methods, is definitely interesting and nicely touches upon many topics of current interest in formal condensed matter theory.

Requested changes

1- none

Recommendation

Publish (easily meets expectations and criteria for this Journal; among top 50%)

---

## Round 1 · Referee Report · Anonymous (Referee 2) · 2025-3-11

Strengths

1-Analyzes the phase diagram of a 2+1d Euclidean lattice model with a topological $\theta$-term using dualities and generalized symmetries.

2-Explores the possible boundary phases of this model.

3-Turns the $\theta$-term into a $\mathbb Z_N$ axion and uncovers a non-invertible symmetry in the gauged version of this model.

Weaknesses

1-This is not the first time this model, or the $\theta$-term, is discussed. I suggest the authors take a look at "Alfred Shepere, Frank Wilczek (1989), Self-dual models with theta terms, Nuclear Physics B, 320(3), 669-695." Specifically, in eq (3.5), they write down the Euclidean Villain partition function for a $d$-dim $p$-form version of the model studied in the current paper. The authors should cite this paper and clearly explain what is new in their work.

Report

The authors were very thorough in analyzing various aspects of this lattice model. The resulting phase diagram, including the oblique phases, is mostly well known, but coupling to background gauge fields brings out some new features, such as the nontrivial SPT phase when $\theta/2\pi$ is not a multiple of $N$. Most of the phases and transitions at rational values of $\theta/2\pi$ follow from the known phases and transitions at $\theta = 0$ via the $\mathcal S$ and $\mathcal T$ operations. The authors also provide possible candidates for the boundary states of the various phases. Finally, they turn the coupling $\theta$ into an axion-like $\mathbb Z_N$-valued field with its own 0-form symmetry that has a mixed anomaly with the rest of the symmetries. Gauging the rest of the symmetries turns the axion 0-form symmetry into a non-invertible symmetry with interesting fusion rules.

Overall, this is a very nice paper that explores generalized symmetries, anomalies, and their consequences on the phase diagram. I have a number of questions/concerns/changes that I list below.

Requested changes

1-On page 5, and elsewhere, the authors say that "the 1-form symmetry cannot be spontaneously broken along the 1+1d boundary." This requires a citation, or at least a clarification in the footnote. Is this related to the fact that there is no topological order in 1+1d?

2-I am confused by (2.28). I would expect that $\mathcal S^2 =1$, which is consistent with the first line of (2.28). However, I would also expect that $\mathcal T^N=1$, which doesn't seem to be consistent with the second line of (2.28). Can the authors clarify?

3-In (3.9), $\Gamma$ is necessarily contractible because $\Gamma = \partial\Sigma$. However, in (3.10), $\Gamma$ doesn't have to be contractible, i.e., there is no need for $\Sigma$. The operator $e^{\frac{iN}L \oint_\Gamma a}$ is gauge invariant for any closed loop $\Gamma$ (contractible or not).

4-Similar to the above comment, in (3.11), there is no need of the string and the other end point. The operator $e^{\frac{iN}L \varphi(\mathcal P)}$ is already gauge invariant.

5-At the end of sec 3, the authors say that whenever the two symmetries are broken to a nontrivial subgroup, they are broken to the same subgroup. Is this just an observation, or is there a proof of this fact? Also, is this a property of this particular lattice model? Because, in the general $\mathbb Z_N$ clock model, as well as the $\mathbb Z_N$ lattice gauge theory, there are $\lfloor N/2 \rfloor$ independent parameters, and one can certainly imagine a phase where the two symmetries are broken to different subgroups.

6-I don't think writing $c_\mu = -kA_\mu/L \mod N$ below (4.4) is valid. One can always add a multiple of $N/L$ to $c_\mu$ while still satisfying (4.2). However, this ambiguity doesn't affect the action because $B_{\mu \nu}$ is also an integer multiple of $L$. This should be clarified.

7-In (5.1), and elsewhere in the paper, the authors refer to $\Phi$ as a background scalar field. This is incorrect. Say one starts with the action $iN/2\pi \int c dA$, where $c$ is a dynamical 1-form $U(1)$ gauge field which ensures that $A$ is a background 1-form $\mathbb Z_N$ gauge field. Integrating by parts, we can write the action as $iN/2\pi \int A dc \leftrightarrow i/2\pi \int (N A \rho - \phi d\rho)$, where $\phi$ is a Lagrange multiplier (hence, dynamical) that imposes $d\rho = 0\implies \rho = dc$. Integrating the second term by parts gives $i/2\pi \int \rho(NA-d\phi)$ where $\phi$ is dynamical, not background. Similar comments apply to $NB = dC$.

Some minor comments:

1-Below (2.34), "integer multiples of $\theta/2\pi$" should be "integer values of $\theta/2\pi$".

2-In (2.36), $a_\mu$ should be $k_\mu$.

3-Above (4.4), the references to Eq. (4.3) should be Eq. (4.2).

Recommendation

Ask for minor revision

  • validity: high
  • significance: good
  • originality: good
  • clarity: high
  • formatting: excellent
  • grammar: excellent

Author:  Benjamin Moy  on 2025-03-31  [id 5324]

(in reply to Report 2 on 2025-03-11)

We thank the referee for detailed comments and a close reading of our manuscript. We provide detailed responses below to the specific points raised.

Referee:

1-This is not the first time this model, or the $\theta$-term, is discussed. I suggest the authors take a look at "Alfred Shepere, Frank Wilczek (1989), Self-dual models with theta terms, Nuclear Physics B, 320(3), 669-695." Specifically, in eq (3.5), they write down the Euclidean Villain partition function for a $d$-dim $p$-form version of the model studied in the current paper. The authors should cite this paper and clearly explain what is new in their work.

Our response: We thank the referee for bringing up this paper. Indeed, Shapere and Wilczek introduced this model in their paper, and they demonstrated that the model is self-dual, though they did not explore its global symmetries or the topological physics of its phases as we do in our paper. In Section 1, we have added a citation to their paper (and in Section 2.1, where we introduce the model) and an explanation of how we go beyond their work.

Referee:

1-On page 5, and elsewhere, the authors say that "the 1-form symmetry cannot be spontaneously broken along the 1+1d boundary." This requires a citation, or at least a clarification in the footnote. Is this related to the fact that there is no topological order in 1+1d?

Our response: Since we are dealing with a discrete one-form symmetry, yes, this fact is related to the nonexistence of topological order in (1+1)d. We have added two citations in the text where this fact is mentioned, and we also included a footnote on p. 5 where it is first stated.

Referee:

2-I am confused by (2.28). I would expect that $\mathcal{S}^2=1$, which is consistent with the first line of (2.28). However, I would also expect that $\mathcal{T}^N=1$, which doesn't seem to be consistent with the second line of (2.28). Can the authors clarify?

Our response: The condensed charges here are labeled by their embedding in a $U(1)$ theory. In a $U(1)$ theory, we do not expect $\mathcal{T}^N=1$. It is only after recalling the $\mathbb{Z}_N$ nature of the theory and analyzing the allowed probes that it is clear that $\mathcal{T}^N=1$. For example, the phase with $(0,1)$ condensed is trivial. Acting with $\mathcal{T}$, the $(-N,1)$ phase is an SPT, signaled by the fact that an open surface version of the symmetry operator for the 0-form symmetry ends on charge $-1$ electric loops. Acting on the $(0,1)$ phase with $\mathcal{T}^N$, though, gives the $(-N^2,1)$ phase, which is trivial—the open surfaces now end on charge $-N$ electric loops, which are trivial in the $\mathbb{Z}_N$ theory.

Referee:

3-In (3.9), $\Gamma$ is necessarily contractible because $\Gamma=\partial\Sigma$. However, in (3.10), $\Gamma$ doesn't have to be contractible, i.e., there is no need for $\Sigma$. The operator $e^{i\frac{N}{L}\oint_\Gamma a}$ is gauge invariant for any closed loop $\Gamma$ (contractible or not).

4-Similar to the above comment, in (3.11), there is no need of the string and the other end point. The operator $e^{i\frac{N}{L}\varphi(\mathcal{P})}$ is already gauge invariant.

Our response: We agree, which is why we referred to Eq. (3.10) as a "genuine loop operator" and Eq. (3.11) as a "genuine local operator". We have revised the discussion in this section so that it is clearer.

Referee:

5-At the end of sec 3, the authors say that whenever the two symmetries are broken to a nontrivial subgroup, they are broken to the same subgroup. Is this just an observation, or is there a proof of this fact? Also, is this a property of this particular lattice model? Because, in the general $\mathbb{Z}_N$ clock model, as well as the $\mathbb{Z}_N$ lattice gauge theory, there are $\lfloor N/2 \rfloor$ independent parameters, and one can certainly imagine a phase where the two symmetries are broken to different subgroups.

Our response: The argument is to consider the action of the most general set of $\mathcal{S}$ and $\mathcal{T}$ transformations, given in Eq. (2.39), on the phases at $\Theta=0$. The resulting phase generically has an effectively field theory that is a multi-component version of Eq. (2.37). One can explicitly check that if either the $\mathbb{Z}_N$ zero-form or $\mathbb{Z}_N$ one-form symmetries is broken to a nontrivial subgroup, then both symmetries are broken to the same subgroup. We have added a footnote sketching this proof.

Yes, other terms can be added to the action that result in phases that break the symmetries to different subgroups.

Referee:

6-I don't think writing $c_\mu=-k A_\mu/L$ mod $N$ below (4.4) is valid. One can always add a multiple of $N/L$ to $c_\mu$ while still satisfying (4.2). However, this ambiguity doesn't affect the action because $B_{\mu\nu}$ is also an integer multiple of $L$. This should be clarified.

Our response: We meant to write $c_\mu=-k A_\mu/L$ mod $N/L$ below Eq. (4.4). We have corrected this statement and added a footnote clarifying that this ambiguity does not affect the response action.

Referee:

7-In (5.1), and elsewhere in the paper, the authors refer to $\Phi$ as a background scalar field. This is incorrect. Say one starts with the action $iN/2\pi\int cdA$, where $c$ is a dynamical 1-form $U(1)$ gauge field which ensures that $A$ is a background 1-form $\mathbb{Z}_N$ gauge field. Integrating by parts, we can write the action as $iN/2\pi\int Adc \leftrightarrow i/2\pi \int (NA\rho-\phi d\rho)$, where $\phi$ is a Lagrange multiplier (hence, dynamical) that imposes $d\rho=0\implies \rho=dc$. Integrating the second term by parts gives $i/2\pi \int \rho(NA-d\phi)$ where $\phi$ is dynamical, not background. Similar comments apply to $NB=dC$.

Our response: We thank the referee for bringing this issue to our attention and have corrected it in the manuscript.

Referee:

Some minor comments:

1-Below (2.34), "integer multiples of $\theta/2\pi$" should be "integer values of $\theta/2\pi$".

2-In (2.36), $a_\mu$ should be $k_\mu$.

3-Above (4.4), the references to Eq. (4.3) should be Eq. (4.2).

Our response: We have now corrected these misprints.

---

## Editorial Decision

resubmitted